# Pioneer *neurog1* expressing cells ingress into the otic epithelium and instruct neuronal specification

Esteban Hoijman[1]*[†][‡], L Fargas[1][†], Patrick Blader[2], Berta Alsina[1]*

[1]Department of Experimental and Health Sciences, Universitat Pompeu Fabra, Barcelona, Spain; [2]Centre de Biologie du Développement, Centre de Biologie Intégrative, Université de Toulouse, CNRS, UPS, Toulouse, France

**Abstract** Neural patterning involves regionalised cell specification. Recent studies indicate that cell dynamics play instrumental roles in neural pattern refinement and progression, but the impact of cell behaviour and morphogenesis on neural specification is not understood. Here we combine 4D analysis of cell behaviours with dynamic quantification of proneural expression to uncover the construction of the zebrafish otic neurogenic domain. We identify pioneer cells expressing neurog1 outside the otic epithelium that migrate and ingress into the epithelialising placode to become the first otic neuronal progenitors. Subsequently, neighbouring cells express neurog1 inside the placode, and apical symmetric divisions amplify the specified pool. Interestingly, pioneer cells delaminate shortly after ingression. Ablation experiments reveal that pioneer cells promote neurog1 expression in other otic cells. Finally, ingression relies on the epithelialisation timing controlled by FGF activity. We propose a novel view for otic neurogenesis integrating cell dynamics whereby ingression of pioneer cells instructs neuronal specification.

*For correspondence: esteban. hoijman@crg.eu (EH); berta. alsina@upf.edu (BA)

[†]These authors contributed equally to this work

**Present address:** [‡]Centre for Genomic Regulation, Barcelona, Spain

**Competing interests:** The authors declare that no competing interests exist.

## Introduction

Neural specification relies on proneural genes, which are expressed in specific patterns and underlie the genesis, organisation and the function of the neurons that will subsequently differentiate (*Bertrand et al., 2002*; *Huang et al., 2014*). Many signals that pattern the nervous system have been identified. For example, gradients of Shh, BMP and Wnt establish thirteen different domains of neural progenitors in the mouse neural tube (*Ulloa and Briscoe, 2007*); FGF8 and FGF3 control the site of retinogenesis initiation in chick and fish through regulation of *ath5* expression (*Martinez-Morales et al., 2005*); and EGFR signalling determines the expression of a wave of *l(1)sc* in the *Drosophila* optic lobe (*Yasugi et al., 2010*).

Concomitant with cell specification, neural tissues undergo phases of morphogenesis and/or growth. Thus, the cells within a given domain are not static but perform complex cell behaviours. Recently, the contribution of such cell dynamics to neural patterning has been identified. In the neural tube, for instance, sharply bordered specification domains involve the sorting of cells along a rough Shh-dependent pattern (*Xiong et al., 2013*). Additionally, differences in the rate of differentiation of cells (which migrate out of the tissue) between distinct domains of the neural tube help to establish the overall pattern during tissue growth (*Kicheva et al., 2014*). Thus, dynamic spatial rearrangements of cells within a field that is being specified are integrated with patterning mechanisms of positional information by morphogens.

In the inner ear, developmental defects in neurogenesis could result in congenital sensorineural hearing loss (*Manchaiah et al., 2011*). Neurogenesis begins when an anterior neurogenic domain appears at the placode stage by the expression of the proneural gene *neurog1*, which specifies

**eLife digest** The inner ear is responsible for our senses of hearing and balance, and is made up of a series of fluid-filled cavities. Sounds, and movements of the head, cause the fluid within these cavities to move. This activates neurons that line the cavities, causing them to increase their firing rates and pass on information about the sounds or head movements to the brain. Damage to these neurons can result in deafness or vertigo. But where do the neurons themselves come from?

It is generally assumed that all inner ear neurons develop inside an area of the embryo called the inner ear epithelium. Cells in this region are thought to switch on a gene called *neurog1*, triggering a series of changes that turn them into inner ear neurons. However, using advanced microscopy techniques in zebrafish embryos, Hoijman, Fargas et al. now show that this is not the whole story.

While zebrafish do not have external ears, they do possess fluid-filled structures for balance and hearing that are similar to those of other vertebrates. Zebrafish embryos are also transparent, which means that activation of genes can be visualized directly. By imaging zebrafish embryos in real time, Hoijman, Fargas et al. show that the first cells to switch on *neurog1* do so outside the inner ear epithelium. These pioneer cells then migrate into the inner ear epithelium and switch on *neurog1* in their new neighbors. A substance called fibroblast growth factor tells the inner ear epithelium to let the pioneers enter, and thereby controls the final number of inner ear neurons.

The work of Hoijman, Fargas et al. reveals how coordinated activation of genes and movement of cells gives rise to inner ear neurons. This should provide insights into the mechanisms that generate other types of sensory tissue. In the long term, the advances made in this study may lead to new strategies for repairing damaged sensory nerves.

neuronal precursors. The rest of the otic placode is non-neurogenic and generates non-neuronal cell types (*Ma et al., 1998*; *Andermann et al., 2002*; *Abello and Alsina, 2007*; *Radosevic et al., 2011*). In the neurogenic domain, *neurog1* induces *neurod1* (*Ma et al., 1996*, *1998*) expression, which is required for delamination of neuroblasts from the epithelium (*Liu et al., 2000*). Delaminated neuroblasts subsequently coalesce to form the statoacoustic ganglion (SAG) and differentiate into mature bipolar neurons (*Hemond and Morest, 1991*; *Haddon and Lewis, 1996*). The spatial restriction of the otic neurogenic domain relies on the integration of diffusible signals such as FGFs, SHH, Retinoic acid and Wnt (reviewed in *Raft and Groves, 20142015*) as well as the function of transcription factors such as Tbx1 (*Radosevic et al., 2011*; *Raft et al., 2004*), Sox3 (*Abelló et al., 2010*), Otx1 (*Maier and Whitfield, 2014*), Eya1 (*Friedman et al., 2005*) and Six1 (*Zou et al., 2004*). In the inner ear, several FGFs (*Adamska et al., 2001*; *Mansour et al., 1993*; *Léger et al., 2002*; *Alsina et al., 2004*; *Vemaraju et al., 2012*; *Alvarez et al., 2003*), regulate the sequential steps of neurogenesis starting from the expression of *neurog1* (*Vemaraju et al., 2012*; *Léger et al., 2002*; *Alsina et al., 2004*) and continuing to later events involving neuroblast expansion (*Vemaraju et al., 2012*). Together with the regulation of spatial regionalisation, the number of neuronal progenitors produced depends on local cell–cell interactions mediated by the Notch pathway (*Adam et al., 1998*). Remarkably, to date no studies have addressed how morphogenesis, cell behaviour and proneural dynamics impact otic neuronal specification.

Here we use the zebrafish inner ear as a model to analyse the role of cell dynamics on neuronal specification. We identify pioneer cells that are specified outside the otic epithelium, ingress into the placode during epithelialisation and control local neuronal specification, suggesting an instructive role of these cells. Furthermore, we show that FGF signalling affects otic neurogenesis through the regulation of otic placode morphogenesis, influencing pioneer cell ingression.

## Results

### Visualising neuronal specification dynamics

We have previously identified cell behaviours contributing to otic vesicle morphogenesis (*Hoijman et al., 2015*) and here we focused on the influence of cell dynamics in the establishment of the neurogenic domain. For this, we used a zebrafish BAC reporter line that expresses the

fluorescent protein DsRed-Express (DsRedE, a faster maturation version of DsRed [*Bevis and Glick, 2002*]) under control of the *neurog1* regulatory elements (*Drerup and Nechiporuk, 2013*). We imaged in 4D the otic development from stages of otic placode morphogenesis (15 hpf) until neuroblast delamination is abundant and the central lumen is expanding (20.5 hpf, *Figure 1A and B*; *Videos 1* and *2*). The overall pattern of DsRedE expression is highly consistent between embryos, being restricted to the most ventroanterolateral region of the placode until 19 hpf and expanding

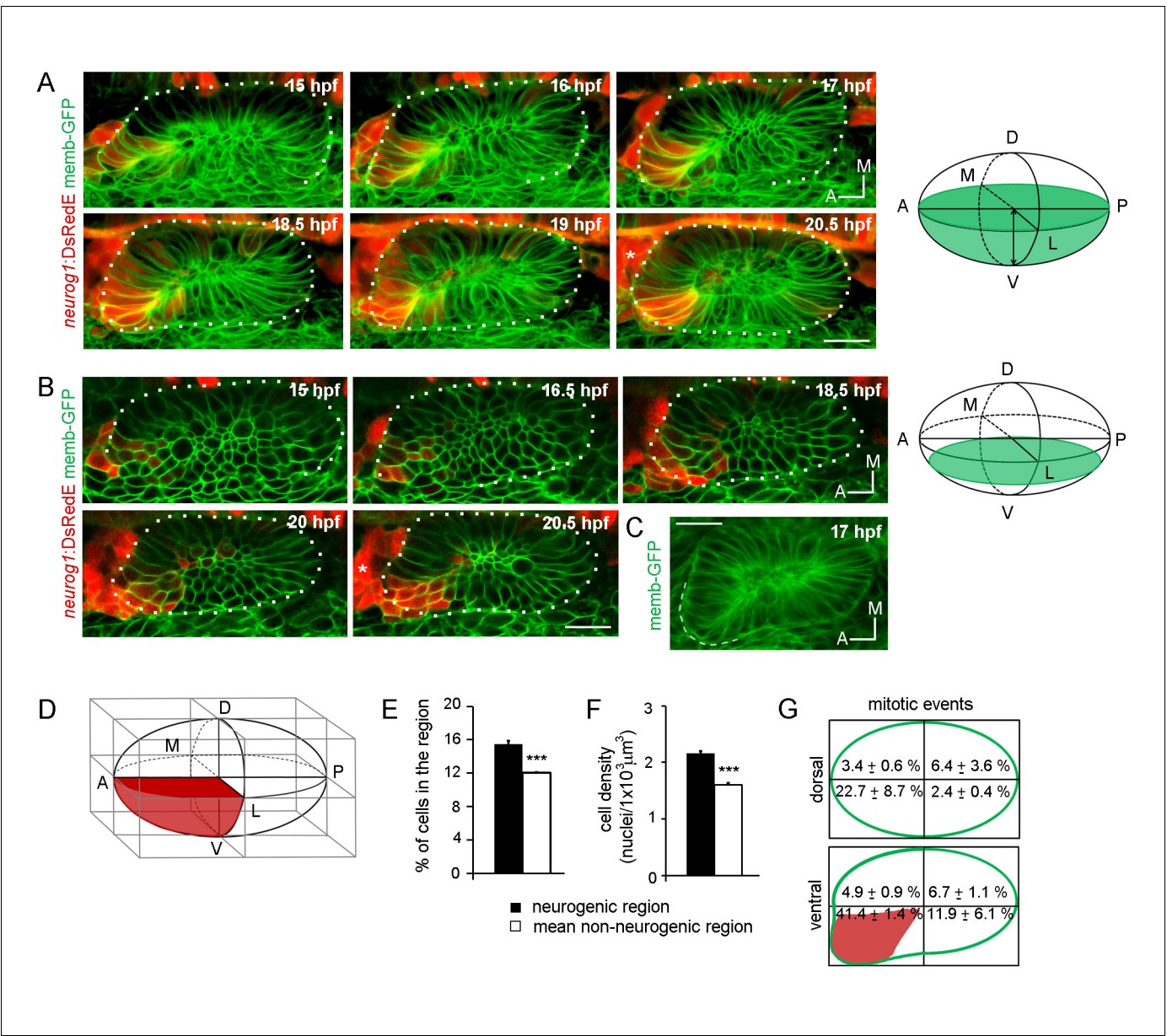

**Figure 1.** Specification dynamics and morphogenesis of the otic neurogenic domain. (**A,B**) Selected frames of a video of an otic placode from a TgBAC (*neurog1:DsRedE*)[n16] embryo shown in (**A**) 3D reconstructions (dorsal view) and (**B**) coronal ventral planes. Green in the right schemes shows the region imaged. Membranes are stained with memb-GFP. D:dorsal, V:ventral, A:anterior, P:posterior, M:medial and L:lateral. The asterisk indicates the region where the SAG is forming. Medial to the otic vesicle, DsRedE is also expressed in the neural tube. (**C**) Averagez-projection (dorsal view) of the inner ear at 17 hpf. Dashed line indicates the protuberance. (**D**) Scheme of the rectangular cuboid used for quantifications. Neurogenic region is shown in red. (**E, F,G**) Quantification of the number of cells (**E**), the cellular density (**F**) and mitotic events (**G**) in the indicated regions at 19 hpf (n = 11) (**E,F**) or between 14 and 18.5 hpf (n = 2) (**G**). Data are mean ± s.e.m. ***p<0.0001 one sample t-test in (**E**) and unpaired t-test (**F**). Scale bars, 20 µm. Dotted lines outline the limits of the otic vesicle.

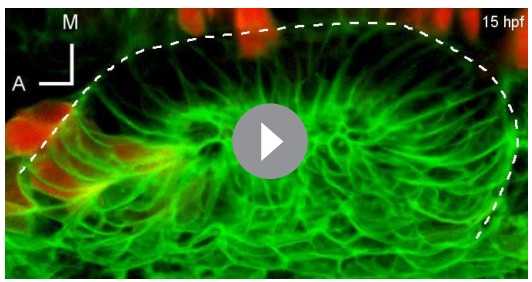

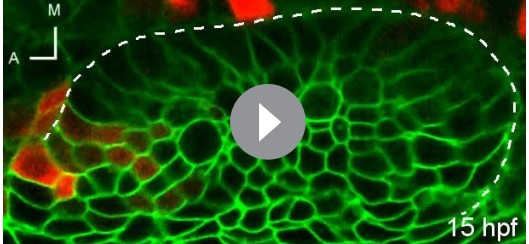

**Video 1.** 4D imaging of otic neuronal specification. 3D reconstructed time-lapse of the otic vesicle from a TgBAC(neurog1:DsRedE)[n16] embryo. Red: DsRedE fluorescence. Green: memb-GFP. Dorsal view. Time from the first frame is indicated.

**Video 2.** Specification dynamics visualized in individual cells. Selected coronal ventral planes from the z-stacks used for 3D reconstructions in **Video 1**.

posteromedially at around 20.5 hpf (**Figure 1A and B**; **Videos 1** and **2**). This DsRedE expression pattern recapitulates the endogenous spatiotemporal pattern of neurog1 as analysed by in situ hybridisation (ISH) (**Radosevic et al., 2014**; **Vemaraju et al., 2012**; **Andermann et al., 2002**). Moreover, DsRedE expressing cells delaminate (Figure 3H; **Videos 1** and 11) and are incorporated into the SAG (**Figure 1A and B**; **Video 3**), supporting the use of this line to analyse single cell dynamics of neuronal specification.

We also analysed the cellular organisation of the neurogenic domain by performing a 3D morphometric analysis of this region. During the stages of neuronal specification, the shape of the otic vesicle is asymmetric, exhibiting a protuberance in the anterolateral region (**Figure 1C**). To compare the properties of the neurogenic region with the rest of the otic vesicle, we built a rectangular cuboid with the vertices of the vesicle and divided it in eight regions of equal volume (**Figure 1D**), in which we quantified the number of cells and the volume of tissue. By 19 hpf, the neurogenic domain region accumulated more cells (15.4 ± 0.4% of the total number of cells in the vesicle, 49 ± 3 cells of 311 ± 16 cells respectively) than other regions (mean non-neurogenic region: 12.0 ± 0.1%, 36 ± 2 cells, **Figure 1E**) and presented higher cellular density (**Figure 1F**; neurogenic region: 2.16 ± 0.03 nuclei/$1 \times 10^3$ $\mu m^3$, mean non-neurogenic region: 1.60 ± 0.03 nuclei/$1 \times 10^3$ $\mu m^3$). Quantification of all the mitotic events inside the vesicle between 14 and 18.5 hpf revealed that cell proliferation is also highly enriched in this region (**Figure 1G**). While the increase in cell number in the neurogenic domain was moderate (about 3% more cells than other regions), the enrichment in mitotic events led to about 41% of the total number of divisions to occur in this domain. Thus, in addition to a phase of transit-amplification of neuroblasts after delamination (**Vemaraju et al., 2012**), neuronal progenitors also appear to multiply inside the otic vesicle. This analysis indicates that the neurogenic domain presents high cell number, high cell density and an increased proliferative activity.

## The first otic neurogenic cells are specified outside the otic epithelium and ingress during placode formation

To analyse how the neurogenic domain is built, we decided to evaluate when and where cells of the neurogenic domain start to express neurog1. We first aimed to capture the earliest specified cells. Epithelialisation of the otic placode progresses from 12.5 hpf until about 18 hpf (**Hoijman et al., 2015**). While it has been reported that neurog1 expression in the otic placode begins at 15 hpf (**Radosevic et al., 2014**), we found that already at 13 hpf there are rows of DsRedE expressing cells lateral to the neural tube and anterior to the epithelializing otic placode (**Figure 2A**; **Video 4**). These cells coincide

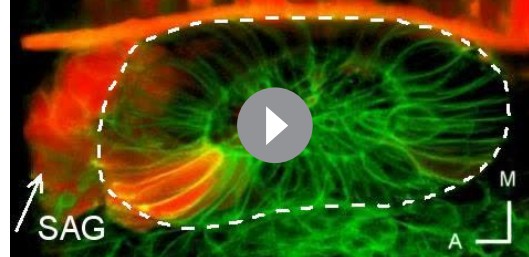

**Video 3.** neurog1 expressing cells locate in the SAG after delamination. 3D reconstruction of the otic vesicle at 21 hpf. White arrow indicates the position of the SAG.

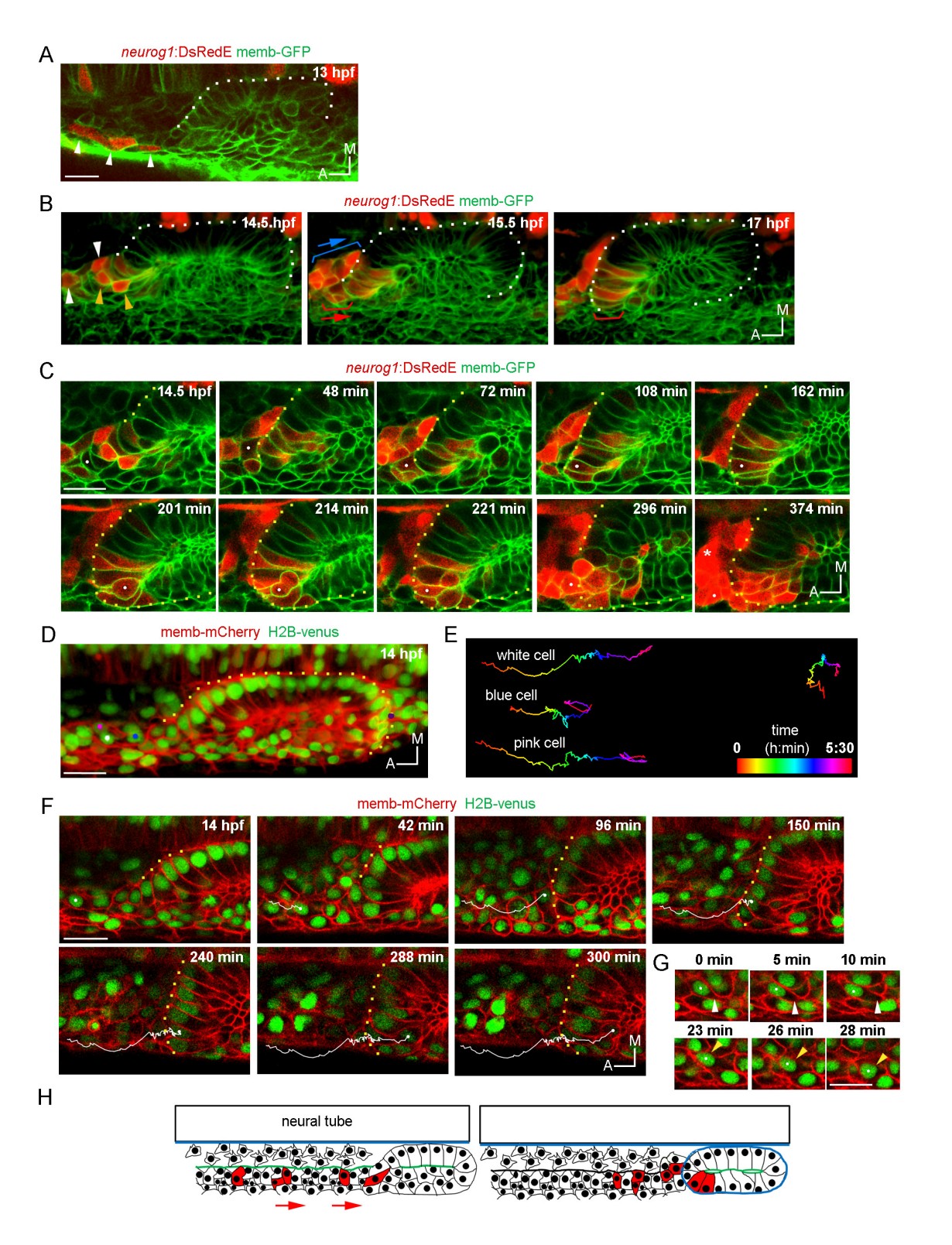

**Figure 2.** Ingression of *neurog1*+ cells. (**A**) The otic epithelium and its anterior region at 13 hpf. Arrowheads highlight *neurog1*+ cells outside the otic epithelium. (**B**) Selected frames of a 3D reconstruction (dorsal view) of the otic placode following the movement of the anterior *neurog1*+ cells. Arrowheads at 14.5 hpf indicate *neurog1*+ cells before epithelialisation (white: cells outside the placode, orange: ingressing cells). At 15.5 hpf red bracket identifies cells that will ingress (shown at 17 hpf) and blue bracket cells that will not ingress. In (**A**) and (**B**) the contrast of the red signal was

*Figure 2 continued on next page*

*Figure 2 continued*

increased to improve visualisation. (**C**) Selected planes of a 3D tracking of a single cell specifying during ingression (white dot). At 108 min the cell is already epithelialised. Asterisk indicates the SAG. (**D–F**) 3D tracking of single cells during ingression. (**D**) 3D reconstruction (dorsal view) showing the initial position of the tracked cells (white, pink and blue dots) at 14 hpf. The violet dot indicates the posterior vertex of the placode. (**E**) 2D visualisation of the 3D tracks shown in (**D**) are displayed in a temporal color code. Each track was displaced in the y axis for better visualisation. The track of the posterior vertex of the placode is shown on the right (see also *Figure 2D*). (**F**) Selected frames for the cell of the white track. At 150 min the cell is ingressing and completed at 240 min. At 300 min cytokinesis occurs. Membranes are stained with memb-mCherry. Embryos are Tg(*actb:H2B-venusFP*). (**G**) Selected planes showing cell-membrane displacements during migration of the cell tracked in (**F**). White arrowheads indicate protrusion of the cell front and orange arrowheads the position of the nucleus. (**H**) Schematic representation of the migration and ingression during epithelialisation (see *Figure 2—figure supplement 1* for further details). Blue line: laminin, green line: actin layer, red cells: *neurog1*[+] cells, red arrows: migration of *neurog1*[+] cells towards the otic placode. Scale bars, 20 μm. Dotted lines outline the limits of the otic vesicle.

The following figure supplement is available for figure 2:

**Figure supplement 1.** Morphogenetic features related to ingression.

with *neurog1* expressing cells detected by ISH (*Figure 2—figure supplement 1A*), and previously assumed to belong to the anterior lateral line placode (*Andermann et al., 2002*). Unexpectedly, when we followed these cells we found that some of them migrate posteriorly and become incorporated into the anterolateral region of the otic epithelium, in a position corresponding to the neurogenic domain (red brackets in *Figure 2B*; *Video 5*). Therefore, these cells develop into otic and not lateral line cells. To confirm this cell ingression, we injected NLS-Eos mRNA at 1 cell stage to obtain a homogeneous nuclear staining with the photoconvertible protein throughout the embryo. At 13 hpf, we photoconverted Eos protein (from green to red fluorescence) in a group of nuclei anterior to the otic epithelium where the migrating cells are located. At 20 hpf, we detected photoconverted nuclei inside the vesicle (*Figure 2—figure supplement 1B*).

We also detected in the same anterior region a second pool of *neurog1*[+] cells (expressing also *neurod1*; *Figure 2—figure supplement 1C*) that moves posteromedially without ingressing, remaining in the region of the SAG (blue brackets in *Figure 2B*; *Video 5*). The migrating cells are located laterally relative to a population of sparse cells from which they are segregated by an F-actin rich layer that runs anteroposteriorly until it reaches the placode (*Figure 2—figure supplement 1F* and *Figure 2H*). These observations suggest that *neurog1* expression is not sufficient for cell ingression. Additionally, *neurog1* expression was not required for cell ingression, as some *neurog1*- cells ingress. Consistently, we detected cell ingression events in *neurog1* mutant embryos (*neurog1*[hi1059], *Figure 2—figure supplement 1G*).

Interestingly, 3D tracking of individual cells of the ingressing pool revealed that some cells activate *neurog1* expression while moving towards the epithelium and before their epithelialisation (*Figure 2C*; *Video 6*). Immediately after ingressing into the neurogenic domain, these cells divide and delaminate, thus undergoing a complete cycle of epithelialisation and de-epithelialisation in only a few hours. Analysis of the movement of these cells suggests that their migration is a directional process occurring in individual cells (*Figure 2D,E and F*; *Video 7*; some cells of the same region migrate in other directions). We also observed that the leading front of cells periodically protrudes, followed by a rapid forward translocation of the nucleus (*Figure 2G*; insets of *Video 7*), as has been described during fibroblast migration (*Petrie and Yamada, 2015*). When tracking three neighbouring cells, we observed that while two of them ingress (white and pink tracks), the third one (blue track), which is initially positioned closer to the otic placode, divides during migration and the daughters do not ingress (*Figure 2D,E and F*; *Video 7*). These observations highlight that ingressing cells are interspersed with other cells that do not join the otic placode, and factors other than anteroposterior positional cues within the migrating population determine whether a cell will ingress or not into the otic placode.

Particular morphogenetic features could facilitate the ingression of cells from the anterior region. As we previously reported, the otic placode is only epithelialised medially at these stages (*Hoijman et al., 2015*). As epithelialisation progresses, at 14 hpf the posterior part of the placode is segregated from the surrounding cells, while the anterior region of the placode is not (*Figure 2—figure supplement 1D*; *Video 8*). Thus, the posterior part folds approximately 3 hr before the

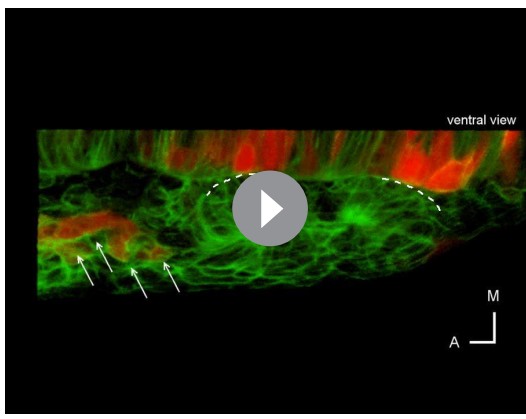

**Video 4.** Early *neurog1* expressing cells located anterior to the otic vesicle. 3D reconstruction of an otic vesicle and the anterior region at 13 hpf, showing the presence of DsRedE expressing cells (white arrows). DOI: 10.7554/eLife.25543.009

anterior one (*Figure 2—figure supplement 1D*; *Video 8*). During this period, and by the anterior unfolded region, migrating cells ingress into the otic epithelium. Moreover, the basal lamina at these early stages is only rudimentary and not continuous (contrary to the one present at later stages surrounding the whole organ; *Figure 2— figure supplement 1E*). Therefore, the fact that the epithelium is still organizing could allow the migrating cells to ingress into the tissue before it is fully formed.

In summary, our results show cells that are being specified outside the otic epithelium, migrate and ingress into the prospective neurogenic domain, constituting the earliest neuronal specified cells of the organ.

## Generation of *neurog1* expressing cells by local specification and cell division

We next evaluated if, in addition to ingressing cells, other cells start to express *neurog1* within the neurogenic domain. We visualised the activation of *neurog1* expression inside the otic vesicle in real-time (*Figure 3A*; *Video 9*), a process that we refer to as 'local specification'. Dynamic quantification of DsRedE fluorescence levels in individual cells ($F_{cell}$) indicated that the rate of increase in the signal is variable among cells (*Figure 3B*, mean rate of increase ranging between 0.15 and 0.54 a.u./min, n = 11 cells). However, we found that when the signal reaches a critical level (between 45.5 and 52.5 a.u. in *Figure 3B*, gray region with red dots), cells begin to delaminate (visualised by the movement of the cell body to the basal domain of the epithelium). This suggests that cells delaminate relative to *neurog1* levels and not to the time elapsed since they initiated *neurog1* expression (*Figure 3B and C*).

As we mentioned above, higher mitotic events occur in the neurogenic domain. Therefore, division could also contribute to the domain by adding *neurog1* expressing cells (*neurog1*[+] cells) to the domain. To address this, we performed a 4D analysis of cell divisions and found that every cell divides only once in the 7 hr period analysed (n = 27/27). Mitotic cells are found either contacting the central lumen (*Figure 3D*) or not (peripheral divisions) (*Figure 3E*). Interestingly, these latter

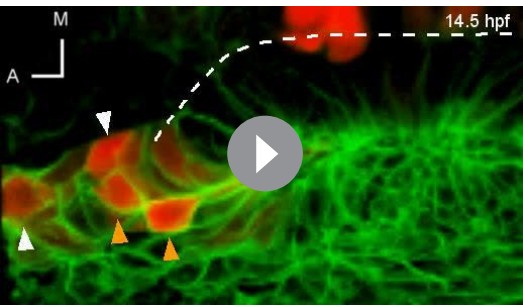

**Video 5.** *neurog1* expressing cells ingress in the otic epithelium. 3D reconstructed time-lapse showing the ingression of *neurog1* expressing cells. Orange arrowheads indicate ingressing cells and white arrowheads cells that are outside the organ. Cells that will ingress are highlighted with a red bracket and the direction of movement by a red arrow. The group of *neurog1* expressing cells that do not ingress is indicated by a blue bracket and arrow. DOI: 10.7554/eLife.25543.010

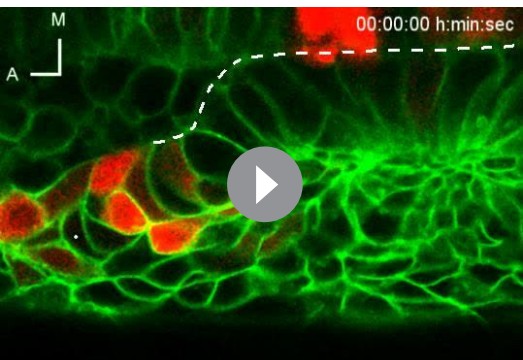

**Video 6.** 3D tracking of an individual cell during ingression, division and delamination. Coronal ventral planes from z-stacks selected to track an ingressing cell (white dot). Note that it begins to express *neurog1* before epithelialization. DOI: 10.7554/eLife.25543.011

cells are apposed to an accumulation of the apical determinant Pard3 that forms a scaffold perpendicular to the central luminal surface of the vesicle, running from the lumen to the periphery (*Figure 3F*; *Video 10*). Thus, similar to the apical mitosis occurring in the central lumen, peripheral divisions are also in contact with an apical surface (*Figure 3G and H*).

In neurogenic tissues, either asymmetric (daughter cells become one progenitor and one neuron) or symmetric (both daughter cells with the same fate) divisions can occur (*Taverna et al., 2014*; *Chenn and McConnell, 1995*; *Das and Storey, 2012*). This depends on factors such as the apicobasal position of the dividing cell and the orientation of the mitotic spindle (*Das and Storey, 2012*). Our dynamic analysis of *neurog1* activation allowed us to assess the modes of divisions within the otic neurogenic domain. We observed that all divisions in the neurogenic domain have the cleavage plane perpendicular to the apical surface regardless of their position in the epithelium or their *neurog1* expression (*Figure 3G and H*). When analysing the fate of the daughter cells after division, we found all were symmetric (27/27): both daughter cells delaminate after division (20/27 delaminate during the timeframe analysed, 7/27 are positioned to delaminate at the end of the acquisition). However, division can occur either before (13/25) or after (12/25) the induction of *neurog1* expression. Interestingly, daughter cells from mitoses of a *neurog1*$^+$ cell with high levels of DsRedE expression (*neurog1*$^{+Hi}$ cell) rapidly delaminate, remaining in close contact as they move to the periphery of the tissue (*Figure 3I and J*; *Video 11*). On the other hand, daughter cells from mitosis of cells not expressing *neurog1* (*neurog1*$^-$), or only at low levels (*neurog1*$^{+Low}$), remain in the epithelium after division, where they increase the DsRedE signal over a variable period of time (*Figure 3—figure supplement 1*).

In summary, divisions in the neurogenic domain are symmetric and apical. Furthermore, there is not a preferential sequence of events concerning *neurog1* activation and division. Taken together, our analysis of the origin of *neurog1*$^+$ cells revealed that they are added to the neurogenic domain by three different mechanisms: cell ingression, local expression and cell division.

## Ingressing cells instruct neuronal specification

The incorporation of the ingressing cells and their rapid exit from the otic vesicle led us to wonder about their role in the establishment of the neurogenic domain. These early-specified cells might contribute to the neurogenic domain by their inclusion as specified cells and/or play additional roles.

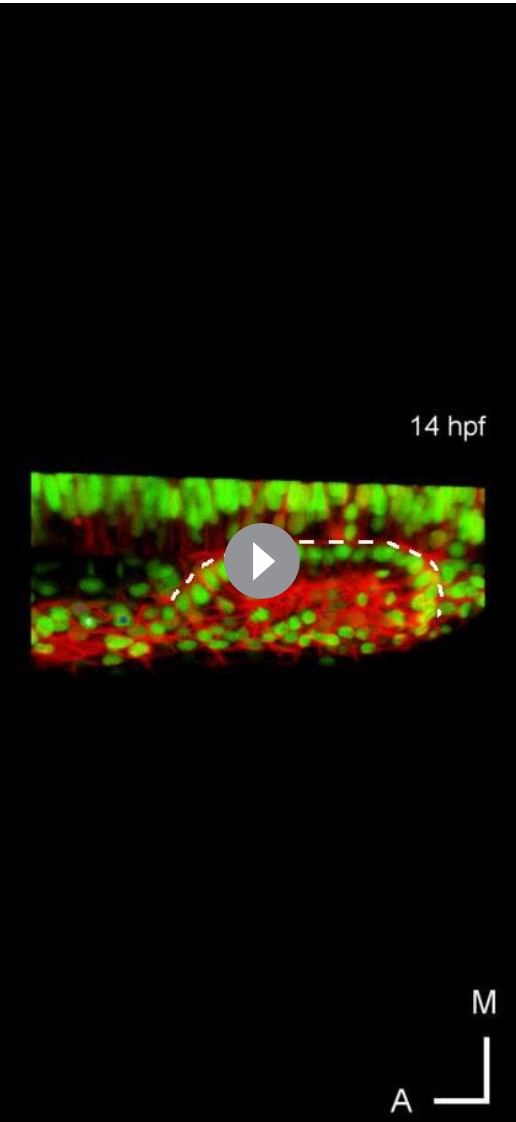

**Video 7.** 3D tracking of multiple cells during ingression. Initially, the position of three cells anterior to the otic epithelium is shown (white, pink and blue dots). Tracking (upper panels) and 2D trajectory of each cell (lower panel, yellow track shows the position of the posterior vertex of the placode) are depicted. Insets highlight the mode of migration, with leading edge of the cell protruding (white arrowheads) before the forward displacement of the nucleus (yellow arrowheads).

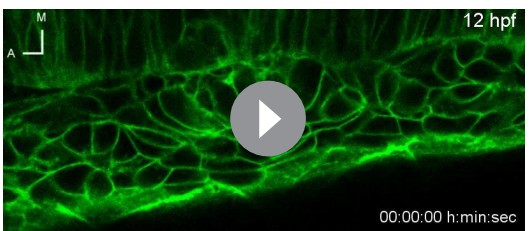

**Video 8.** Detailed view of the morphogenesis of the otic placode. Time-lapse of memb-GFP expressing embryos showing the different stages of tissue epithelialisation. Note that the posterior region folds before the anterior one (orange arrowhead highlights the unfolded anterior region). Lines indicate the epithelialised regions.

To address this question, we decided to eliminate these cells during their migration, before they reach the otic epithelium. For this, we identified the stream of migrating cells by their DsRedE signal (*Figure 4A*), laser-ablated them unilaterally at 12.5 hpf (*Figure 4B*), and examined the effects on neuronal specification in 3D in the otic vesicle at 18.5 hpf (*Figure 4C–H*; *Video 12*), before delamination becomes significant. *Neurog1* expression was analysed by quantification of the $F_{cell}$ in all cells belonging to the neurogenic domain (*Figure 4C and D*). Ablation of a limited number of cells (2–3 cells per laser pulse; see Material and methods for more details) led to a decrease in the global level of DsRedE expression (calculated as the sum of the $F_{cell}$ for all *neurog1*+ cells) in the vesicle of the ablated side as compared to the contralateral vesicle on the non-ablated side of the embryo (*Figure 4C and E*; non-ablated side: 1492 ± 58, ablated side: 454 ± 44 a.u.). Applying an increased number of laser pulses ablated more cells, which seems to lead to a more severe specification phenotype (compare embryos 1 and 2 from *Figure 4C*, which received 1 and 3 laser pulses respectively), despite the overall morphology of the neurogenic domain being unaffected. Analysis of both *neurog1* expression in the otic epithelium at 21 hpf and the phenotype of the SAG at 42 hpf confirms that the effect of ablation persists and, thus, does not appear to represent a delay in neuronal specification (*Figure 4—figure supplement 2A,B and C*; *Video 12*). The effect of ablation is specific to otic *neurog1* expression, since DsRedE expression in the neural tube was not affected (*Figure 4—figure supplement 2D*). Moreover, we observed a phenotype only after ablating anterior future ingressing cells: ablation of *neurog1*+ cells in another location (posterior to the placode at 13 hpf, *Figure 4—figure supplement 1B*) or developmental stage (anterior to the vesicle at 19 hpf, *Figure 4—figure supplement 1C*) did not affect *neurog1* expression in the otic vesicle.

When comparing the number of *neurog1*+ cells (N*neurog1*+), we also found a reduction in the ablated side vesicle compared to the control vesicle (*Figure 4F*; non-ablated side: 23.8 ± 1.4 cells, ablated side: 10.0 ± 0.8 cells). This result could be partially explained by the failure of the ablated cells to ingress into the forming neurogenic domain. These results also indicate that when ablating the cells that will be part of the neurogenic domain, the cells now located in the same position do not change their fate and become neural specified, as expected if cell identity would be dictated by cell position. Interestingly, the number of cells eliminated by ablation (and the ones produced by their divisions) would be too small to account for the large decrease in the number of *neurog1*+ cells in the vesicles of the ablated side (*Figure 4F*). This suggests that ingressing cells play an instructive role on the specification of other cells of the neurogenic domain (i.e. local specification). To shed light on this possibility, we calculated the mean value for $F_{cell}$ ($\bar{F}_{cell}$) in vesicles from each experimental condition. This parameter was also reduced by the ablation (*Figure 4G*; non-ablated side: 60.1 ± 2.5, ablated side: 43.6 ± 4.8 a.u.), suggesting that the global reduction in fluorescence was not only caused by a decrease in the number of *neurog1*+ cells (*Figure 4Iii*), but that the *neurog1* transcriptional activity inside these cells was also reduced. Accordingly, the number of *neurog1*+Hi cells (N*neurog1*+Hi) was also significantly lowered by ablation (*Figure 4H*; non-ablated side: 6.0 ± 0.6, ablated side: 1.0 ± 0.4 cells). However, it is possible that the *neurog1*+Hi cells at the time point analysed are mainly ingressed cells, and thus by eliminating them, we decreased the $\bar{F}_{cell}$ in each vesicle by a relative increase in *neurog1*+Low cells (*Figure 4Iiii*, see figure legend for detailed explanation of the scheme). We discarded this possibility by backtracking cells identified as *neurog1*+Hi at 19 hpf from non-ablated embryos, and observing that most of them are *neurog1*- cells at 13 hpf positioned inside the epithelising placode before ingression takes place, therefore belonging to the pool of cells specified locally (*Figure 4I and J*).

Given that both the number and expression levels of neurog1+ cells were reduced by ablation, it is possible that a cell community effect takes place, in which the presence of more neurog1+ cells

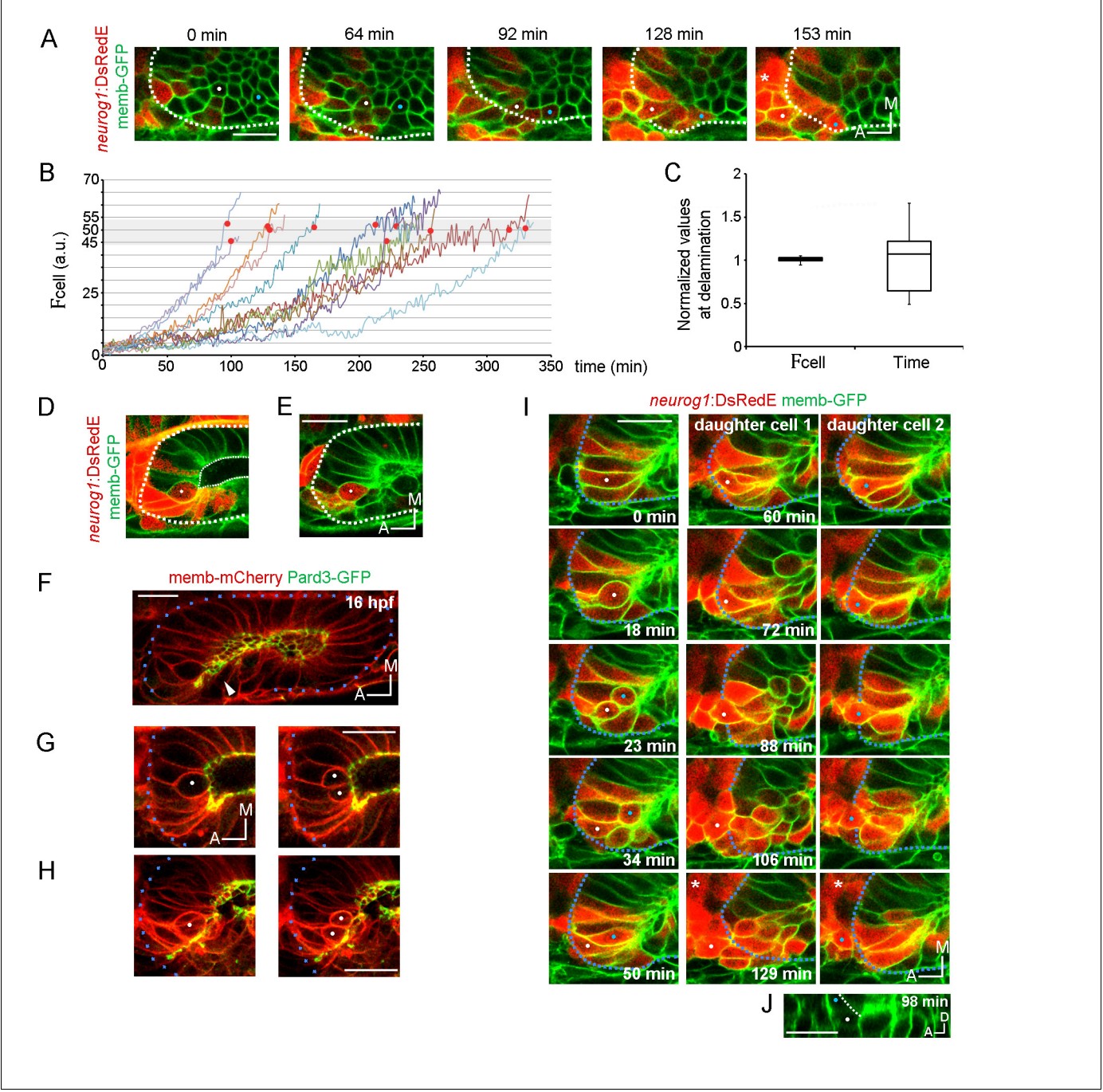

**Figure 3.** Local specification and divisions of *neurog1* expressing cells. (**A**) Selected planes showing DsRedE expression dynamics in locally specified cells (white and blue dots) from TgBAC(*neurog1*:DsRedE)[n16] embryos expressing memb-GFP. Asterisk indicates the SAG. The embryo is 16.5 hpf at the beginning of the time-lapse. (**B**) Quantification of DsRedE fluorescence over time for 11 cells locally inducing *neurog1*. Red dots indicate beginning of delamination. The gray region highlights the interval of fluorescence levels at which all cells delaminate. (**C**) Box plot made from the quantifications shown in (**B**), illustrating that at the moment of delamination, the time elapsed from the initiation of *neurog1* expression is highly variable, while the expression levels are not. The value for each cell was normalized by the mean of the cell group. (**D,E**) *neurog1*+ mitotic cells (white dots) contacting (**D**) or not (**E**) the central lumen (dashed line). 19 (**D**) and 17 (**E**) hpf embryos are shown. (**F**) Pard3-GFP localisation in the central lumen and the anterolateral region (white arrowhead). Membranes are stained with memb-mCherry. (**G,H**) Divisions (white dots) located in the lumen (**G**) or the apical scaffold (**H**, z-projection). 20 (**G**) and 18 hpf (**H**) embryos are shown. (**I**) Selected planes from a 3D time-lapse of a *neurog1*+ mitosis. White and blue dots track the daughter cells. Dashed lines indicate the approximated limit of the vesicle. Selected planes for each daughter cell are shown from 60 min onwards. At 129 min cells are delaminated. Asterisk indicates the SAG. The embryo is 18 hpf at the beginning of the time-lapse. (**J**) Reslice of a frame at 98 min from

*Figure 3 continued on next page*

*Figure 3 continued*

the video shown in (H) showing the z proximity between the tracked daughter cells during delamination (the red signal was removed for better visualisation). Scale bars, 20 μm. Dotted lines outline the limits of the otic vesicle.

The following figure supplement is available for figure 3:

**Figure supplement 1.** Cell division can precede *neurog1* expression.

favours higher expression levels in the pool of progenitors being specified. However, the effect of cell ablation was not recapitulated when proliferation was blocked by incubation with aphidicolin and hydroxyurea (AH) (*Hoijman et al., 2015*). This treatment decreased the number of *neurog1+* cells at 20 hpf (fold change AH/DMSO: 49,6 ± 6.3%, *Figure 4—figure supplement 2E*) but the mean levels of *neurog1* expression were not affected (fold change AH/DMSO: 110 ± 11%, *Figure 4—figure supplement 2E*). This result suggests that cell number and expression levels are not necessarily linked during otic *neurog1* expression and highlights the specific relevance of the ingressing cells in promoting the transcription of the *neurog1* gene.

Altogether, these results indicate that these cells act as pioneer neurogenic cells, contributing to the neurogenic domain both through their incorporation as *neurog1*$^+$ cells and by promoting *neurog1* expression non-autonomously in other cells of the domain.

## FGF controls otic epithelialization

To understand how the specification processes identified above are promoted, we decided to explore the role of FGF signalling, a pathway reported to control both *neurog1* expression in the vesicle and the number of neurons in the SAG (*Wang et al., 2015*; *Vemaraju et al., 2012*). To this aim, *neurog1:DsRedE* embryos were incubated with the FGFRs inhibitor SU5402 from 11 hpf until 19 hpf, beginning the treatment after placode induction and before otic morphogenesis starts (*Figure 5A and B*). Analysis of neuronal specification indicated that SU5402 treatment reduced the global level of DsRedE expression (*Figure 5C*), in agreement with the previous ISH analysis of *neurog1* expression (*Vemaraju et al., 2012*; *Léger et al., 2002*). This reduction was caused not only by a decreased mean level of *neurog1* expression in each cell (*Figure 5B and C*), but also by a reduction in the number of *neurog1*$^+$ cells (*Figure 5C*, and particularly in the *neurog1*$^{+Hi}$ cells). To confirm that the FGF pathway is mediating the mentioned phenotype, we crossed a transgenic line expressing a dominant negative isoform of the FGF receptor 1 fused to GFP under the control of a heat-shock (hs) promoter (*hsp70:dnfgfr1-EGFP*) (*Norton et al., 2005*) with the *TgBAC(neurog1:DsRedE) nl6* line. Inducing transgene expression at 10 hpf phenocopied at 20 hpf the effect on otic *neurog1* expression observed in SU5402 treated embryos (*Figure 5D and E*).

We realised that the phenotypes produced by blocking FGF signalling are similar to those resulting from cell ablation. Furthermore, given that FGF blockade strongly reduces the number of SAG neurons when it is performed early during otic development (*Wang et al., 2015*), we hypothesise that FGF signalling might control the early cell ingression event. We tested this idea by blocking the FGF signalling from 11 hpf onwards (both using SU5402 or the *hsp70: dnfgfr1-EGFP* transgene), photoconverting NLS-Eos in cells located anterior to the otic epithelium at 13 hpf (*Figure 5F and H*, left panels) and, subsequently, quantifying the number of photoconverted nuclei inside the otic vesicle at 18 hpf (*Figure 5F and H* (right panels), G and I). As shown in *Figure 5G and I*, SU5402 treatment

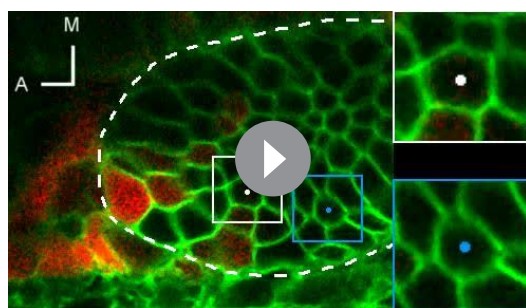

**Video 9.** Real-time activation of *neurog1* expression in local specified cells. Coronal ventral planes from z-stacks selected to follow the beginning of DsRedE expression in two individual cells that are being specified locally (white and blue dots). Insets show higher magnification images.

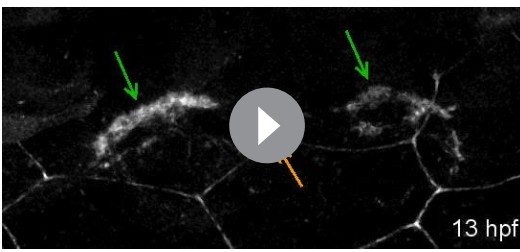

**Video 10.** Apical scaffold formation dynamics. 3D reconstructed time-lapse of Pard3-GFP (gray) localization during otic morphogenesis (dorsal view). Pard3-GFP in the otic vesicle (green arrows) or in the superficial external superficial (orange arrows) is shown. The anterolateral apical scaffold forms early during placode development and is transitory.

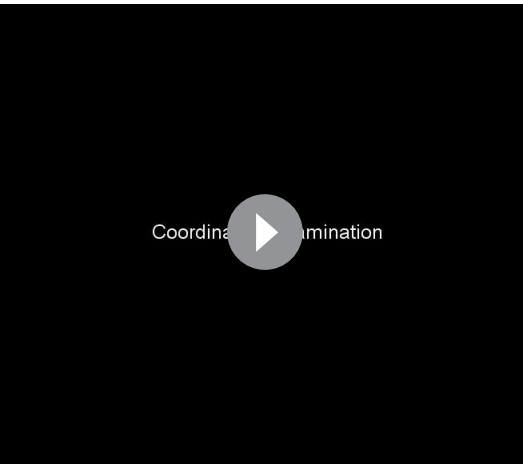

**Video 11.** Coordinated and quick delamination after division of *neurog1* expressing cells. Coordinated delamination: in the upper panel, coronal planes tracking an individual cell before division (white dot) and their daughters after division and until delamination (white and blue dots) are shown. In the lower panel, 2D movement of the tracked cells is shown. Note the coordinated behaviour of daughter cells moving in close contact to the periphery of the tissue and delaminating simultaneously. Quick delamination after division: tracking of other cell including sagittal planes in the lower panel. Only one daughter is tracked (white dot). White lines indicate the limits of the vesicle.

or DNFGFR1-EGFP induction significantly reduce the number of ingressed cells (DMSO: 4.7 ± 1.1 cells, SU5402: 1.0 ± 0.4 cells; heat-shocked siblings 5,3 ± 0.4 cells; heat-shocked *hsp70:dnfgfr1-EGFP/+*: 0.2 ± 0.2 cells). These results suggest that the FGF pathway contributes to neuronal specification in the otic vesicle by promoting the ingression of the pioneer cells into the neurogenic domain.

To gain insights into how the FGF pathway influences cell ingression, we performed time-lapse imaging during otic placode morphogenesis in embryos expressing DNFGFR1-EGFP. Tracking of photoconverted cells in these embryos showed that they still move towards the otic epithelium but remain outside (*Figure 5—figure supplement 1E*). Interestingly, in these embryos the anterior region of the epithelium folds at an earlier stage in development than in control embryos (*Figure 5J*; *Video 13*), becoming synchronous with folding of the posterior region (and not asynchronously as in the wild type embryos, *Figure 2—figure supplement 1D*; *Video 8*). Additionally, the otic basal lamina also formed earlier in DNFGFR1-EGFP expressing embryos than in siblings (*Figure 5K*). Conversely, overexpression of FGF3 by heat-shocking a hsp70:fgf3 line did not affect the anterior events (folding and cell ingression, *Figure 5—figure supplement 1F and G*) suggesting that endogenous anterior FGF levels are sufficient to mediate these processes. However, this manipulation led to a delay in folding of the posterior part of the epithelium, (a region where endogenous FGFs are not acting), supporting the notion that FGFs regulate otic epithelialisation. Altogether, these results suggest that endogenous FGF activity delays the final steps of anterior otic placode morphogenesis, providing time for cell ingression before the epithelial barriers appear.

Although important in other contexts, the control of proliferation does not seem to play a central role in the FGF signalling effect on otic specification, as blocking FGF did not modify the number of otic cells positive for phospho-Histone 3 (pH3+ cells, *Figure 5—figure supplement 1, A and B*). Moreover, not only does the FGF pathway control the number of *neurog1+* cells but also the mean levels of *neurog1* expression (as we show above with the AH experiments, both parameters were not coupled).

## Discussion

We have identified a new group of cells that act as pioneers of the otic neurogenic domain. These cells have two essential roles: they constitute the first specified cells of the domain and they promote

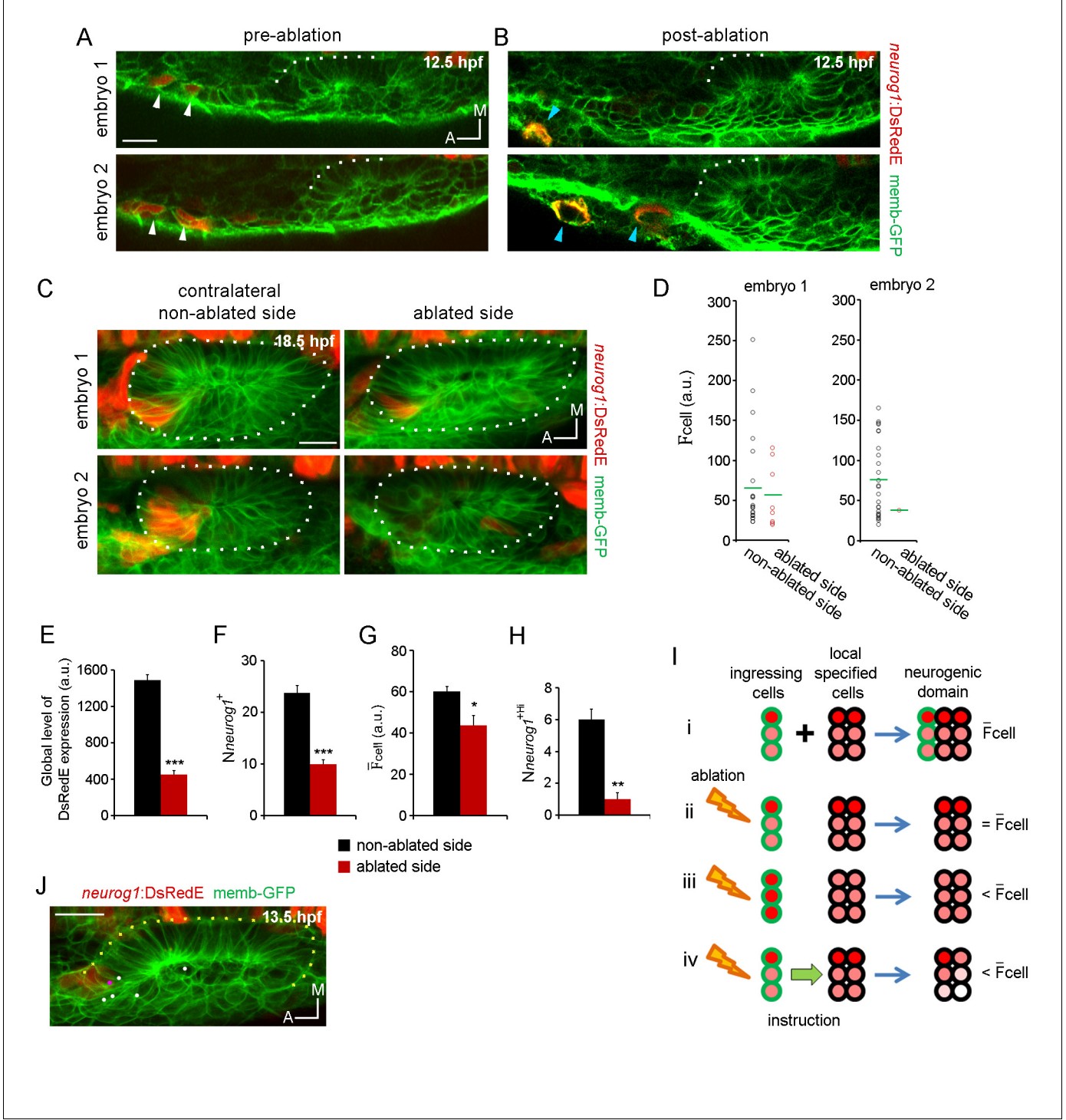

**Figure 4.** Ingressing cells instruct local neuronal specification. (**A,B**) Laser ablation of *neurog1*+ cells before ingression. Two different embryos are shown. Images of the otic epithelium and its anterior region at 12.5 hpf just before (**A**) and after (**B**) laser-ablation. White arrowheads indicate *neurog1*+ cells. Blue arrowheads localise the ablated region. Embryo 1 only received one laser pulse and embryo 2 three laser pulses (only two are visible in this plane). The contrast of the red signal was increased to improve visualisation. (**C–H**) *neurog1* expression pattern inside the vesicle after ablation. (**C**) Average z-projections of embryos shown in (**A,B**) 5 hr after ablation (18.5 hpf). The ablated side and their contralateral non-ablated side of the same embryo are shown. (**D**) Quantification of $F_{cell}$ in each *neurog1*+ cell of the vesicles shown in (**C**). Each dot indicates one cell. Green lines indicate the mean of each condition. The number of *neurog1*+ cells in each vesicle is: embryo 1, non-ablated side: 24, ablated side: 8; embryo 2, non-ablated side: 25, ablated side: 2. (**E–H**) Parameters of neuronal specification at the single cell level are shown: global level of DsRed expression (**E**) N*neurog1*+ (**F**), $\bar{F}_{cell}$ (**G**), and N*neurog1*+Hi (**H**). Data are mean ± s.e.m. (n = 6). t-test ***p<0.0001, **p<0.0005, *p<0.05. (**I**) Scheme with of different

*Figure 4 continued on next page*

*Figure 4 continued*

explanations of how early ablation of ingressing cells influences $\bar{F}_{cell}$ inside the vesicle at later stages. (i) In absence of cell ablation the neurogenic domain is composed by ingressing and local specified cells, with a characteristic value for $\bar{F}_{cell}$. (ii) If the distribution of cells with high and low fluorescence levels is equal between the ingressing and the local specified cells, ablation of ingressing cells does not change the $\bar{F}_{cell}$. Thus, this possibility does not explain the observed decrease in $\bar{F}_{cell}$ after ablation. (iii) If the *neurog1*$^{+Hi}$ cells are mainly ingressing cells, ablation of these cells reduces the $\bar{F}_{cell}$. However, *Figure 4J* shows that *neurog1*$^{+Hi}$ cells are mainly resident cells of the epithelium. (iv) If an instruction from the ingressing cells to the local specified cells is present, ablation of the ingressing cells decreases the $\bar{F}_{cell}$. The intensity of red depicts the DsRedE level of expression in each cell. (J) Dots show the location at 13.5 hpf of backtracked cells corresponding to *neurog1*$^{+Hi}$ cells at 19 hpf in a non-ablated embryo. Pink dot: *neurog1*$^{+}$ ingressed cell. White dots: *neurog1*$^{-}$ cells. The 3D reconstruction of the placode shown is representative of two different analysed embryos. All embryos are TgBAC(*neurog1:DsRedE*)$^{n16}$ and membranes are stained with memb-GFP. Scale bars, 20 µm. Dotted lines outline the limits of the otic vesicle.

The following figure supplements are available for figure 4:

**Figure supplement 1.** Calibration and specificity of ablation experiments.

**Figure supplement 2.** Late neurogenic phenotypes after ablation and specification analysis of non-proliferative otic placodes.

specification of resident cells of the vesicle, thus spreading commitment to a neural fate (*Figure 5L*). To our knowledge, this is the first example of neuronal progenitors instructing specification of other progenitors. In the mammalian developing brain, differentiated neurons of the cortical plate migrate to invade the dorsal telencephalon and are able to control the timing of progenitor neurogenesis (*Teissier et al., 2012*). Our analysis challenges the view that otic neuronal specification takes place in a static tissue. Indeed, the results presented here show that elaborate cell behaviours underlie development of the neurogenic domain, including intra-organ cell movements, delamination, cell divisions and importantly, cell ingression (*Figure 5L*).

Ingression of progenitors to the otic epithelium could also be relevant for sequential stages of their own differentiation, in a similar way that migration is important for maturation of either immature neurons in the mouse cortex (*Ayala et al., 2007*), or progenitors of the *Drosophila* optic lobe (*Apitz and Salecker, 2015*). Thus, the sequential epithelialisation and de-epithelialisation could be a general and crucial step for differentiation, as it has been recently proposed (*Zheng et al., 2014*). Our data indicate that the SAG integrates neuronal cells from at least two different origins: the ingressing cells and the ones specified locally. Different neuronal populations have been already identified in the SAG, including vestibular and auditory neurons (*Torres and Giráldez, 1998*; *Bell et al., 2008*). It still needs to be addressed whether the different populations of progenitors contributing to the neurogenic domain will differentiate into different functionally subgroups of neurons inside the ganglion.

In chick, a transitory population of cells surrounding the invaginating otic placode was described and termed 'otic crest cells' (*Hemond and Morest, 1991*). This population of cells seem to migrate to the rostral part of the SAG. These cells could be similar to the second pool of neurog1+ cells described here migrating directly to the SAG, suggesting similarities between chick and zebrafish. Given that single cells were not followed over time, a putative ingression of 'otic crest' into the otic placode might have been missed. Moreover, ingression of cells from outside to the otic epithelium might be an evolutionarily conserved event, since it was also reported to occur during mouse otic development (Freyer et al., 2011). Some of these ingressing cells have been shown to ultimately reside in the SAG. Whether these cells also have a function in neuronal specification of other cells remains to be explored.

## Pioneer cells and positional information

The otic neurogenic domain emerges in a defined ventroanterolateral position due to the dialogue of several signalling pathways that regionalise the otic placode (*Maier et al., 2014*; *Fekete and Wu, 2002*; *Abello and Alsina, 2007*; *Raft and Groves, 20142015*). In light of this, within the otic placode the fate of each cell would be dictated by its position in the tissue (*Bok et al., 2007, 2005*; *Brigande et al., 2000*; *Whitfield and Hammond, 2007*) upon the influence of the extrinsic signals. However, we observe that some ingressing cells are specified prior to their incorporation to

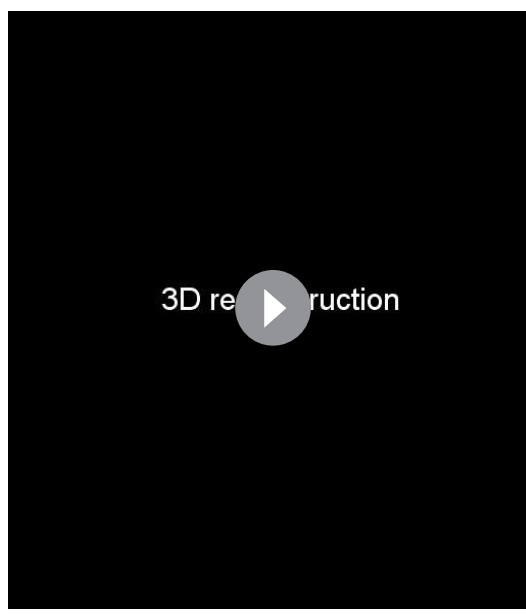

**Video 12.** Ablation of pioneer cells before ingression affects *neurog1* expression in the neurogenic domain at later stages. 3D reconstruction: DsRedE signal in the neurogenic domain (red) of otic vesicles at 21 hpf corresponding to the previously ablated and contralateral non-ablated sides of the same embryo. A single plane of the memb-GFP signal from each vesicle is shown for better 3D orientation (green). The DsRedE fluorescence coming from cells outside the otic vesicle was removed with FIJI to improve the visualisation of the phenotype inside the vesicle. z-stack: sequence of coronal planes from dorsal to ventral of *neurog1* expression in the otic vesicle at 21 hpf in ablated and contralateral non-ablated sides of the embryo. The DsRedE expression levels can be visualised in single cells (quantifications of specification phenotypes were performed on this type of z-stacks).

the anterolateral domain of the otic epithelium. Moreover, when ingressing cells are laser ablated, the cells in the otic vesicle located in the position of the ingressed cells (i.e. receiving the same putative diffusing morphogens) do not seem to adopt a neurogenic fate. This suggests that secreted factors establish a region competent for neurogenic specification, to which the ingressing cells (and probably other mechanisms) provide instructive signals to induce *neurog1* expression. In agreement with this possibility, Tbx1, the main transcription factor involved in otic neurogenic regionalisation, is a repressor of *neurog1* expression. Tbx1 is excluded from the anterior part of the vesicle, making the region competent to be induced by neurogenic signals (***Bok et al., 2011***; ***Radosevic et al., 2011***; ***Raft et al., 2004***). Thus, in addition to the reported role of cell movements on the spatial delimitation of different domains of the neural tube (***Xiong et al., 2013***; ***Kicheva et al., 2014***), we propose that coordination between cell movement and cell communication contributes to the neuronal pattern of the otic vesicle.

## Signals for ingression and instruction

In embryos mutant for FGF3, FGF8 and FGF10, and embryos in which FGF signalling has been temporally blocked, distinct phases of otic neural development are impaired (***Wright and Mansour, 2003***; ***Zelarayan et al., 2007***; ***Pirvola et al., 2000***; ***Léger et al., 2002***; ***Vemaraju et al., 2012***; ***Alsina et al., 2004***; ***Alvarez et al., 2003***). Our work indicates that FGF signalling promotes ingression of pioneer cells into the neurogenic domain, suggesting that some of the previously reported effects on *neurog1* expression could be due to this novel role.

Additionally, FGF signalling is known to control cell behaviour in other organs, such as epithelialisation and cell migration during kidney tubulogenesis and lateral line development (***Atsuta and Takahashi, 2015***; ***Aman and Piotrowski, 2008***). Particularly in the inner ear, FGF signalling controls epithelial invagination during otic morphogenesis in the chick (***Sai and Ladher, 2008***). We have identified a role of this pathway in zebrafish otic morphogenesis, delaying tissue folding during epithelialisation, and thus influencing neurogenesis. Additionally, it is possible that the FGF pathway also impinges on cell migration. The candidate ligands for the FGF effects on morphogenesis might be FGF8 and FGF3 coming from the hindbrain (Maves et al., 2002) and FGF3 from the endoderm and mesoderm (***McCarroll and Nechiporuk, 2013***). FGF10a is also expressed at these stages in the region where the pioneer cells are migrating (***McCarroll and Nechiporuk, 2013***). However, *neurog1* expression is normal in otic vesicles of FGF10a mutant embryos (***Figure 5—figure supplement 1C and D***), indicating that this ligand is most probably not involved in these processes.

A question that emerges from our analysis is how ingressing cells regulate *neurog1* expression in their neurogenic domain neighbours. The Notch pathway could participate in this process. However, since Notch activation reduces the number of specified neuronal cells via lateral inhibition (***Haddon et al., 1998***; ***Abelló et al., 2007***) and ingression enhances it, the instructive signal should inhibit Notch activity in the resident cells of the vesicle. Given that inhibition of cell ingression

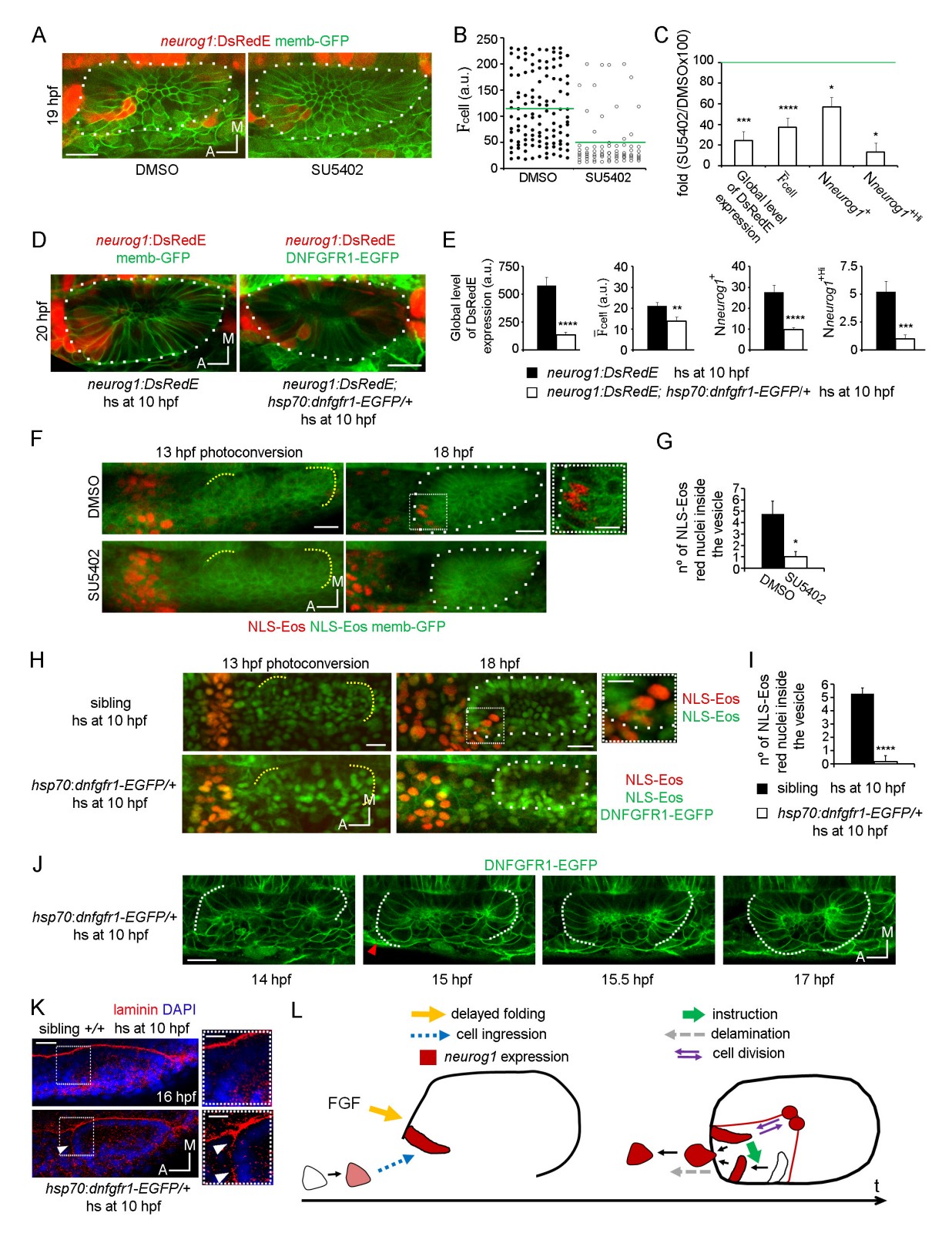

**Figure 5.** FGF control of neuronal specification. (**A–C**) *neurog1* expression pattern inside the vesicle in embryos incubated in DMSO or SU5402. (**A**) Images of otic vesicles at 19 hpf incubated from 11 hpf in DMSO or SU5402 (ventral planes). (**B**) Quantification of $F_{cell}$ for cells of vesicles from the groups shown in (**A**). Each dot indicates one cell. Green lines indicate the mean of each condition. n = 5 for DMSO and n = 6 for SU5402. (**C**) Parameters of neuronal specification at the single cell level for the data shown in (**B**): global level of DsRed expression, $\bar{F}_{cell}$, N*neurog1*[+] and

*Figure 5 continued on next page*

*Figure 5 continued*

N*neurog1*[+Hi] are shown as fold change of SU5402/DMSOx100. (D,E) *neurog1* expression pattern inside the vesicle from *neurog1:DsRedE;hsp70:dnfgfr1-EGFP/+* or *neurog1:DsRedE* embryos heat-shocked at 10 hpf. (D) Z-projections of otic vesicles at 20 hpf. (E) Parameters of neuronal specification are shown: global level of DsRed expression, $\bar{F}_{cell}$, N*neurog1*[+] and N*neurog1*[+Hi] (n = 8). (F) Photoconversion at 13 hpf of NLS-Eos stained nuclei in a region anterior to the otic epithelium. Embryos expressed memb-GFP and were treated with DMSO or SU5402 from 11 hpf (z-projections). At 18 hpf, photoconverted nuclei is observed inside the vesicle of the DMSO treated embryo. High magnification in the right (dotted square, Scale bar 10 μm). Yellow dotted lines indicate the limits of the otic epithelium. (G) Quantification of the number of photoconverted nuclei inside the vesicle (n = 6 for DMSO and n = 7 for SU5402). (H,I) Photoconversion experiments as in (F,G) but on *hsp70:dnfgfr1-EGFP/+* and sibling embryos heat-shocked at 10 hpf. (H) Z-projections of the photoconversion and cell ingression. (I) Quantification of the number of photoconverted nuclei inside the vesicle (n = 7 for siblings and n = 6 for *hsp70:dnfgfr1-EGFP/+*). (J) Selected images from a time-lapse of *hsp70:dnfgfr1-EGFP/+* embryos heat-shocked at 10 hpf. Note that as early as 14 hpf the anterior part of the otic tissue is already folding, at 15 hpf the process is advanced (red arrowhead), and at 15.5 hpf the anterior and posterior regions seem to be symmetrically folded (see also *Video 13*). (K) Laminin immunostainings at 16 hpf in *hsp70:dnfgfr1-EGFP/+* and sibling embryos heat-shocked at 10 hpf. The nuclei were counterstained with DAPI. High magnification in the right (dotted square, Scale bar 10 μm). The images are representative of 6 embryos analysed. Note the formation of a continuous layer of laminin in some regions (white arrowheads). (L) Scheme of cell dynamics playing a role in neuronal patterning of the inner ear. FGF signalling delays anterior tissue folding allowing the ingression of pioneer *neurog1*[+] cells in the prospective neurogenic domain of the otic epithelium. These pioneer cells promote *neurog1* expression in other cells of the neurogenic domain. In addition, *neurog1*[+] cells divide symmetrically and delaminate. Data are mean ± s.e.m. t-test ****p<0.001, ***p<0.005, **p<0.01, *p<0.05. Scale bars, 20 μm. White dotted lines outline the limits of the otic vesicle.

The following figure supplement is available for figure 5:

**Figure supplement 1.** Analysis of cell division controlled by SU5402 and neurog1 expression in FGF10a mutant embryos.

reduced not only the number of *neurog1+* cells but also the mean expression levels, the mechanism for instruction seems to rely on the activation of the *neurog1* promoter more than in stimulation of proliferation. This hypothesis is supported by the fact that: (a) FGF pathway blockade reduced both the number of *neurog1+* cells and the mean *neurog1* expression levels without affecting proliferation, and (b) AH inhibition of proliferation did not affect the mean levels of *neurog1* expression.

## Divisions in the neurogenic domain are symmetric and apical

Our 4D analysis allowed us to address the mode of division in the otic neurogenic domain for first time. We found that in all cases including both *neurog1*[−] and *neurog1*[+] cells, both daughter cells acquire a neuronal fate. During the time frame analysed, no divisions were found where one daughter cell remained as a *neurog1*- progenitor while the other activated the proneural expression, as has been described in the neural tube (*Wilcock et al., 2007*; *Das and Storey, 2012*; *Taverna et al., 2014*). We cannot exclude, however, that asymmetric divisions occur at later times or at very low frequency.

Studies of fixed chick otic vesicles described the presence of mitosis in the basal side of the epithelium in addition to the luminal ones (*Alvarez et al., 1989*). Such mitoses were termed 'basal divisions' similar to the ones taking place in the retina in which mitotic cells are no longer polarized apically and in contact with the ventricular membrane (*Weber et al., 2014*). In our study, we also observed non-luminal mitoses, but our data show that these divisions remain in contact with a Pard3 scaffold and therefore still keep their apical polarity.

## Spatiotemporal dynamics of proneural expression

Neural specification usually occurs in epithelialised tissues. However, we observed activation of *neurog1* expression in pioneer cells before epithelialisation, suggesting that stable cell-cell contacts would be dispensable to initiate proneural expression. Similarly, in mouse *neurog2* is

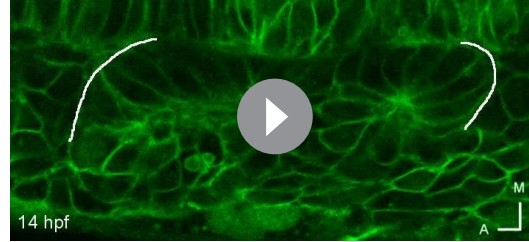

**Video 13.** Synchronous folding of the anterior and posterior regions of the otic placode in *dnfgfr1-EGPF* expressing embryos. Time-lapse during placode morphogenesis in *Tg(dnfgfr1-EGFP)* embryos heat-shocked at 10 hpf. Lines indicate the epithelial folding.

expressed in migrating sensory neuron precursors (*Marmigère and Ernfors, 2007*), although its expression begins before exiting the epithelium and migration (*Zirlinger et al., 2002*). We were able to visualise the transit of an otic neuronal progenitor from *neurog1* expression to delamination. Analysis of *neurog1* expression levels suggests that delamination occurs once a given threshold of proneural expression is reached; probably associated to *neurod1* induction.

The otic placode and other cranial placodes originate from a large common pre-placodal region (PPR) adjacent to the neural plate (*Bailey and Streit, 20052006*). Precursors from the PPR segregate and coalesce into individual cranial placodes, which progressively acquire specific identities (*Breau and Schneider-Maunoury, 2014*; *Streit, 2002*; *Bhat and Riley, 2011*; *Saint-Jeannet and Moody, 2014*; *McCarroll et al., 2012*). Our data revealed that otic *neurog1* is expressed before of what it was conceived and outside the epithelium by a group of cells that ingress during morphogenesis. This suggests that neural specification might precede the acquisition of a defined placodal identity. Thus, we propose that some PPR precursors might already be committed to a neural fate and that their subsequent allocation into the placodes (by random or directed movements) provides them one or another placodal identity. Further work in this direction might shed light into this hypothesis.

In conclusion, our study reveals that cell movements underlie an instruction essential for otic neuronal specification, a crucial step in neurogenesis. Unravelling the complex mechanisms that determine the number of neurons incorporated in a forming ganglion may provide insights leading to a better understanding of the anomalies associated with auditory neuropathies.

## Materials and methods

### Zebrafish strains and maintenance

The following zebrafish lines were used in this study: AB wild-type, *TgBAC(neurog1:DsRedE)nl6* (*Drerup and Nechiporuk, 2013*), *Tg(neurod:GFP)* (*Obholzer et al., 2008*), *Tg(actb1:Lifeact-GFP)* (*Behrndt et al., 2014*) *Tg(Xla.Eef1a1:H2B-Venus)* (*Recher et al., 2013*), *Tg(hsp70:dnfgfr1-EGFP)pd1* (*Lee et al., 2005*), *Tg(elA:GFP)* (*Labalette et al., 2011*), *neurog1*[hi1059] (*Golling et al., 2002*), *Tg (hsp70:fgf3)* (*Hammond and Whitfield, 2011*), and a cross between the *TgBAC(neurog1:DsRedE)nl6* and the mutant fgf10a+/− (*Norton et al., 2005*). They were maintained and bred according to standard procedures (*Westerfield, 1993*) at the aquatic facility of the Parc de Recerca Biomèdica de Barcelona (PRBB). All experiments conform to the guidelines from the European Community Directive and the Spanish legislation for the experimental use of animals.

### Live imaging and image processing

Live embryos were embedded in low melting point agarose at 1% in embryo medium including tricaine (150 mg l$^{-1}$) for dorsal confocal imaging using a 20x (0.8 NA) glycerol-immersion lens. Imaging was done using a SP5 Leica confocal microscope in a chamber heated at 28.5°C. 20 to 80 µm thick z-stacks spanning a portion or the entire otic vesicle (a z-plane imaged every 0.5–2 µm) were taken every 1 to 3 min for 2–12 hr. Raw data were processed, analysed and quantified with FIJI software (*Schindelin et al., 2012*). For visualisation purposes, the images were despeckled. For quantifications of *neurog1* expression, images were not modified. Videos were assembled selecting a plane from every z-stack at every time point to better visualise the phenotype (or track a cell) or shown as 3D reconstructions. A representative video from at least three different embryos is shown. Images in figures are either shown as confocal coronal sections, 3D reconstructions or average z-projections. To track the trajectory of individual cells, 3D videos were analysed using the MtrackJ, Manual tracking plugins of ImageJ (*Meijering et al., 2012*), and temporal colour code applied to generate a single image of the tracks.

### Morphometric and proliferation analysis

To perform quantifications in different regions of the otic vesicle, we live imaged a z-stack and built a rectangular cuboid defined by external vertices of the otic vesicle. The cuboid was divided in eight equally sized regions, and quantifications were performed inside each region. Before quantification, the z-stacks were aligned in 3D to correct for variability in orientations during mounting to guarantee the coronal sectioning of the vesicle. For volume calculation, the x-y area of the tissue in each plane

of the z-stack was measured and then multiplied by the z spacing every plane (the volume of the lumen was subtracted). The number of cells in each region was determined manually by counting H2B-mCherry stained nuclei on z-stacks, using the Cell counter plugging of ImageJ. 3D visualisation of Lyn-GFP plasma membrane staining helped the identification of each single cell. To quantify the number of cell divisions in the otic epithelium in a period of time, high temporal resolution videos (1 min frequency) in 3D of H2B-GFP stained nuclei were analysed manually to detect every chromosome segregation event. The number of divisions in each region of the vesicle was determined building a cuboid as described above for each time point.

## Two photon laser ablation

To ablate a group of cells, a two-photon laser beam (890 nm) from a Leica SP5 microscope was applied over one side of the embryos mounted in agarose (the contralateral side was maintained intact as a control). We used embryos with mosaic H2B-mCherry nuclear staining (mRNA injected at 16 cell stage) to calibrate the settings of the microscope required to ablate 2–3 cells in each ablation pulse (*Figure 4—figure supplement 1A*; *Video 14*). Each pulse consisted in approximately 5 s of 30% laser power applied in a ROI of about 70 μm$^2$ imaged with a 20x air objective and a digital zoom of 64x. In *neurog1-DsRedE* embryos, the cells to ablate were identified by single photon confocal imaging recognizing the DsRedE fluorescence in cells anterior (or posterior) to the otic placode/vesicle. Right after ablation, imaging of the vesicle was performed to confirm the damage caused (dead cells were clearly visualised). Sequential pulses at different locations were applied to ablate an increased number of cells. No damage outside the ablated region was observed. Ablated embryos were maintained mounted at 28°C until the moment in which specification analysis was performed (see below).

## Photoconversion experiments

To detect ingression of cells into the epithelium, photoconversion of NLS-Eos expressing nuclei was performed with UV light (λ = 405 nm, using a 20x objective in a Leica SP5 system) on 13 hpf mounted embryos. A 3D ROI of about $1 \times 10^5$ μm$^3$ located 25 μm apart from the anterior limit of the epithelialising placode was photoconverted. Photoconversion was checked by confocal imaging right after UV illumination. The number of photoconverted cells was quantified using the Cell counter plugin from FIJI (DMSO = 58 ± 9 cells; SU5402 = 59 ± 7 cells, n = 8). The embryos were then removed from the agarose and incubated in embryo medium until 20 hpf to check for cell ingression by 3D imaging. When blockade of FGFR was performed, the embryos were dechorionated at 11 hpf, incubated with SU5402 or DMSO in embryo medium until 13 hpf, mounted in agarose including SU5402 or DMSO, photoconverted, imaged, unmounted, and incubated in presence of the drugs in solution until 19 hpf. In some cases, the TgBAC(*neurog1:DsRedE*)$^{nl6}$, the *neurog1*$^{hi1059}$ (embryos genotyped by PCR after imaging), *Tg(hsp70:fgf3)*, or *Tg(hsp70:dnfgfr1-EGFP)* lines were used. In the latter case, time-lapses at 5 min resolution time were performed to track photoconverted nuclei over time.

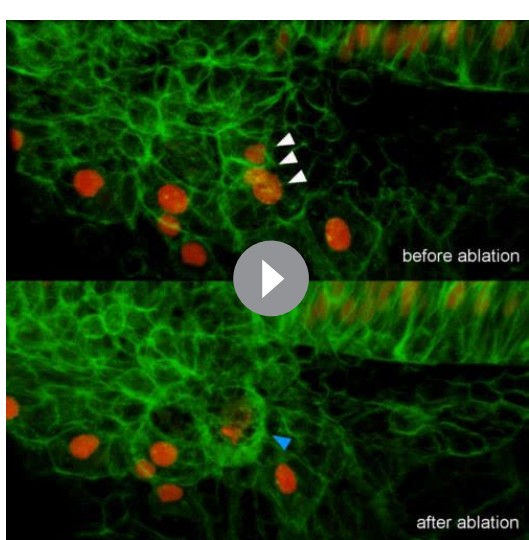

**Video 14.** Calibration of cell ablation. 3D reconstruction of z-stacks acquired before and after ablation of embryos expressing H2B-mCherry in some cells adjacent to the neural tube. The neighbouring cells remain undamaged after ablation of the targeted cells (white arrowhead). The damage is indicated by the blue arrowhead. The embryos also express globally memb-GFP.

## Specification analysis

To analyse specification phenotypes z-stacks were acquired with fixed settings (laser power and detector gain) between different

experimental groups (or vesicles in the case of ablations). The settings were adjusted to detect a range of increased or decreased fluorescence levels without saturation or lack of signal. DsRedE fluorescence was quantified in single slices using imageJ. A small region of a few pixels was created and a mean fluorescence level in each cell ($F_{cell}$) was calculated by averaging three quantifications in different x, y and z positions of the cytosol (the background was deducted from each measurement). To consider a cell positive for DsRedE expression, a threshold was defined empirically for each set of experiments, as the minimum level at which DsRedE expression in different z slices is unambigously detected (to avoid mistakes produced by fluorescence coming from cells located at other z positions). We then calculated the mean $F_{cell}$ in each vesicle ($\bar{F}_{cell}$), the number of $neurog1^+$ positive cells, and the global level of DsRed expression as the sum of the $F_{cell}$ for all the $neurog1^+$ cells in a vesicle. $neurog1^{+Hi}$ cells were defined as the ones that have fluorescent level higher than 1.5x $\bar{F}_{cell}$ of the control (DMSO or non-ablated side) vesicles. Dynamic quantifications were performed by sequentially measuring fluorescence at consecutive times of a video in the same cell. The mean rate of increase in fluorescence was calculated as $\frac{\Delta F}{\Delta t}$. The same single cell fluorescence quantifications were performed in the neuroepithelial cells of the hindbrain, in a region adjacent to the otic vesicle.

## Microinjection, drug treatment and heat shock experiments

To label cellular and subcellular structures, mRNA encoding for the following fusion proteins were injected at 1 cell stage after being synthesised with the SP6 mMessenger mMachine kit (Ambion): H2B-mCherry, H2B-GFP or NLS-Eos (100–150 pg) (*Sapede et al., 2012*), Pard3-GFP (50–75 pg) (*Buckley et al., 2013*), Lyn-EGFP (memb-GFP 100–150 pg), membrane-mCherry (100–150 pg). For the specification analysis, TgBAC(neurog1:DsRedE)nl6 dechorionated embryos were treated with SU5402 25 μm (Merk Millipore 572630), aphidicolin 300 μM (Merck) in combination with hydroxyurea 100 mM (Sigma), or DMSO (Sigma) added to the embryo medium. For determination of the number of pH3+ cells, DMSO or SU5402 treated embryos from 13 to 16 hpf were fixed and processed for the immunostainings.

The heat shock was performed by incubating 10 hpf embryos in preheated water at 39° during 30 min. Fluorescence from DNFGFR1-EGFP was detectable from about one hour after initiation of the shock. Induced embryos were selected at 12 hpf. For photoconversion or laminin immunostaining, EGFP- embryos were used as controls. For DsRedE expression analysis in which a membrane staining is relevant, *neurog1:DsRedE* embryos injected with memb-GFP at 1 cell-stage were heat shocked and used as controls.

For experiments using the *fgf10a+/-; neurog1:DsRedE* line, the embryos were mounted and imaged at 20 hpf for DsRedE expression analysis, recovered from the agarose, and incubated until 5 dpf, when the *fgf10a-/-* mutants embryos were identified by the absence of pectoral fins.

## Immunostaining

For immunostaining, dechorionated zebrafish embryos were fixed in 4% PFA overnight at 4°C and immunostaining was performed either on whole-mount or cryostat sections. Embryos for sections were cryoprotected in 15% sucrose and embedded in 7.5% gelatine/15% sucrose. Blocks were frozen in 2-Methylbutane (Sigma) for tissue preservation and cryosectioned at 14 μm on a Leica CM 1950 cryostat. After washing in 0.1% PBT, and blocking in 0.1% PBT, 2% Bovine Serum Albumin (BSA), and 10% normal goat serum (NGS) for 1 hr at RT, embryos were incubated overnight at 4°C in blocking solution with the appropriate primary antibodies: rabbit anti-Laminin (Sigma, 1:200), rabbit anti-pH3 (Abcam, 1:200). After extensive washing in 0.1% PBT, donkey anti-rabbit Alexa-488 (Thermo fisher scientific A21206; 1:400) was incubated overnight at 4°C in blocking solution. Sections were counterstained with 1 μg/ml DAPI, mounted in Mowiol (Sigma-Aldrich) and imaged in a Leica SP5 confocal microscope.

## In situ hybridisation

Synthesis of antisense RNA and whole-mount in situ hybridisation were performed as previously described (*Thisse et al., 2004*) to generate a probe against *neurog1* (*Itoh and Chitnis, 2001*). Dechorionated Tg(elA:GFP) (which express GFP in rhombomeres 3 and 5) zebrafish embryos were fixed in 4% paraformaldehyde (PFA) overnight at 4°C and dehydrated in methanol series, rehydrated again and permeabilized with 10 mg/ml proteinase K (Sigma) at RT for 5–10 min depending on their

stage. Digoxigenin-labeled probe was hybridised overnight at 70°C, detected using anti-digoxigenin-AP antibody at 1:2000 dilution (Roche) and developed with NBT/BCIP (Roche). After the ISH, an immunostaining for the GFP expressed from the transgene was performed (primary antibody: rabbit anti-GFP (Torrey Pinnes; 1:400), secondary antibody: anti-rabbit Alexa-488 (Thermo fisher scientific A21206; 1:400)). Embryos were post-fixed overnight in 4% PFA and used for imaging mounted in 100% glycerol.

### Statistics

All statistical comparisons are indicated in figure legends including one sample and unpaired t-test performed using GraphPad. The box plot was generated in excel.

## Acknowledgements

We thank A Nechiporuk for the *neurog1:DsRedE* fish line, S Schneider-Maunoury and M Breau for the *hsp70:dnfgfr1-EGFP* line, D Gilmour for the *fgf10a* mutant line, C Pujades for the *eA:GFP* line, E Marti, I Gutierrez Vallejo and E Gonzalez Gobart for helping with the fishes, P Bovolenta for critically reading the manuscript, R Aguillon and J Batut for their help and the fishes, and the members of the Advanced Light Microscopy Unit of the UPF/CRG.

## Additional information

### Funding

| Funder | Grant reference number | Author |
| --- | --- | --- |
| Ministerio de Economía y Competitividad | BFU2011-270006 | Berta Alsina |
| Ministerio de Economía y Competitividad | BFU2014-53203 | Berta Alsina |

The funders had no role in study design, data collection and interpretation, or the decision to submit the work for publication.

### Author contributions

EH, Conceptualization, Data curation, Supervision, Investigation, Visualization, Methodology, Writing—original draft, Writing—review and editing; LF, Data curation, Investigation; PB, Resources, Writing—original draft, Writing—review and editing; BA, Conceptualization, Supervision, Funding acquisition, Writing—original draft, Writing—review and editing

### Author ORCIDs

Esteban Hoijman, http://orcid.org/0000-0003-1927-1989
Berta Alsina, http://orcid.org/0000-0002-6997-1249

### Ethics

Animal experimentation: All experiments conform to the guidelines from the European Community Directive and the Spanish legislation for the experimental use of animals. The protocols (AR081096P4 and AR081098P3) were approved by the Committee on the Ethics of Animal Experiments of the Parc de Recerca Biomèdica de Barcelona (PRBB), Spain.

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
