## [Decision Letter]

[Editors’ note: a previous version of this study was rejected after peer review, but the authors submitted for reconsideration. The first decision letter after peer review is shown below.]

Thank you for submitting your work entitled "Pioneer *neurog1* expressing cells ingress in the otic primordium and instruct neuronal specification" for consideration by *eLife*. Your article has been reviewed by three peer reviewers, and the evaluation has been overseen by a Reviewing Editor and a Senior Editor.

Our decision has been reached after consultation between the reviewers. Based on these discussions and the individual reviews below, we regret to inform you that your work will not be considered further for publication in *eLife*.

Altogether, there was a great deal of enthusiasm about the paper but equally extensive discussion about the need for additional quantitation throughout. In particular, the reviewers request better characterization of the cells that ingress (and do not ingress), better characterization of the ablation experiments, and better characterization of the role of FGF. Given that *eLife* only allows 2 months for revision, the reviewers and editor did not feel you could realistically make the required changes in that window of time. Therefore we are returning the manuscript to you. However, if you should be able to address their comments in the future, we would welcome seeing a revised version that addresses these major concerns and would make every effort to return the paper to the original reviewers.

Reviewer #1:

In this interesting and beautifully illustrated paper, Alsina and colleagues identify a population of cells, which they believe are derived from the lateral line placodes, that express ngn1 and invade the otocyst by insertion into the otic epithelium. They present evidence that these pioneer ngn1+ cells are at least partly necessary to induce further ngn1-expressing cells from the otic epithelium. If these pioneer cells are ablated, the authors suggest this leads to a reduction in the number of induced neurogenic cells in the otocyst.

The images are beautiful and well presented and I think the message of the paper is potentially interesting and challenging to our current ideas of otic neurogenesis. My main concern with the paper is that I feel the evidence for induction of ngn1 cells by the ingressing pioneer cells is less convincing than the rest of the paper. This is due in part to a lack of absolute numbers presented in the paper, which I expand on below.

1) In Figure 1, the authors provide a detailed temporal and spatial reconstruction of ngn1-expressing cells in the developing otocyst. It would be extremely helpful to the reader to give absolute numbers of cells in this part of the paper in addition to percentages. At 19hpf, how many cells are in the otocyst, and how many of them express the ngn1 reporter? This will be important for the rest of the paper (see below).

2) In the Results section the authors write: "The modest increase in the number of cells in this region cannot account for the large enrichment in cell proliferation, suggesting a higher proliferation rate in the NgD region." This sentence does not make sense to me. Can the authors rephrase it?

3) The Eos technique needs to be described better in the main text to make clear that a NLS-Eos construct was injected into the embryos at the 1 cell stage.

4) The authors suggest that the ingressing cells they observe are derived from the lateral line based on ngn1 expression. Are there other more specific lateral line markers or lineage tracers they can use to confirm this? Some posterior lateral line mutants such as cxcr4 and 7 and Kremen1 have migration defects. Are these expressed in cells invading the otocyst, and do mutants in these genes affect the development of ngn1+ cells in the otocyst? Such mutants might support the ablation studies presented in the paper.

5) I would like a better description or estimation of how many lateral line cells the authors believe ingress and contribute to the NgD. Is it possible to use the NLS-Eos labeling to estimate their contribution in the mature ganglion? Is it possible to evaluate whether they divide more than their otocyst counterparts, either in the otocyst itself or as transit amplifying cells once they have delaminated again?

6) The description of the lateral line cell ablation could be more detailed. The authors should indicate exactly how many lateral line cells were ablated per embryo and how many laser pulses were given per cell and per embryo. In addition to the contralateral control, they should also ablate cells outside the otocyst that are not lateral line cells and see if this affects neurogenesis in the NgD. It is possible that simply killing cells in the vicinity of the otocyst affects NgD development (for example, perhaps by altering FGF release or signaling).

7) The reason why points 1 and 6 are so important is that it is not clear whether the deficit in cells seen in the NgD domain after ablation is due to a loss of inductive signals from the ingressing cells (which is what the authors suggest) or is simply the result of depleting cells that would otherwise contribute to the NgD. At the moment, it is not clear to the reader which is more likely. As mentioned above, a better description and discussion of the quantitative data would be in order here – how many lateral line cells do the authors think ingress into an otocyst (1? 10? 50?) versus how many NgD ngn1+ cells are generated within the otocyst. The authors offer their reverse tracing of their movies as evidence to suggest that most cells expressing high levels of the ngn1 reporter come from inside the otocyst and thus the contribution of the ingressing cells is small, but it is not completely clear if this experiment was performed in normal or ablated embryos.

8) The authors suggest that FGF may be responsible for helping the lateral line-associated cells ingress into the otocyst. In support of this, they show a reduction in ngn1-reporter cells in the otocyst after SU5402 treatment, and a reduction in NLS-Eos labeled lateral line cells in the otocyst. I have two questions – have the authors made time lapse films of the SU5402 treated embryos directly showing a failure of ngn1-reporter cells to ingress, and in their NLS-Eos experiments, is the TOTAL number of photoconverted cells (inside the otocyst + outside the otocyst) the same, but it is just their distribution in the two compartments that differs?

9) In their Discussion, the authors mention the Notch signaling pathway in relation to zebrafish otic neurogenesis. Do ngn1+ neurons express Δ ligands, and if so, wouldn't these inhibit rather than promote neurogenesis if they ingressed into a field of Notch-expressing progenitors?

Reviewer #2:

Hoijman and colleagues have provided a detailed description of neurogenesis in the zebrafish otic primordium. They provide a number of interesting findings. Using reporter constructs expressing dsred under regulation of neurog1 genomic elements, they identify cells that express dsred that subsequently ingress into the otic vesicle, while others begin expression within the vesicle before dividing. Additional cells express dsred after division, with both daughters expressing the reporter. Ablating ingressing cells reduces the overall number of dsred+ cells suggesting that the ingressing cells have instructive roles in promoting neurog1+ expression in cells within the otic vesicle. Together the work describes a potentially novel mechanism for neurogenesis within the otic placode that may have more general implications.

1) It is not clear that only dsred+ cells ingress into the developing otic vesicle. It looks like in some videos that dsred- cells also ingress. Moreover it is not clear that only anterior cells ingress – in Video 6 there appear to be cells in the ventral posterior that also do so. These potentially alternate observations might suggest that there is a substantial addition to the otic vesicle after its initial formation but would change the interpretation of dsred+ cells are undergoing a characteristic pioneer behavior. It appears that the authors have the ability to track all cells contributing to the otic vesicle to resolve this issue. It's important to do so, for other results such as the effects of FGF would be interpreted in a different light.

2) It is not clear how specific the ablations are for the dsred+ pioneer cells. Are other dsred- neighbors damaged or spared? If dsred- cells are also ingressing (see comment 1) then the specificity of ablation may be an issue.

3) Is the reduction of dsred+ precursors due to the specific ablation of ingressing cells or is it due to the reduction of the overall number of dsred+ cells? That is, could there be a 'community effect' where dsred+ cells induce others irrespective of their initial origin outside the otic vesicle?

4) Does ablation of dsred+ ingressing cells alter proliferation of cells within the otic vesicle?

Reviewer #3:

In this manuscript the authors study otic neurogenesis. They describe a population of pioneer cells in zebrafish, which arises outside of the otic placode, invades the placode and generates neurons. Their experiments suggest that these cells play a role in promoting neurogenesis in neighbouring otic epithelial cells, since their ablation reduces neurog1 expression (both levels and cell numbers). They also perform morphometric and proliferation analysis of the otic vesicle, and finally some experiments to suggest that FGF signalling somehow controls the integration of the pioneer cells into the vesicle.

The idea of pioneer cells is an interesting and novel finding that warrants further investigation and provides a novel view on how neuroblasts are determined in the ear. However, throughout the manuscript the authors make quite forceful conclusions that are not always supported by their data (e.g. for most experiments numbers are extremely low, or not given). The morphometric analysis does not add much to the paper, and I wonder if it should be removed. The FGF results are very preliminary and in their current form are not conclusive. The authors do not provide a clear model for FGF function and thus do not provide any mechanistic insight.

Below, some specific comments that the authors should address.

1) Throughout the manuscript the figures need to be improved. To help the reader figures need better labelling: label SAG in Figure 1 (and other figures), label delaminating neurons to distinguish them from neuroblasts in the vesicle. Outline the otic placode/vesicle in figures and movies and add arrows for orientation (anterior-posterior, medial-lateral) into the movies. In some videos it is difficult to see which neurog1-dsRedE cells are in the vesicle, remain outside or delaminate. It needs to be clear which are the SAG cells and the placode cells.

2) Morphometric analysis in the Results section: these measurements do not add very much to the paper and do not really allow any particular conclusions and distract from the main part of the paper. Maybe this section should be removed.

If this section remains in the manuscript, the authors need to provide numbers: how many placodes were analysed? How representative are the results? This sentence is unclear: "The modest increase in the number of cells in this region cannot account for the large enrichment in cell proliferation, suggesting a higher proliferation rate in the NgD region." The authors conclude that the neurogenic domain has a higher proliferation rate, but have not assessed proliferation rate and therefore this conclusion is not valid.

3) The authors say that neurog1 cells become integrated into the vesicle during placode formation; can they specify the timing and explain this better for readers unfamiliar with the fish and with the otic development. This is also picked up in the Discussion, however the timing is not very clearly described in the paper.

4) In the Results section; Figure 2: The authors describe that neurog1 + cells from outside the otic vesicle become integrated into the otic vesicle, and generate neurons of the SAG. In Figure 2, Video 6 they show that some cells exhibit this behaviour, while others do not. It is not clear how this behaviour is determined; are those cells that do invade the vesicle always neurog1 positive? The authors need to compare the behaviour of neurog1 + cells; it is quite possible that only neurog1 + cells exhibit this behaviour, and that this is the reason that position of cells is irrelevant.

5) What are the cells in the "second pool of neurog1+ cells"? do they contribute to the SAG, other cranial ganglia?

6) What do the authors mean? "Additionally, other morphological features particular of these stages could contribute to cell ingression"

7) In the Results section: the authors suggest that levels of neurog1 determines when cells delaminate from the vesicle. Can they provide numbers: how many cells were measured? What is the threshold? Please provide statistics to support this claim.

The final conclusion from this section is: "This suggests that cells delaminate relative to neurog1 levels and not to the time elapsed from the beginning of neurog1 expression". The authors do not measure the time a cell spends in the vesicle; how do they come to this conclusion?

8) In the Results section the authors re-visit cell division within the neurogenic domain, which is already described in the first section of the paper. It is not clear what this section adds and why Pard3 staining is relevant in this context, and what this adds to the main message of the paper. The whole section seems a bit out of place. The conclusion is that "our analysis of the origin of neurog1+ cells revealed that they are added to the NgD by three different mechanisms: cell ingression, local expression and cell division. They are not really different mechanisms, neuroblasts are known to proliferate and expand before becoming terminally determined and this observation is therefore not surprising.

9) The authors show that ablation of neurog1+ cells (pioneer cells) leads to a reduction of neurog1 expression in the otic vesicle. They suggest that the number of ablated cells determines the how many neurog1+ cells later appear in the vesicle as well as the level of neurog1 expression in each cell. The authors only show 2 embryos; this is not sufficient to reach this conclusion. Are there more specimen; please provide numbers.

The authors state that otic neurogenesis does not recover after ablation, but do not provide much evidence (2 embryos in Figure 4—figure supplement 2). I would like to see more examples to reach such a firm conclusion. What happens to the SAG? Is there other compensation? Embryo 2 in Figure 4—figure supplement 2 appears to form some neurog1 cells eventually; have the authors looked later?

Do the authors suggest that the majority of otic neuroblasts is induced by the pioneer cells? If so, this is interesting and novel, but this requires more data with solid statistics and longer observations.

The authors measure the level of fluorescence in individual cells in pioneer ablated and non-ablated embryos: how many embryos were analysed, how many cells?

10) In zebrafish the otic vesicle does not delaminate as an epithelium but rather coalesces from cells that then form a lumen. As such it's quite possible that addition of cells to the vesicle continues after initial lumen formation. Therefore the issue in my mind was whether there was specificity for dsred+ cells invading. If nonspecific then the FGF requirement is just a continuation of the requirement for vesicle formation. If nonspecific then the specificity of the ablation needs to be determined – ablation of bystander dsred- cells will also contribute to any observed phenotype.

11) The FGF results are not convincing; more experiments are needed to provide a clear view of what exactly FGF is supposed to do in this context. The authors should provide a clear model of how they suggest that FGF acts: what is the source of FGF? Do pioneer cells have FGF receptors? Do they suggest that FGF is an attractant? Why do not all neurog1 cells invade the placode if that is the hypothesis? How can the effect on pioneer neuron invasion and proliferation be unravelled? Does FGF have multiple effects, not only affecting pioneer invasion, but also local induction of neuroblasts and their expansion?

Simply blocking signaling by SU5402 does not provide definitive proof that FGF is involved since other receptors are also affected by this drug. The authors need to use different ways to show that FGF is involved; these could be dominant negative receptors in pioneer cells, cell type specific knock-down of FGF response or secretion. Without further experiments this part of the manuscript opens up too many unanswered questions.

12) Discussion section. The authors need to be more careful with their choice of words (there are many examples throughout the text); they suggest that the pioneer neurons are the first "specified cells" and "promote neural commitment" – we do not know if these cells are truly specified; this can only be assessed by culturing them in isolation. What do the authors mean by promoting commitment? That pioneer neurons induce neighbouring cells to become neuroblasts?

13) The Discussion section is a bit convoluted and the arguments are not clearly structured, and often the authors seem to over-interpret their data. For example, the authors argue strongly that positional information does not determine whether or not a cell adopts pioneer identity, but it may equally be possible that by the time they evaluate this cells are not competent to do so.

14) It is surprising that the authors do not refer to an older paper by Hemon & Morest 1991 describing the 'otic crest', as well as a more recent paper describing neural crest contribution to the otic vesicle in mouse. This may be very relevant to the current study.

In this context, the authors should discuss whether their model could also be true in higher vertebrates like chick and mouse, where development of the placode is a much longer process than in amphibians and fish. For example, in fish and frog neurogenic markers are expressed much earlier than in mouse and chick; could their observation be species specific?

15) Terminology:

'Ingression' is not the right word to describe that individual cells from outside the otic vesicle are integrated into it. They invade, are incorporated, inserted or similar.

'Local specification': in developmental biology 'specification' is used to describe along which path cells or tissues can differentiate when cultured in isolation. Therefore in this context the term 'local specification' is somewhat misleading. The authors should find a better term.

16) A native speaker should read the manuscript; it contains errors, and peculiar constructions and expressions making the text at times a bit cumbersome to read. E.g. "Pioneer cells specify outside the otic primordium and ingress during otic placode formation". Should read: 'are specified'.

17) The authors should avoid too many abbreviations; they make the text more difficult to read in particular for readers outside the field. There is no reason to abbreviate terms like 'neurogenic domain', 'global level of dsRedE expression' (GLE) or similar.

[Editors’ note: what now follows is the decision letter after the authors submitted for further consideration.]

Thank you for submitting your article "Pioneer *neurog1* expressing cells ingress in the otic primordium and instruct neuronal specification" for consideration by *eLife*. Your article has been reviewed by three peer reviewers, and the evaluation has been overseen by Marianne Bronner as the Reviewing and Senior Editor.

The reviewers have discussed the reviews with one another and the Reviewing Editor has drafted this decision to help you prepare a revised submission.

Summary:

This manuscript provides live imaging data to demonstrate that neuronal specification in the inner ear of zebrafish is dependent on a number of neurogenin 1 (ngn1) expressing pioneer cells that enter the otic primordium transiently, which cause induction of otic epithelial cells to upregulate ngn1 and these cells then delaminate to form the statoacoustic ganglion (SAG). Furthermore, the authors demonstrate that this ingression process is dependent on FGF signaling using dominant negative FGFR transgenic fish and an Fgfr inhibitor, SU5402.

Essential revisions:

1) If their neurog- neighbors are also joining the placode, then ingression is not a distinct behavior of 'pioneer' cells. While this observation does not detract from authors' main conclusions the specific roles of the 'pioneer' cells in promoting the formation of later neurons, it does speak to whether the 'pioneer' cells are actually outside the otic primordium (defined as the group of cells that will form the otic placode). The authors make a convincing argument that global tracking would be time-consuming and not directly applicable to answering the central questions of the work presented. I therefore suggest that they drop claims that these 'pioneer' cells are outside the otic primordium as stated in the Abstract and elsewhere.

2) The study begs the question of whether ngn1 is required for the pioneer cells, the ingression process, the otic epithelial cells to respond to the pioneer cells, or all of the above. It may be difficult to tease out all requirements of ngn1 but whether cell ingression proceed normally in the ngn1 knockout mutants should be investigated.

3) The authors concluded that FGF normally delays epithelial barrier formation in the anterior otic primordium. As a result, less pioneer cells get into the primordium to induce subsequent ngn1+ cells. The authors demonstrated a reduction of ngn1+ cells within the otic epithelium when Fgfr function was knockdown. If the authors are correct, one should expect a pileup of pioneer cells outside the placode. A video and quantification of the pioneer cells demonstrating this fact is lacking and warranted. The premature basement membrane formation of the otic epithelium using anti-laminin staining is not convincing. While the dotted lines outline the border of the otic primordium, it also negated judgement of the data by readers. What happens with gain of FGF function models? Would there be an increase in cell ingression compare to wildtype? If so, would one conclude the rate-limiting step is the barrier formation of the anterior otic primordium. That is if FGF only functions in the otic epithelial cells and not in the pioneer cells.

4) In the Results section the argument claiming that small differences in the cell number but large differences in the mitotic event found in the neurogenic domain suggests an increase in proliferative activity is unclear. A high mitotic event should result in an increase in cell number at a later time point, at the least, or the moderate increase in cell number is due to neuroblast delamination.

5) The authors cited a reference by Raible's lab in regards to ngn1 gene expression in the pioneer cells. The ngn1 expression in Figure 2 of Raible's paper is later than the onset of pioneer cells ingressing the otic epithelium described here. It may be a good idea to determine whether ngn1 transcripts are detected in the pioneer cells. It will complement the functional study of ngn1 suggested above.

6) The illustration in Figure 4 is not helpful to describe all the scenarios that the authors are considering. At least, the authors should describe/cite Figure 4 Iii in the text. The text in the legend also needs clarification.

7) The summary diagram in Figure 5 is misleading and does not serve the manuscript. The summary diagram implies that FGF signaling is required for cell ingression, but according to the authors, FGF signaling mediates cell ingression indirectly by delaying epithelial barrier formation. At the minimum, the summary diagram should be flipped so that the medial region is towards the top to be consistent with the videos and images shown in the figures. The authors should modify the diagram; as it is it somehow suggests that it affects the incorporation of cells into the vesicle.

[Editors' note: further revisions were requested prior to acceptance, as described below.]

Thank you for resubmitting your work entitled "Pioneer *neurog1* expressing cells ingress in the otic primordium and instruct neuronal specification" for further consideration at *eLife*. Your revised article has been favorably evaluated by Marianne Bronner (Senior and Reviewing editor) and three reviewers.

The manuscript has been improved but there are some remaining issues that need to be addressed before acceptance, as outlined below:

In general, the reviewers remain largely enthusiastic about your work. However, they all felt that you argued many of their points rather than making relatively easy changes to the manuscript. I'm afraid I am unable to accept your paper without better attending to the criticisms of the reviewers. I would like to highlight that it is highly unusual for *eLife* to allow more than one round of review. I am willing to do so in this case because I think you can make the needed changes with alterations to the text and figures. I stress that it is essential that you make every attempt to address the reviewers comments rather than arguing them. Otherwise, I will have no choice but to decline the paper. I have included the detailed comments of the individual reviewers so that you can better understand their requests and I ask you also to better address the concerns raised in the original review.

Reviewer #1:

While it seems we are in a silly semantic argument I think the point I made in the original review still stands.

I would argue that the otic primordium/placode is the field of all cells that subsequently form the otic vesicle. I interpret the authors results to show that not all otic primoprdium/placode cells join the otic vesicle at once, as some cells ingress into the otic vesicle epithelium after it forms. I originally pointed out that it looked like cells other than the ones designated as neuronal pioneer cells were also ingressing. As the zebrafish otic vesicle forms by coalescence rather than the epithelial infolding and delamination described for amniote embryos, this type of MET would perhaps be expected.

If a whole bunch of cells are ingressing into the otic vesicle then the behavior is not specific to the neuronal pioneer cells. Indeed I would argue that if neuronal pioneer cells and their neighbors are all joining the otic vesicle, then these cells and their neighbors are part of the otic primordium/placode even if they join the vesicle after it has formed an epithelium. However if they are interspersed with other cells that do not join the otic vesicle, that would be evidence for a distinct behavior more along the lines of what the authors are implying. The authors were unwilling to perform the timelapse analysis and in toto lineage tracing to address this point. I agreed that in toto imaging is not necessary for the main conclusions of the current work. But I do not believe that the authors should state that ingression is a special behavior unique to pioneer cells, which is what is implied as the text currently stands.

Reviewer #2:

The authors have addressed most points raised in previous review.

They did not change the summary diagram in Figure 5; if I remember correctly all three reviewers raised the point that the summary does not accurately reflect the findings of the authors indepdently. THis suggests that the message the authors wish to convey is not clear. In its current form the diagram suggests that FGF affects the behavior of ingressing cells directly, but the authors show that it affectes the formation of the epithelial barries – this is how arrows are interpreted even if this is not the intention of the authors. It should not be very difficult to change this to make the figure accurately reflect their findings.

Reviewer #3:

Specific comments:

Ngn1 not required for the ingression process: The lack of cell ingression in the ngn1 null mutants is interesting and is relevant information. I don't see why it should not be incorporated into the manuscript with some additional quantification.

Knockdown of FGF signaling reduced ingression of ngn1-positive cells: Although there is no strong evidence for the predicted ngn1 positive cells piling up outside the FGF lof placode, the authors proposed several alternate explanations for this observation, which are acceptable. The authors provided additional FGF gain-of-function results, which further supported a role of FGF in regulating barrier formation in the otic placode. These results, together with the lof experiments, argue against a direct role of FGF in the ingression process.

Anti-laminin staining: A higher magnification of the control in a comparable region as the treated sample would help to convince readers. The anti-laminin staining in the mutants, though seems stronger than controls, is blotchy and discontinuous and raises the question whether there is indeed a basement membrane barrier. However, the additional gain-of-FGF function experiments supported a normal role of FGF in delaying barrier formation.

I appreciate the authors' attempt to explain their point of view. In Results paragraph two, it states "The large difference in the change of the two parameters (meaning low cell number increase but high percentage of mitotic events) suggests that the increased number of mitotic events is not a consequence of having more cell dividing at the same rate, but due to a specific increase in the proliferative activity of these cells". I interpreted this sentence to mean that there were not more cells going into cell division but the same cells are dividing faster, which contributed to the higher percentages of mitotic counts. If so, I am confused because I thought the authors measured mitotic events rather than measured how fast each cell go through the cell cycle. Proliferate activity and proliferate rate are not interchangeable terms in my mind. Increased proliferate activity (mitotic events) could be caused by either increased proliferate rate or increased number of cells in division. Either scenario should result in an increased in cell number. The moderate increase in cell number observed at the time of measurement could be simply due to timing. The authors measured mitotic events between 14 to 18.5 hpf. On average, each cell cycle takes about 8-12 hrs. Considering the cells have just reached the placode at 13-14 hpf, peak increase in cell number may not be apparent until after 18.5 hpf. By this time, some neuroblasts will start to delaminate from the epithelium, which will also reduce the total cell number in the region.

I do not find the option 1 and 2 described in the rebuttal letter particularly helpful, partly due of the interchangeable terms of proliferative rate and proliferative activity.

Summary diagram: I might have over-interpreted those videos. Based on the examples shown in the videos, it appears that ingression of cells is occurring at the lateral edge of the placode and delamination is slightly medial to where ingression takes place. This pattern is consistent with the position of the statoacoustic ganglion being located medial to the otocyst. If this is correct, the summary diagram should reflect the spatial relationship between ingression and delamination.

I still think the arrow of the FGF in the summary diagram is misleading, but it is up to the authors to decide the best way to summarize their data.

In summary, this work describes for the first time the cell ingression phenomenon in the otic placode of zebrafish. It also demonstrated that after ingression, the ngn1-positive cells instruct more neighboring cells to turn on ngn-1 within the placode. The mechanisms underlying these cellular events, however, are not known. With the additional data provided, neither ngn-1 nor FGF appears to be directly involved in the cell ingression process. In fact, the data from FGF become a distraction. This ingression process could very well be a general mechanism of how cells populate the otic placode and may not be specific for the neuronal specification, as suggested by reviewer 1. Despite the nice live imaging results, there are not much mechanism to grapple with.

---

## [Author Response]

[Editors’ note: the author responses to the first round of peer review follow.]

*Reviewer #1:*

*In this interesting and beautifully illustrated paper, Alsina and colleagues identify a population of cells, which they believe are derived from the lateral line placodes, that express ngn1 and invade the otocyst by insertion into the otic epithelium. They present evidence that these pioneer ngn1+ cells are at least partly necessary to induce further ngn1-expressing cells from the otic epithelium. If these pioneer cells are ablated, the authors suggest this leads to a reduction in the number of induced neurogenic cells in the otocyst.*

*The images are beautiful and well presented and I think the message of the paper is potentially interesting and challenging to our current ideas of otic neurogenesis. My main concern with the paper is that I feel the evidence for induction of ngn1 cells by the ingressing pioneer cells is less convincing than the rest of the paper. This is due in part to a lack of absolute numbers presented in the paper, which I expand on below.*

*1) In Figure 1, the authors provide a detailed temporal and spatial reconstruction of ngn1-expressing cells in the developing otocyst. It would be extremely helpful to the reader to give absolute numbers of cells in this part of the paper in addition to percentages. At 19hpf, how many cells are in the otocyst, and how many of them express the ngn1 reporter? This will be important for the rest of the paper (see below)*

We have now added the absolute values for cell number of the otic vesicle and the neurogenic region in the main text. At this stage, the placode has around 311 ± 16 cells and the neurogenic region 49 ± 3 cells (counted from 11 embryos). A detailed quantification with absolute numbers of the dsred+ cells in the neurogenic domain was already present in the text and Figure 4. These numbers are kept in the text related to Figure 4 because is the part where ablation and the numbers are discussed.

*2) In the Results section the authors write: "The modest increase in the number of cells in this region cannot account for the large enrichment in cell proliferation, suggesting a higher proliferation rate in the NgD region." This sentence does not make sense to me. Can the authors rephrase it?*

We have now rephrased the sentence as: “While the increase in cell number in the neurogenic domain was moderate (about 3% more cells than other domains), the enrichment in mitotic events led to a 41% of the total number of divisions to occur in this domain. The large difference in the change of the two parameters suggests that the increased number of mitotic events is not a consequence of having more cells dividing at the same rate, but due to a specific increase in the proliferative activity of these cells.”

*3) The Eos technique needs to be described better in the main text to make clear that a NLS-Eos construct was injected into the embryos at the 1 cell stage.*

We have changed the text to make this clear. “To confirm this cell ingression, we injected NLS-Eos mRNA at 1 cell stage to obtain a homogeneous nuclear staining with this photoconvertable protein throughout the embryo. At 13 hpf, we photoconverted Eos (from green to red fluorescence) in a group of nuclei anterior to the primordium where the migrating cells are located.”

*4) The authors suggest that the ingressing cells they observe are derived from the lateral line based on ngn1 expression. Are there other more specific lateral line markers or lineage tracers they can use to confirm this? Some posterior lateral line mutants such as cxcr4 and 7 and Kremen1 have migration defects. Are these expressed in cells invading the otocyst, and do mutants in these genes affect the development of ngn1+ cells in the otocyst? Such mutants might support the ablation studies presented in the paper.*

It is possible that was not clear in the text, but we do not propose that these ingressing cells are lateral line cells. We just mentioned the fact that the DsRedE signal that we observed coincides spatiotemporally with a signal of neurog1 detected by ISH, that had previously been assigned to the lateral line cells. We believe that this assignment was based on their localization out of the otic placode and the proximity to the anterior lateral line placode, but it relies in a neurog1 ISH of a single time point without a cell tracking, a marker or another experiment that confirms the identity of the cells. Our high-resolution dynamic analysis shows that these cells are finally otic cells because they ingress in the otic placode and then delaminate to form the statoacoustic ganglion. We now have added a sentence to avoid this confusion (“Therefore, these cells develop into otic and not lateral line cells”).

Independently of this clarification, as the reviewer proposes, it is possible that also the pioneer cells migrate to the otic placode through the CXCR-CXCL pair of proteins, as happens during the migration of the posterior lateral line. Indeed, we have evaluated this possibility by firstly performing ISH for CXCL12a, CXCL12b, CXCR4a and CXCR4b (see the Figure 6).

Author response image 1.**DOI:**
http://dx.doi.org/10.7554/eLife.25543.027

If this pathway is involved in this migration, we expected expression of receptors to be detected in the migrating cells and the ligands to be expressed in the placode or nearby. The CXCL12a-CXCR4b pair does not fit this expectation, as while CXCR4b is expressed only in the posterior part of the placode (right panel in A; in this panel signal from a krox20 probe (asterisks) is also present, which was used to positionate the otic placode at these early stages), CXCL12a is present at low levels very anterior to the placode (left panel in B). On the other hand, CXCR4a is expressed in the region where the ingressing cells are migrating at 13 hpf (upper left panel in A). However, at 15 hpf the signal did not progress to be closer to the placode and instead was weaker (lower left panel in A), suggesting that this signal most probably do not belong to the ingressing cells (as we mentioned above, this region overlaps with cells of the lateral line and epibranchial placodes, making difficult to distinguish these populations by ISH at these stages). Moreover, CXCL12b is not expressed at or near the placode (in agreement with the data from ZFIN), suggesting that this pair of ligand-receptor is problably neither involved in the migration of the ingressing cells. To evaluate by another means if CXCR4a could be involved in the ingression of these cells, we analysed the expression of neurog1 by ISH in mutant embryos for CXCR4a -/- (if this receptor is required for ingression we should see a reduction of *neurog1* expression in the neurogenic domain similar to when the ingressing cells are ablated). We did not observe a change in *neurog1* expression (left and right panels in C). Altogether these results suggest that these ligands and receptors are not involved in the migration analyzed in this work. Due to these negative results, we have not included this Figure in the main text. If the reviewer still considers that these results should be included as supplementary information, we could do so.

*5) I would like a better description or estimation of how many lateral line cells the authors believe ingress and contribute to the NgD. Is it possible to use the NLS-Eos labeling to estimate their contribution in the mature ganglion? Is it possible to evaluate whether they divide more than their otocyst counterparts, either in the otocyst itself or as transit amplifying cells once they have delaminated again?*

We do not believe that using the NLS-Eos for this quantification is appropiate, because is not possible to photoconvert all the cells that will ingress given that the exact location of all cells that will ingress is difficult to determine before they start to express neurog1. Moreover, photoconversion of cells that are very close to the placode without also photoconverting resident cells is also very difficult to achieve. In our experiments we photoconverted a group of cells at a distance from placode, allowing us to determine about cell migration and ingression but not numbers. Additionally, as the region anterior to the otic vesicle is very dynamic with many cells moving in a variety of directions, photoconversion at a single time point will not be enough to calculate the total number. In spite of these limitations, based on our videos and photoconversion experiments, we can estimate that about 10 cells ingress, although this might be an underestimation. To precisely quantify the exact total number of *neurog1* cells that ingress would require a global and long tracking with different imaging settings and analysis of the one used in this work, and we believe that these quantifications are not essential for our conclusions.

The reviewer wonders about the total contribution of ingressing cells to the mature ganglion. This is an interesting question but this can also not be quantified using NLS-Eos, given that other cells located in the region of the ingressing cells, either o not ingress into the otic vesicle and are incorporated directly into the ganglion (“the second pool, neurod1 positive cells”) or they are anterior lateral line cells. Regarding divisions, we observed that every ingressing cell divides only once before or after ingression, similar to the local specified cells. We cannot discard that after delamination they dispaly a different division rate.

*6) The description of the lateral line cell ablation could be more detailed. The authors should indicate exactly how many lateral line cells were ablated per embryo and how many laser pulses were given per cell and per embryo. In addition to the contralateral control, they should also ablate cells outside the otocyst that are not lateral line cells and see if this affects neurogenesis in the NgD. It is possible that simply killing cells in the vicinity of the otocyst affects NgD development (for example, perhaps by altering FGF release or signaling).*

We have now added a better description about how we performed the ablation, including examples with the calibration using nuclear stainings. The added parts include a new figure (Figure 4—figure supplement 1), a new video (Video 14) and the following text:

In the main text: “Ablation of a limited number of cells (2-3 cells per laser pulse)”

In the Materials and methods: “We used embryos with mosaic H2B-mCherry nuclear staining (mRNA injected at 16-cell stage) to calibrate the settings of the microscope required to ablate 2-3 cells in each ablation pulse (Figure 4—figure supplement 1; Video 14). Each pulse consisted in approximately 5 seconds of 30% laser power applied in a ROI of about 70µm2 imaged with a 20x air objective and a digital zoom of 64x. In neurog1-DsRedE embryos”,

In the figure legends: “Calibration of cell ablations. A laser pulse (as described in Materials and methods) was applied to embryos expressing H2B-mCherry in some cells adjacent laterally to the neural tube. In example 1, two nuclei were stained in the imaged region before ablation (white arrowheads). After the laser pulse, a red ablation buble was observed as consequence of the death of the two stained cells (blue arrowhead). In example 2, the nuclei of neighbouring cells (numbered from 1 to 5) are surrounding two target cells (white arrowheads). Imaging after ablation indicated that the targeted cells died, but the neighbouring cells remained healthy and only slightly displaced in space. In example 3, a similar behavior than in example 2 can be observed, but the intact neighbouring cells are in close contact with the dead cells, highlighting the fact that ablation is highly specific and restricted to the targeted cells (white arrowheads, see also Video 14)”).

We have now indicated exactly how many pulses received each embryo (“compare embryos 1 and 2 from Figure 4, which received 1 and 3 laser pulses respectively”). Thanks to our high cellular resolution imaging before and after ablation, we believe that we hve achieved a higher precision in the control of the number of ablated cells by two photon-laser ablation than the usually achieved when this methodology was used for these purposes.

As suggested by the reviewer, we have performed new controls for ablation of DsRedE expressing cells, applying laser pulses at other locations (posterior to the placode, 13 hpf) or developmental stages (anterior to the placode, 19 hpf) than in our initial experiments (anterior to the placode, 13 hpf). In both controls, we did not observe a difference in specification a few hours after ablation, indicating that the effect is specific for the pioneer cells and not a general consequence of cell death close to the placode (Figure 4—figure supplement 1, subsection “Ingressing cells instruct neuronal specification”). Moreover, ablation experiments at late time points (when the neurogenic domain ihas already formed, Figure 4—figure supplement 1, subsection “Ingressing cells instruct neuronal specification”), also excludes that photobleaching of the resident cells is playing a role (something not possible to analyze when ablation is performed at 13 hpf because the low level of *neurog1* expression).

Regarding the FGF ligands expression, at these stages only *Fgf10a* is expressed in the ectodermal region where the migrating cells are located. We have performed new experiments crossing *Fgf10a* mutant embryos with the *neurog1:DsRedE* line, and analyzed the *neurog1* expression pattern in the otic vesicle. We did not observe a reduction in *neurog1* expression in the neurogenic region of placodes of the mutants compared with the siblings (this experiment is now included in the revised manuscript, (Figure 5—figure supplement 1 and subsection “Signals for ingression and instruction” in the text). This result suggests that this ligand is not a mediator of either: 1) an unspecific effect of ablation on its secretion that affects neurogenesis, or 2) the instruction performed by the ingressing cells. Only the hindbrain and the mesoderm express other FGF ligands at these stages, two tissues that were not affected by ablation.

*7) The reason why points 1 and 6 are so important is that it is not clear whether the deficit in cells seen in the NgD domain after ablation is due to a loss of inductive signals from the ingressing cells (which is what the authors suggest) or is simply the result of depleting cells that would otherwise contribute to the NgD. At the moment, it is not clear to the reader which is more likely. As mentioned above, a better description and discussion of the quantitative data would be in order here – how many lateral line cells do the authors think ingress into an otocyst (1? 10? 50?) versus how many NgD ngn1+ cells are generated within the otocyst. The authors offer their reverse tracing of their movies as evidence to suggest that most cells expressing high levels of the ngn1 reporter come from inside the otocyst and thus the contribution of the ingressing cells is small, but it is not completely clear if this experiment was performed in normal or ablated embryos.*

Here again we tried to explain the logic underlying our interpretation of the ablation results, which in our opinion undoubtedly led to think in an instructive role for the ingressing cells. In order to clarify this logic in the manuscript, we have now added some schematic representations in Figure 4.

As we mentioned in the Materials and methods, each pulse killed 2-3 cells. However, only one pulse is able to decrease the number of *neurog1+* cells inside the neurogenic domain by about 14 cells (23.8 against 10.0). These numbers were originally included in the text and, from our point of view, clearly suggest that the reduction in *neurog1+* cells in the placode cannot just be a consequence of the elimination of ingressing cells that will form the neurogenic domain. Moreover, the cells inside the vesicle remaining *neurog1* positive after ablation also reduce their *neurog1* expression levels (mean expression levels are reduced, Figure 4), reinforcing the idea of an influence of the ingressing cells on those already within the placode. This result per se indicates the presence of a cell communication as we observed an effect over resident cells (not ingressing), but we have killed cells outside of the vesicle. An alternative explanation for the decrease in mean expression levels after ablation is that *neurog1+hi* cells are mainly ingressing cells, and if we remove some ingressing cells by ablation, we could decrease the mean expression levels without the requirement of an instruction. However, the backtrack (performed in non-ablated embryos) demonstrates that the *neurog1+hi* cells are predominantly not ingressing cells, which excludes the possibility that ablation is preferentially killing *neurog1+hi* cells and consequently the mean expression levels decrease. Altogether, these results indicate that both mean expression levels and the number of *neurog1+* cells are regulated by an instructive signal coming from the ingressing cells. Thus, ablation is not only eliminating future *neurog1+* cells that are incorporated in the neurogenic domain, because this possibility cannot explain our results.

We mentioned above our estimation about how many cells ingress in the vesicle. As we stated above, in order to precisely quantify the exact total number of neurog1 cells coming from local specification or by ingression would require a global and long-term tracking with different imaging settings than used in this work. Although interesting to perform, we believe that these quantifications are not required for our conclusions.

*8) The authors suggest that FGF may be responsible for helping the lateral line-associated cells ingress into the otocyst. In support of this, they show a reduction in ngn1-reporter cells in the otocyst after SU5402 treatment, and a reduction in NLS-Eos labeled lateral line cells in the otocyst. I have two questions – have the authors made time lapse films of the SU5402 treated embryos directly showing a failure of ngn1-reporter cells to ingress, and in their NLS-Eos experiments, is the TOTAL number of photoconverted cells (inside the otocyst + outside the otocyst) the same, but it is just their distribution in the two compartments that differs?*

Time-lapse imaging in continuous presence of SU5402 is difficult to perform because the drug is light sensitive. We tried adding fresh drug periodically into the medium covering the agarose, but couldn´t achieve reproducible results.

Independently of the feasibility of the SU5402 experiment, as there are *neurog1+* cells migrating that ingress and others that do not, showing selected tracks of *neurog1+* cells that do not ingress would not be satistyingly informative (this should happen both in control and SU5402 treated embryos). Indeed, to assess quantitatively the effect on ingression would require the tracking of all migrating cells. Again, this would be extremely time consuming, and choe instead the NLS experiments that have provided us with quantification of the effects of blocking FGF signaling on ingression in a subset of cells (Figure 5). Moreover, the NLS-Eos technology was able to reproduce the results of ingression directly observed by timelapse in controls (Figure 2). To reinforce the results on the FGF effects, we have added new data using a heatshock-inducible dominant negative version of the FGF receptor 1 fused to GFP (Figure 5; Video 13). Using this tool, we have obtained similar results to those with SU5402 treatments, confirming the involvement of the FGF pathway both in cell ingression (using the NLS-Eos system) and neuronal specification (using the *neurog1-DsRedE* embryos). Additionally, timelapse of embryos expressing the truncated receptor allowed us to reveal a phenotype in the folding of the anterior placodal epithelium, taking place earlier when the FGF pathway is blocked (together with an early formation of the basal lamina). These results suggest that a morphogenetic alteration might impair cell ingression and underlies the effect of FGF blockade in cell ingression and neuronal specification.

Regarding the number and distribution of all photoconverted cells at the end of the experiment, it is difficult to quantify the photoconverted cells outside the otic vesicle because there are cells migrating to distant regions of the embryo, including deep layers and they are therefore missed. To guarantee the reproducibility of the photoconverted region at the initiation of the experiment, after embryo mounting we first measured the distance from the anterior limit of the otic placode, built a 3D ROI of the same dimensions, photoconverted this region and acquired a Z-Stack after photoconversion. Due to the variability in embryo orientation during mounting, we then quantified the number of photoconverted cells immediately after ablation using the cell counter plugin of FIJI, and obtained a similar number of photoconverted cells for DMSO (58 ± 9 cells, n=8), and SU5402 (59 ± 7 cells, n=8) treated embryos (we added these data in subsection “Photoconversion experiments”). Therefore, at the beginning of the experiment, the number of photoconverted cells was similar between the two groups, but at the end the number of cells inside the vesicle was highly different.

*9) In their Discussion, the authors mention the Notch signaling pathway in relation to zebrafish otic neurogenesis. Do ngn1+ neurons express Δ ligands, and if so, wouldn't these inhibit rather than promote neurogenesis if they ingressed into a field of Notch-expressing progenitors?*

Most probably Δ ligands are expressed in the *neurog1+* cells, as are observed in the otic neurogenic region (Haddon et al., 1998, as expected from the Notch mediated-lateral inhibition model). We agree with the reviewer that expression of these ligands in the ingressing cells would be expected to inhibit neurogenesis of the otic resident cells via the activation of Notch. However, what we were proposing was that perhaps in addition to expression of Notch ligands, the ingressing cells, by an unknown mechanism, reduce the levels of Notch signaling in the resident cells, thus promoting neuronal differentiation. (“The instructive signal should inhibit Notch activity in the resident cells of the vesicle”). Thus, in this putative possibility, the instructive signal would not be the Δ ligand itself expressed in the ingressing cells, but this ligand would participate in a downstream effect.

*Reviewer #2:*

*Hoijman and colleagues have provided a detailed description of neurogenesis in the zebrafish otic primordium. They provide a number of interesting findings. Using reporter constructs expressing dsred under regulation of neurog1 genomic elements, they identify cells that express dsred that subsequently ingress into the otic vesicle, while others begin expression within the vesicle before dividing. Additional cells express dsred after division, with both daughters expressing the reporter. Ablating ingressing cells reduces the overall number of dsred+ cells suggesting that the ingressing cells have instructive roles in promoting neurog1+ expression in cells within the otic vesicle. Together the work describes a potentially novel mechanism for neurogenesis within the otic placode that may have more general implications.*

*1) It is not clear that only dsred+ cells ingress into the developing otic vesicle. It looks like in some videos that dsred- cells also ingress. Moreover it is not clear that only anterior cells ingress – in Video 6 there appear to be cells in the ventral posterior that also do so. These potentially alternate observations might suggest that there is a substantial addition to the otic vesicle after its initial formation but would change the interpretation of dsred+ cells are undergoing a characteristic pioneer behavior. It appears that the authors have the ability to track all cells contributing to the otic vesicle to resolve this issue. It's important to do so, for other results such as the effects of FGF would be interpreted in a different light.*

It is possible that cell ingression occurs at very limited extent also in other regions of the vesicle. To confirm this, 3D tracking as done for the anterior regions should be done, to distinguish cell ingression of a migrating cell from normal epithelialization of local cells also occurring during mophogenesis. Indeed, cell invasion in the otic vesicle was described in mouse (as suggested the reviewer 3, we now added a paragraph about this in the Discussion). However, we concentrated our experiments, on the abundant, coordinated and spatiotemporally reproducible ingression that is observed in the anterolateral region, which concentrated where the neurogenic domain is forming. We concentrated in these cells because our aim was to understand the construction of the neurogenic domain. These cells resulted particularly interesting because they express *neurog1* at early stages, becoming the earliest otic neurogenic cells. Moreover, these cells are pioneer not only because they are the first cells of the neurogenic domain that express *neurog1*, but also because they promote *neurog1* expression in resident cells of the vesicle. Putative cell ingression events in other regions of the vesicle, of *neurog1-* cells and not incorporating into the neurogenic domain, although potentially interesting for other aspects of otic development, they would not be related to neurogenesis. Therefore they cannot be called neurogenic pioneer cells. We did not focus on these other cells, because our aim was not to analyze the general contribution of cell ingression to otic development, but to understand neurogenesis. In summary, the putative role of other cells on neurogenesis is difficult to analyze, and we think this does not interfere with our conclusions regarding the role of the group of anterior cells identified in this work. We want to stress that in order to precisely identify all the events of cell ingression would require a global and long tracking with different imaging settings and analysis of the one used in this work, which is out of the scope of this work.

Most cells that are incorporated anteriorly are *neurog1+,* although we detected exceptions of *neurog1-* cell also ingress in the vesicle but most of them express *neurog1* briefly after entering. It is possible that these cells also act as pioneers, but it is more difficult to analyze their role by ablation (by the lack of a marker that favors visualization at these early points) and thus we restricted our conclusion to the *neurog1+* cells outside the vesicle. The ablation of ingressing cells *neurog1* negative would only contribute to analyze if a pioneer behavior requires neurog1 be expressed before entering but do not affect our conclusions. We concentrated in the *neurog1+* cells: 1) to highlight that *neurog1* expression in otic cells begins in cells during migration and outside the otic vesicle, and 2) as this could guide ablation and photoconversion experiments. As mentioned above, we called pioneer to these cells not because they ingress into the placode, but because they participate in the building of the neurogenic domain.

We do not fully understand how ingression in other regions of the vesicle would affect our results about ingression in the neurogenic domain. We now added new results expressing a dominant negative version of the FGF receptor 1 (Figure 5, Video 13), which indicate that the FGF signaling delays the folding of the anterior otic epithelium and thus might facilitate ingression of the pioneer cells. Since the posterior morphogentic folding is not different in these dominant-negative FGF blockade embryos compared with controls, it seems that FGF blockade is specifically affecting the anterior events. Altogether, this reinforces the specific role of the anterior ingression cells in neurogenesis.

*2) It is not clear how specific the ablations are for the dsred+ pioneer cells. Are other dsred- neighbors damaged or spared? If dsred- cells are also ingressing (see comment 1) then the specificity of ablation may be an issue.*

As mentioned originally in Materials and methods, each pulse killed 2-3 cells. It appears that this was not presented in a clear enough manner in our original submission as Reviewer 1 also raised this point. As such, we have added a better description about how the ablation was performed, including examples with the calibration using nuclear stainings. We have included a new figure (Figure 4—figure supplement 1), a new video (Video 14) and text in subsection “Ingressing cells instruct neuronal specification”. We only targeted red cells for ablation and imaging after ablation confirmed the restricted damage in these DsRed+cells. We cannot discard that, exceptionally, a neighbouring *neurog1*- cell could also be damaged. However, as we show in the new figure and video, the ablation is highly specific and neighbouring cells were not affected.

*3) Is the reduction of dsred+ precursors due to the specific ablation of ingressing cells or is it due to the reduction of the overall number of dsred+ cells? That is, could there be a 'community effect' where dsred+ cells induce others irrespective of their initial origin outside the otic vesicle?*

If we understood correctly, what the reviewer proposes is that the phenotype observed after ablation (a reduction in both the number of *neurog1+* cells and the mean expression levels) could be produced by a specific action of the ingressing cells on otic neuronal specification or be a consequence of a reduction in the number of *neurog1+* cells independent of their identity as ingressing cells. As ablating only resident *neurog1+* cells inside the otic vesicle is not simple, therefore we designed an experiment to distinguish between the two proposed explanations for the ablation phenotype (specific role of ingressed cells vs community effect). To reduce the number of *neurog1+* cells by other means, we inhibited proliferation during vesicle morphogenesis using a combination of aphidicolin and hydroxyurea (AH), as we have done previously (Hoijman et al., 2015). The aim was to reduce the total number of cells, and thus probably the number of *neurog1+* cells, and to evaluate what happens to the other parameter affected by ablation: the mean *neurog1* expression levels. If ablation led to a reduction in these levels due to a reduction in the number of *neurog1+* cells, reducing the number of cells by other means should produce a similar reduction in mean expression levels observed in ablation experiments. Our results (Figure 4—figure supplement 2, subsection “Ingressing cells instruct neuronal specification”) indicated that although the number of *neurog1+* cells was highly decreased by the AH treatment, the mean expression levels of the remaining cells did not change. This experiment suggests that both parameters (cell number and mean expression levels) are not necessarily linked one to the other, and also that a reduction in the number of *neurog1+* cells is not enough to produce the same phenotype observed by cell ablation. Thus, a “community effect” mentioned by the reviewer does not seem to underly the ablation phenotype. Rather, pioneer cells seem to have a specific influence on *neurog1* expression.

*4) Does ablation of dsred+ ingressing cells alter proliferation of cells within the otic vesicle?*

We cannot discard that this is in fact occurring, although several arguments suggest that is not the case. First, the morphology of the otic vesicle and the neurogenic domain were not affected by ablation, suggesting that a major change in proliferation does not occur (in AH experiments, the vesicle and the neurogenic domain becomes smaller). Second, the inhibition of cell ingression by FGF blockade, which does not eliminate the ingressing cells themself but only their ingression, impair neuronal specification without affecting cell division (new experiment added to the manuscript showing that the number of pH3+ cells in the vesicle is not affected by FGF blockade, Figure 5—figure supplement 1, subsection “FGF controls otic epithelialization”). Third, as the AH experiment shows, cell number and *neurog1* levels are not neccesarily coupled. Although the decrease in the number of *neurog1+* cells after ablation could be mediated by the reduction in proliferation induced by the ingressing cells, the observed decrease in the mean *neurog1* expression levels must be achieved by another mechanism. This would require two independent pathways for instruction, which seems less probable to occur. We have now added two fragments of text related to the role of proliferation in *neurog1* activation.

*Reviewer #3:*

*In this manuscript the authors study otic neurogenesis. They describe a population of pioneer cells in zebrafish, which arises outside of the otic placode, invades the placode and generates neurons. Their experiments suggest that these cells play a role in promoting neurogenesis in neighbouring otic epithelial cells, since their ablation reduces neurog1 expression (both levels and cell numbers). They also perform morphometric and proliferation analysis of the otic vesicle, and finally some experiments to suggest that FGF signalling somehow controls the integration of the pioneer cells into the vesicle.*

*The idea of pioneer cells is an interesting and novel finding that warrants further investigation and provides a novel view on how neuroblasts are determined in the ear. However, throughout the manuscript the authors make quite forceful conclusions that are not always supported by their data (e.g. for most experiments numbers are extremely low, or not given).*

All the numbers of embryos analysed in each experiment were already provided in the legends of the figures, as suggested in *eLife* instructions.

We believe that the numbers are large enough to support our conclusions. Moreover, given that extensive quantitations were performed on each embryo to obtain quantitative data, the statistical analyses performed also support our conclusions. This is not what ususally happens in developmental biology papers, in which qualitative representative pictures of the phenomena (immunostainings, in situ hibridization) are shown. Here we provide not only representative images, but also quantifications of different properties of those images, in some cases a dynamic behaviour from videos, and thus we quantified the data coming from all the embryos of the experiment.

*The morphometric analysis does not add much to the paper, and I wonder if it should be removed. The FGF results are very preliminary and in their current form are not conclusive. The authors do not provide a clear model for FGF function and thus do not provide any mechanistic insight.*

*Below, some specific comments that the authors should address.*

*1) Throughout the manuscript the figures need to be improved. To help the reader figures need better labelling: label SAG in Figure 1 (and other figures), label delaminating neurons to distinguish them from neuroblasts in the vesicle. Outline the otic placode/vesicle in figures and movies and add arrows for orientation (anterior-posterior, medial-lateral) into the movies. In some videos it is difficult to see which neurog1-dsRedE cells are in the vesicle, remain outside or delaminate. It needs to be clear which are the SAG cells and the placode cells.*

As suggested by the reviewer, we have incorporated the axes in all figures and videos. We also added labels for the SAG (in Figure 1 the SAG was already labeled (last panels of Figure 1), and a 3D reconstruction video showing where the SAG is located (immediately anterior and ventral to the otic vesicle). We want to highlight the extraordinary 4D spatiotemporal resolution obtained in our videos and showed in our images, that is able to reveal the position of single cells and cell shape changes at all times.

As the reviewer asked, we have added outlines of the otic primordium/placode/vesicle to all figures and videos, in order to help the readers to distinguish delaminated cells from placodal epithelialised cells. However, some considerations should be taken into account. Firstly, with the usual resolution images of ISH or immunohistochemistry expression, the establishment of a limit for the placode is very rough but sufficient for a qualitative spatial analysis. In our work, some phenotypes were shown in Z-projections images because they are better observed in the conjuction of all confocal planes and not evident in a single plane (this is a way to provide a 2D representation of 3D information). However, in z-projections an exact and unique delimitation of the otic vesicle is not accurate, given that the epithelium limit is positioned different in each z-plane due to its curved shape in the dorso-ventral axis. Independently of the visual representation of the phenotypes, we would like to highlight that our conclusions were based on the quantitative analysis performed in single planes. Whether cells are inside or not of the vesicle and their quantification of the DsRedE fluorescence level were performed plane by plane in each single cell (around 50 Z-planes for each embryo since acquisition is taken every 0.5 µm in the z direction, see the right panel in new Video 12), and comparing different planes for each cell. These quantification and classification processes were already explained in the Materials and methods section. Secondly, analyzing single planes from z-stacks, we took a criterion to classify a cell as delaminating: the movement of the cell body to the basal domain of the epithelium. However, outlining the exact limit of the otic vesicle at this subcellular micrometer scale has no real meaning, as it is a dynamic process. Thirdly, at early stages of tissue epithelialisation, although it is easy to distinguish by cell shape if a cell is epithelialised or not, a clear limit of the placode is difficult to outline in some regions, because it is in fact not a real entity. Finally, as some of our images concentrate on what happens in the limits of the placode (cell ingression and delamination), the outline of the limits impairs the visualisation.

To make more clear the terminology of the field for the reviewer, the cells that delaminate are not neurons, are expressing *neurod1* and they have to transit to the SAG where they will differentiate (becoming post-mitotic, extending axons) to become neurons. The cells that express *neurog1* in the vesicle (neuronal precursors) are not a distinct population from the delaminating ones, but different stages of maturation of the same pool of cells.

*2) Morphometric analysis in the Results section: these measurements do not add very much to the paper and do not really allow any particular conclusions and distract from the main part of the paper. Maybe this section should be removed.*

*If this section remains in the manuscript, the authors need to provide numbers: how many placodes were analysed? How representative are the results? This sentence is unclear: "The modest increase in the number of cells in this region cannot account for the large enrichment in cell proliferation, suggesting a higher proliferation rate in the NgD region." The authors conclude that the neurogenic domain has a higher proliferation rate, but have not assessed proliferation rate and therefore this conclusion is not valid.*

The morphometric analysis provides the first quantitative evidence of a regional enriched proliferation in the neurogenic domain that highlights the importance of cell division for the homeostasis of the neurogenic domain. This raises the question of how cell divisions are coordinated with neuronal specification, which we address later in the manuscript. Moreover, the regional counts in cell number allow us to suggest that the increased number does not underlie the increased proliferation observed, but that proliferation is being promoted in this region, providing relevance to this process during neuronal specification. As suggested by the reviewer, we have rephrased the text on our revised manuscript to make this point more clear:

“While the increase in cell number in the neurogenic domain was moderate (about 3% more cells than other domains), the enrichment in mitotic events led to about 41% of the total number of divisions to occur in this domain. The large difference in the change of the two parameters suggests that the increased number of mitotic events is not a consequence of having more cells dividing at the same rate, but due to a specific increase in the proliferative activity of these cells.”

We consider that in fact have measured a proliferation rate. By rate, we understand “A quantity measured with respect to another measured quantity”. Frequently the second quantity is time. We have measured proliferation by identifying each event of cell division in the different regions of the vesicle during a period of 4.5 hours. We believe that this can be called a proliferation (or cell division) rate, that is, the number of mitotic events per unit of time. In our phrase, we discard a probable cause for this increased rate: the presence of more cells in this domain.

The numbers of vesicles analyzed are in the figure legend of Figure 1 = 11). These results are highly representative; we provided the mean and standard error and a statistical test to compare means and distributions.

The shape, density and cell number analysis are interesting to highlight that the neurogenic region has a particular morphogenesis, which is important to link with cell ingression and the morphogenetic mechanism underlying the FGF effects.

*3) The authors say that neurog1 cells become integrated into the vesicle during placode formation; can they specify the timing and explain this better for readers unfamiliar with the fish and with the otic development. This is also picked up in the Discussion, however the timing is not very clearly described in the paper.*

We thank the reviewer for the suggestion and have added a phrase about this. We also moved a paragraph originally in the legend of Figure 2—figure supplement 1 to the main text, which we think might be useful to better explain the epithelialisation process and timing. Developmental times were already indicated with high detail in Figure 2 and Video 5, Video 6 and Video 7 (and now in the added Video 8).

*4) In the Results section; Figure 2: The authors describe that neurog1 + cells from outside the otic vesicle become integrated into the otic vesicle, and generate neurons of the SAG. In Figure 2, Video 6 they show that some cells exhibit this behaviour, while others do not. It is not clear how this behaviour is determined; are those cells that do invade the vesicle always neurog1 positive? The authors need to compare the behaviour of neurog1 + cells; it is quite possible that only neurog1 + cells exhibit this behaviour, and that this is the reason that position of cells is irrelevant.*

In Figure 2 and Video 6, the conclusion about position of the cells was included to highlight the fact that not all the cells from the migrating field ingress in the vesicle, and thus this is not a global tissue movement, but the migration of individual cells. Concerning ingressing cells, we know that most are *neurog1+* (or express *neurog1* shortly after ingression). Furthermore, a comparison of the behavior of different *neurog1+* cells is included in the manuscript, and as we stated, it is clear that many *neurog1+* cells of this field do not ingress in the otic vesicle either. These cells belong to the “second pool of neurog1+ cells”, which also migrate in the same region but move to the SAG. We conclude that *neurog1* expression in these migrating cells does not seem to target them to be incorporated in the otic vesicle.

*5) What are the cells in the "second pool of neurog1+ cells"? do they contribute to the SAG, other cranial ganglia?*

The cells belonging to a “second pool of *neurog1*+ cells” contribute directly to the SAG without transitting through the otic epithelium. Although these cells migrate in the same region as the cells that are incorporated in the otic vesicle, we observed in double *Tg(neuroD1:GFP);Tg(neurog1:dsRed)* transgenics that this second pool begins to express neuroD1 during migration, while cells that ingress do not (they express neuroD1 only after delamination). Our observations might imply that the second pool cells are in a more advanced phase of neurogenesis. Alternatively, perhaps this observation reflects heterogeneity in neuronal populations of the SAG. All this is speculative, however, and since the study presented here does not focus on the SAG, we have yet to design and perform experiments into this second pool of migrating cells. We will certainly continue studying them in a future work.

*6) What do the authors mean? "Additionally, other morphological features particular of these stages could contribute to cell ingression"*

As indicated in the text, this phrase refered to Figure 2—figure supplement 1. In the original figure legend of that supplement figure, it was exhaustivelly explained what we meant. To make this text more visible, we have now moved it to the main text. The aim of the phrase was to highlight the fact that many morphogenetic events might be allowing the cells to ingress at these stages.

Here we clarify the explanation for the reviewer: the otic placode is epithelialising in these stages and the posterior region folds before the anterior one (Figure 2—figure supplement 1 and Video 8). This could give more time to anterior cells to ingress. We also show that basal lamina of the otic vesicle is not continuous at this stage (Figure 2—figure supplement 1), another feature that could facilitate ingression. Additionally, we identify an F-actin layer that separates the field of migrating cells from a more medial region (Figure 2—figure supplement 1), which probably helps to restrict the migration of cells in the anteroposterior direction. We decided to incorporate this part in the main text because we now added a time-lapse of embryos expressing a dominant negative of the FGF receptor 1 (Figure 5 and Video 13, see comment 11 for more details). This video allowed us to visualise an anomaly in the folding of the otic placode when the FGF pathway is blocked and therefore the description of the morphogenesis of the placode became more relevant. When the dominant negative is expressed, the anterior region of the vesicle folds earlier than in controls, leading to a simultaneous folding of the anterior and posterior regions, and thus possibily impairs anterior cell ingression. Additionally, laminin staining in these embryos also revealed a precocious formation of the basal lamina (Figure 5, see comment 11 for more details).

*7) In the Results section: the authors suggest that levels of neurog1 determines when cells delaminate from the vesicle. Can they provide numbers: how many cells were measured? What is the threshold? Please provide statistics to support this claim.*

Each cell measured requires a long 3D track in which fluorescence levels are measured over time. We have now increased the numbers of cells and measured 11 cells. The absolute value for the threshold is about 45-52 a.u., but these numbers do not have any biological meaning, as they depend on the settings of the microscope. We believe that in the new graph (Figure 3) is easy to visualize the fact that delamination occurs at a similar level of DsRedE fluorescence (the value in the Y axis corresponding to the red dots, now also highlighted by a gray region) but at different time points (the value of the X axis corresponding to the red dots). We also added a box plot (Figure 3) of the data normalized by their mean, illustrating the highly different dispersion between fluorescence levels and elapsed time at the moment of delamination.

*The final conclusion from this section is: "This suggests that cells delaminate relative to neurog1 levels and not to the time elapsed from the beginning of neurog1 expression". The authors do not measure the time a cell spends in the vesicle; how do they come to this conclusion?*

We do not understand exactly the question raised by the reviewer. As we indicated in the text, we indeed measured the fluorescence levels in cells starting to express the DsRedE protein inside the vesicle and all along their way to the moment that they leave the vesicle by delamination. However, the reviewer refers to the time that a cell spend in the vesicle (the moment from epithelialization or ingression, to delamination), and this is not the same that the time that elapses from the moment that *neurog1* start to be expressed in cells of the vesicle to the moment that these cells delaminate. We measured and made statements about the second parameter, but we did not measure and did not claim anything about the first one.

*8) In the Results section the authors re-visit cell division within the neurogenic domain, which is already described in the first section of the paper. It is not clear what this section adds and why Pard3 staining is relevant in this context, and what this adds to the main message of the paper. The whole section seems a bit out of place. The conclusion is that "our analysis of the origin of neurog1+ cells revealed that they are added to the NgD by three different mechanisms: cell ingression, local expression and cell division. They are not really different mechanisms, neuroblasts are known to proliferate and expand before becoming terminally determined and this observation is therefore not surprising.*

One of our aims was to understand the cell dynamics of specification, and how *neurog1* expression is induced and increases over time in the vesicle. Until now, otic *neurog1* expression (and this can be extended to many other tissues) was analyzed in fixed samples, and therefore this led to the idea that resident interphase cells start to express *neurog1* and divide after delamination during the transit to the SAG. Our analysis of proliferation changes this view. Firstly, it identifies the neurogenic domain as a preferential site of proliferation. Secondly, and most importantly, *neurog1+* cells appear to divide inside the vesicle, which was not known before, and is one of the causes increasing the number of *neurog1+* cells inside the vesicle. Thirdly, the analysis of division mode is relevant to couple this process with the fate of the cells and compare it to what is known from other neural tissues. While in the CNS, it has been well documented that initially there is a phase with proliferative symmetric divisions, later on asymmetric neurogenic divisions emerge. Therefore, our analysis of the modes of divisions, a question still not tackled in the otic vesicle, is relevant. Moreover, we observed that there is not a hierarchical sequence between division and *neurog1* expression. We find this section important, therefore, as our aim was to understand the dynamics of *neurog1* expression and the construction of a neurogenic domain.

The Pard3 staining is relevant for the analysis of division mode, and exemplifies how increasing 3D resolution and specific labeling helps avoid spurious conclusions. In a previous work, Alvarez et al. (J Comp Neurol. 1989 Dec 8;290(2):278-88) the presence of otic basal divisions in the chick neurogenic domain was preported. Here, we show that the peripheral divisions that we observe are in fact apical. In the CNS, neural stem cells divide in the luminal side, while intermediate progenitors divide symmetrically in basal locations not in contact with the lumen. In those works, whether those basal divisions are connected to extensions of pard3 is unknown. Altogether, these observations highlights that, the fact that in other neural tissues neuroblast divide basally or express a proneural gene before division, does not imply that this should occur in every tissue. Our analysis clarifies for first time how this occurs in the otic neuronal precursors. Thus, neuroblast “expansion” does not explain how they perform this expansion, and particularly, how this is temporally coordinated with proneural expression. We believe that adding a detailed description of otic neurogenesis is significant for comparative analysis in the overall field of neurogenesis.

We also believe that mentioning three mechanisms to establish the neurogenic domain (a field of *neurog1+* cells) is appropriate, given that all these analysis cannot be resumed as a simple “expansion” of neuroblasts. We have analyzed the origin of neuroblasts, the beginning of *neurog1* expression before division; how these cells start to belong to the neurogenic domain. This allowed us to highligh the different sources for the origin of all *neurog1+* cells present in the neurogenic domain.

*9) The authors show that ablation of neurog1+ cells (pioneer cells) leads to a reduction of neurog1 expression in the otic vesicle. They suggest that the number of ablated cells determines the how many neurog1+ cells later appear in the vesicle as well as the level of neurog1 expression in each cell. The authors only show 2 embryos; this is not sufficient to reach this conclusion. Are there more specimen; please provide numbers.*

We have focused in the effect of ablation that we showed in the quantifications of Figure 4 (originally 4 embryos, now performed 2 new ablations and thus 6 embryos in total). It is true that we show only 2 embryos for the dose-dependent effect of ablation pulses on *neurog1* expression in the vesicle. However, we observed a clear trend in ablations with different number of pulses, although we do not have enough number of embryos to perform statistics about this issue. We still think this dose-dependent effect is in fact occurring and it is relevant for the readers of the work, therefore we have maintained these data but tuned-down our conclusion. If the reviewer considers that we should remove this data on the dose-dependency from the manuscript, we could do it.

*The authors state that otic neurogenesis does not recover after ablation, but do not provide much evidence (2 embryos in Figure 4—figure supplement 2). I would like to see more examples to reach such a firm conclusion. What happens to the SAG? Is there other compensation? Embryo 2 in Figure 4—figure supplement 2 appears to form some neurog1 cells eventually; have the authors looked later?*

*Do the authors suggest that the majority of otic neuroblasts is induced by the pioneer cells? If so, this is interesting and novel, but this requires more data with solid statistics and longer observations.*

*The authors measure the level of fluorescence in individual cells in pioneer ablated and non-ablated embryos: how many embryos were analysed, how many cells?*

We showed 2 embryos in Figure 4—figure supplement 1 instead of only a representative embryo as usually used in the papers. We also previously included a detailed visualisation of a third one in Video 10 of the previous version of the manuscript (now improved in Video 12), and the total number of embryos for this stage is 4 (now Figure 4—figure supplement). To better characterize a possible late compensation and their influence in the SAG, we have performed new ablation experiments at the same time and position as previously, but led the embryos to develop until 42 hpf. In Figure 4—figure supplement 2 C, it is possible to see that only one laser pulse led to a severe reduction in the size of the SAG at late stages (n =3), as compared to the contralateral control side of the embryo. This suggests that compensation is not occurring and the defect is permanent.

That embryo 2 in Figure 4 indeed presents some *neurog1+* cells (and with reduced mean levels) is not surprising, as not all the ingressing cells are killed. Indeed, it is very difficult to determine exactly the effect of eliminating all the ingressing cells, because this is technically impossible. Thus, it is also possible that some resident cells do not require the ingressing ones to express *neurog1*. For this reason, we do not claim that all the resident neuroblasts are instructed by the ingressing cells. In our opinion, however, the fact that killing only 2-3 pioneer cells leads to very strong phenotypes in otic neuronal specification is highly novel and supports our claim of the importance of ingression.

The numbers of measured embryos were originally indicated in figures legends. We used 4 embryos in the original version and obtained very significant results, given that the effect is strong and reproducible. We now added 2 more embryos to the data shown on Figure 4. The number of *neurog1+* cells analysed was already explained in the main text and the text of the figure legend, and we now added the words “all cells” to make it more visible. We have quantified cell by cell (and plane by plane of the 3D stack), all the *neurog1+* cells present in the neurogenic domain of each vesicle. When quantification of two single vesicles were shown as examples (Figure 4 and Figure 4—figure supplement 2), the fluorescence of each cell is shown as a dot, the number of dots is the number of *neurog1+* cells. When the graph is a quantification of several embryos (Figure 4, Figure 5, and Figure 5—figure supplement 5C and D; n of embryos is indicated in the figure legend), mean and standard errors are depicted, and statistical tests to compare means performed. For example, the mean number of *neurog1+* cells in Figure 4 are 23.8 ± 1.4 for the contralateral non-ablated side, and 10.0 ± 0.8 cells for the ablated one (these numbers were originally included in the main text).

*10) In zebrafish the otic vesicle does not delaminate as an epithelium but rather coalesces from cells that then form a lumen. As such it's quite possible that addition of cells to the vesicle continues after initial lumen formation. Therefore the issue in my mind was whether there was specificity for dsred+ cells invading. If nonspecific then the FGF requirement is just a continuation of the requirement for vesicle formation. If nonspecific then the specificity of the ablation needs to be determined – ablation of bystander dsred- cells will also contribute to any observed phenotype.*

As we discuss above, the ingressing cells are mainly *neurog1+* cells (see the answer to comment 4). We do conceive the requirement of FGF as a requirement for vesicle morphogenesis, because we have shown that ingression of these cells is an integral part of vesicle morphogenesis, which is coordinated with the timing of anterior folding of the placode. However, when FGF is blocked, the failed ingression did not seem to be a consequence of a large anomaly in vesicle formation, because it does form similarly, but with a specific defect in otic neurogenesis. Moreover, this phenotype comprises both a reduction of the number of *neurog1+* cells and a reduction in the mean levels of expression of this gene in each cell.

We did not detect ingression of cells after the basal lamina is completelely formed, and in fact we observed that delamination involved local breaks in this lamina (our unpublished results), suggesting that the period for ingression is transitory and associated to placode formation and completed epithelialization.

We have now added a better description about how we performed the ablation, including examples with the calibration using nuclear stainings. The added parts include a new figure (Figure 4—figure supplement 1), a new video (Video 14) and the following text:

In main text: “Ablation of a limited number of cells (2-3 cells per laser pulse)”

In the Materials and methods: “We used embryos with mosaic H2B-mCherry nuclear staining (mRNA injected at 16-cell stage) to calibrate the settings of the microscope required to ablate 2-3 cells in each ablation pulse (Figure 4—figure supplement 1; Video 14). Each pulse consisted in approximately 5 seconds of 30% laser power applied in a ROI of about 70µm2 imaged with a 20x air objective and a digital zoom of 64x. In neurog1-DsRedE embryos”,

In the figure legends: “Calibration of cell ablations. A laser pulse (as described in Materials and methods) was applied to embryos expressing H2B-mCherry in some cells adjacent laterally to the neural tube. In example 1, two nuclei were stained in the imaged region before ablation (white arrowheads). After the laser pulse, a red ablation buble was observed as consequence of the death of the two stained cells (blue arrowhead). In example 2, the nuclei of neighbouring cells (numbered from 1 to 5) are surrounding two target cells (white arrowheads). Imaging after ablation indicated that the targeted cells died, but the neighbouring cells remained healthy and only slightly displaced in space. In example 3, a similar behavior than in example 2 can be observed, but the intact neighbouring cells are in close contact with the dead cells, highlighting the fact that ablation is highly specific and restricted to the targeted cells (white arrowheads, see also Video 14)”).

We targeted red cells to ablate and we observed the damage in these cells, therefore ablation mainly kills DsRed+cells. Thanks to our high cellular resolution imaging before and after ablation, we believe that we achieved a higher precision in the control of the number of dead cells by two photon-laser ablation than the usually achieved when this methodology was used for these purposes. We have also performed new controls for ablation of DsRedE expressing cells, applying laser pulses at other locations (posterior to the placode, 13 hpf) or developmental stages (anterior to the placode, 19 hpf) than in previous experiments (anterior to the placode, 13 hpf). In both controls, we did not observe a difference in specification a few hours after ablation, indicating that the effect is specific for the *neurog1* positive and anterior cells and not a general consequence of cell death (or high laser power) close to the placode (Figure 4—figure supplement 1 and subsection “Ingressing cells instruct neuronal specification”).

*11) The FGF results are not convincing; more experiments are needed to provide a clear view of what exactly FGF is supposed to do in this context. The authors should provide a clear model of how they suggest that FGF acts: what is the source of FGF? Do pioneer cells have FGF receptors? Do they suggest that FGF is an attractant? Why do not all neurog1 cells invade the placode if that is the hypothesis? How can the effect on pioneer neuron invasion and proliferation be unravelled? Does FGF have multiple effects, not only affecting pioneer invasion, but also local induction of neuroblasts and their expansion?*

*Simply blocking signaling by SU5402 does not provide definitive proof that FGF is involved since other receptors are also affected by this drug. The authors need to use different ways to show that FGF is involved; these could be dominant negative receptors in pioneer cells, cell type specific knock-down of FGF response or secretion. Without further experiments this part of the manuscript opens up too many unanswered questions.*

In response to the reviewer´s comment, we have added new experiments, text and figures to clarify the participation of FGF in the processes analyzed in the work. We obtained a transgenic line to express transiently a dominant negative protein of the FGF receptor1 transiently after a heat-shock (*hsp70:dnfgfr1-GFP*). As this protein is fused to GFP and once expressed it goes to the plasma membrane, we checked for expression of the dominant negative at the microscope. We incubated 10 hpf embryos at 39 ºC for 30 minutes and then selected the induced (FGF signaling blockade). With this line, we performed two types of experiments: 1) the NLS-Eos photoconversion experiments to evaluate ingression (Figure 5), and 2) the *neurog1-DsRedE* expression analysis to evaluate neuronal specification (Figure 5; in this case the *hsp70:dnfgfr1-GFP* line was crossed with the *neurog1:DsRedE* line). For both, we obtained similar results to the ones obtained with the SU5402 inhibitor. From this, we strongly believe we can conclude that the FGF signaling pathway is involved in these processes. To gain insight about how this pathway is controlling cell ingression, we also performed time-lapse imaging after the induction of the dominant negative. We observed that, different to the asynchronic folding of the anterior and posterior regions of the placode observed in the control embryos, when the dominant negative is induced the anterior region folds earlier than in controls, and thus a symmetric anterior and posterior folding occurs (Figure 5 and Video 13). These results led us to propose that FGF is delaying the folding of the anterior region, thus giving time to the migrating cells to arrive before folding. When the pathway is blocked, the tissue folds earlier and cells cannot ingress. Accordingly, laminin immunostainings indicated that expression of the dominant negative accelerates the formation of the basal lamina (compared to the control embryos), which can interfere with ingression (Figure 5). Thus, we propose that the FGF pathway is controlling essential steps of the otic placode morphogenesis, thus influencing the capacity of the pioneer cells to be incorporated in the epithelium. Accordingly, FGF controls morphogenesis in other organs, and particularly in the chick otic vesicle. Altought we cannot discard the possibility that migration of the pioneer cells is also directly affected, we do not think in a chemoatractant effect of FGF. Firstly, by suggestion of reviewer 1, we evaluated and discarded a possible participation of the CXCL12a/b-CXCR4a/b chemokines-receptors (see Figure 6). Secondly, as here the reviewer indicates, the presence of *neurog1* expression does not seem to be sufficient to induce the expression of a chemotractant, because many *neurog1*+ cells from this region do not move to the otic vesicle and ingress (maybe the presence of *neurod1* in these other cells can provide such specificity). Thirdly, the main FGF ligands expressed at these stages are *FGF3* and FGF8, which are located in the hindbrain and the mesoderm. Therefore they do not seem to be able to direct cell migration to the lateral side of the otic placode (*Fgf10a* is also expressed at this stage, in a region close to the pioneer cells; we discarded a possible role of this FGF on neuronal specification by crossing a mutant *fgf10a* line with the *neurog1:DsRedE* line and performing the specification analysis as described; we did not find any differences with the siblings embryos, see Figure 5—figure supplement C and D).

Regarding the FGF ligands mediating the effect on placodal morphogenesis and ingression, most probably are *FGF3* or FGF8, the two ligands expressed at these stages in the region. Given that mutant embryos for these genes produced earlier phenotypes related to otic induction, they cannot be used to analyze their participation in later events, and conditional mutants are still not available. We added a discussion about the ligands in the text.

To evaluate the effect of FGF on proliferation, we now performed immunstainings for phospho-Histone 3 in the otic vesicle (Figure 5—figure supplement 1). We did not find an effect of SU5402 on the number of dividing cells in the otic vesicle, suggesting that this is not the main mechanism by which FGF controls neuronal specification. Moreover, a putative proliferative effect would affect the number of neurog1+ cells, but not necesarily will affect the mean expression levels (this is exacty the phenotype that we observed in a new experiment incubating the embryos with aphidicolin and hydroxyurea to block cell proliferation (Figure 4—figure supplement 2).

As we stated originally in the Discussion, the inhibition of cell ingression by the FGF blockade would explain some of the known FGF pathway effects on neuronal specification, but a direct action on *neurog1* expression levels could also occur.

*12) Discussion section. The authors need to be more careful with their choice of words (there are many examples throughout the text); they suggest that the pioneer neurons are the first "specified cells" and "promote neural commitment" – we do not know if these cells are truly specified; this can only be assessed by culturing them in isolation. What do the authors mean by promoting commitment? That pioneer neurons induce neighbouring cells to become neuroblasts?*

The term “specification” has been used for many years in developmental biology for the commitment of progenitors to neural or neuronal fates “in vivo” (two recent examples are: Kicheva et al., Science 2014, 345:1254927; Xiong et al., Cell 2013,153:550). Moreover about 20 years ago *neurog1* was one of the earliest discovered genes with typical proneural function, responsible for neuronal specification (required and sufficient), and acting in the otic field and also in zebrafish (Ma et al., Neuron 1998, 20:469; Andermann et al., Dev Biol 2002, 251:45; Adam et al., Development 1998, 125:4645; Bertrand et al., Nat Rev Neurosci 2002, 3:517; Blader et al., Development 1997, 124:4557). As we mention in the Introduction section, *neurog1* expression leads to *neurod1* expression, then these cells delaminate, transit-amplify, incorporate to the ganglion and finally differentiate as neurons. Therefore, we have chosen to keep the term used in the field of neural development in general, and otic development and zebrafish development specifically (recent examples: Kantarci et at, Plos Genetics 2015, 11:e1005037; Hans et al., Development 2013, 140:1936; Radosevic et al., Development 2011, 138:397).

By promoting commitment, we mean to promote other cells to express neurog1 and thus commit to a neuronal fate. In this case, as the reviewer mentioned, we could use “pioneer cells induce neighbouring cells to become neuroblast”. However we did not used this expression for several reasons: Firstly, we did not want to use induction, because in the field of placodal development this terminology is associated with other developmental processes. Secondly, as we observed an increase in *neurog1* levels, promotion sounds more accurate than induction, which evokes more an all or nothing phenomenon. Thirdly, we used commitment with the word spreading, to transmit the idea that pioneer cells are the first specified cells and that their presence leads to an expansion of neurog1 expression locally.

*13) The Discussion section is a bit convoluted and the arguments are not clearly structured, and often the authors seem to over-interpret their data. For example, the authors argue strongly that positional information does not determine whether or not a cell adopts pioneer identity, but it may equally be possible that by the time they evaluate this cells are not competent to do so.*

In the section of the Discussion related to the positional information we used the term “suggest” as we were aware that this is speculative. When calling into question the traditional view of positional information, we were addressing what happens inside the vesicle. We mentioned that positional information within the neurogenic domain does not seem to be acting as it has been presented in the past, because our data shows that some cells are specified before ingressing. More importantly, we never talk about the acquisition of pioneer identity inside the neurogenic domain: “when ingressing cells are laser ablated, the cells in the otic vesicle located in the position of the ingressed cells (i.e. receiving the same putative diffusing morphogens) do not seem to adopt a neurogenic fate”. This sentence refers to the resident cells that are inside the vesicle and talk about how these cells acquire, not pioneer identity (ingress and instruct) but a neurogenic fate (neurog1 expression). Thus, what is stated is that the *neurog1* signal in the neurogenic domain coming from the pioneer cells that ingressed is not replaced by other resident cells of the vesicle located in the same place. The resident cells in the epithelium, in absence of the pioneers, do not take the place of the ingressed cells and acquire neuronal committment by the action of a diffusing morphogen. This highlights the importance of coordination of cell movement with the signals secreted from the surrounding tissues. The hypothesis proposed by the reviewer about the time is interesting, but supports instead of contradicting our idea that cell dynamics contributes to a possible positional information mechanism. There are cells in the vesicle located where the ablated pioneer cells should be, but they are not being specified, due to a “temporal” or another unknown reason (our proposal is as simple as that ingressing cells are specified outside and then move, they are not *neurog1+* because they are there receiving a morphogen when inside the vesicle, they have been *neurog1+* from some time and then move there). Therefore, a temporal explanation also suggests that there is more than positional information.

*14) It is surprising that the authors do not refer to an older paper by Hemon & Morest 1991 describing the 'otic crest', as well as a more recent paper describing neural crest contribution to the otic vesicle in mouse. This may be very relevant to the current study.*

*In this context, the authors should discuss whether their model could also be true in higher vertebrates like chick and mouse, where development of the placode is a much longer process than in amphibians and fish. For example, in fish and frog neurogenic markers are expressed much earlier than in mouse and chick; could their observation be species specific?*

The Hemond and Morest paper was originally cited in our references as one of the first ones studying delamination, but it was wrongly cited as Morest only (this has been amended in the references).

We thank the reviewer for the suggestion to discuss on the otic crest. We have added a paragraph in Discussion section. It is possible that otic crest could correspond to the “second pool” of migrating *neurog1+* cells that we found to move directly to the SAG. The invasion reported in mouse was useful to discuss a possible evolutionary conservation of ingression in the otic vesicle.

In chick, the otic placode is being formed at 10 ss, while in the zebrafish it happens the same. The invagination of the otic placode in chick initiates at stage HH12 (16ss) and in zebrafish lumen appears also by the same time. While in zebrafish everything indeed goes faster in time, the relevant process ocurr at same developmental times, indicating a conserved mechanism of otic development in spite of differences in the formation of the lumen.

Interestingly, in chick we published data showing that Δ is expressed inside and outside the forming placode at very early stages (Figure 4, Abello et al., MoD 2007). The cells outside where believed to be from the geniculate placode. However, the present paper highlights the dynamic nature of cells at those stages, suggesting that maybe in chick those anterior Δ cells could ingress in the chick placode.

*15) Terminology:*

*'Ingression' is not the right word to describe that individual cells from outside the otic vesicle are integrated into it. They invade, are incorporated, inserted or similar.*

We understand the comment of the reviewer, but we use ingression to define something (a cell) that is first outside and then inside of a structure (an organ), which believe is the most accurate word to describe the process. We are open to change the term if the reviewer suggests a better word. The ones mentioned are not completely satisfying. “Invasion” is frequently used for cancer cells that invade organs already formed, and also suggests an active mechanism of penetration between cells (instead of the mesenchymal-epithelial transition observed here). “Insertion” is similar to invasion. “Are incorporated” although useful in some contexts, is not a good term to refer to this group of cells as “the incorporated cells”. We believe “ingressing cells” and “ingressed cells” is more clear.

*'Local specification': in developmental biology 'specification' is used to describe along which path cells or tissues can differentiate when cultured in isolation. Therefore in this context the term 'local specification' is somewhat misleading. The authors should find a better term.*

As we mentioned above, it is strongly demostrated that otic cells are specified by *neurog1* expression as in the absence of the proneural factor neurons of the SAG fail to form, or are not specified. By “local specification” we we have added a modifier that defines where the process takes place, and emerges from our dynamic analysis. We use this term to highlight that, in some cells but not in all cells (as was previously assumed by the static and low resolution analysis of otic neuronal specification performed until now in the field), *neurog1* expression is induced inside the epithelium (“locally”). Using this term, we can distinguish these cells from the ones who express neurog1 outside the vesicle and before epithelialization and ingress in the primordium while already *neurog1* positive. By using the word local, we shorten the sentence. If the reviewer still thinks that is not the appropiateterm we could change “local” to “inside the tissue” or “intra-organ”, but we think they are too long and local is more clear.

*16) A native speaker should read the manuscript; it contains errors, and peculiar constructions and expressions making the text at times a bit cumbersome to read. E.g. "Pioneer cells specify outside the otic primordium and ingress during otic placode formation". Should read: 'are specified'.*

A native speaker has corrected our text and we hope that now the text is improved.

17) The authors should avoid too many abbreviations; they make the text more difficult to read in particular for readers outside the field. There is no reason to abbreviate terms like 'neurogenic domain', 'global level of dsRedE expression' (GLE) or similar.

We removed these abbreviations from our revised manuscript.

[Editors' note: the author responses to the re-review follow.]

*Essential revisions:*

*1) If their neurog- neighbors are also joining the placode, then ingression is not a distinct behavior of 'pioneer' cells. While this observation does not detract from authors' main conclusions the specific roles of the 'pioneer' cells in promoting the formation of later neurons, it does speak to whether the 'pioneer' cells are actually outside the otic primordium (defined as the group of cells that will form the otic placode). The authors make a convincing argument that global tracking would be time-consuming and not directly applicable to answering the central questions of the work presented. I therefore suggest that they drop claims that these 'pioneer' cells are outside the otic primordium as stated in the Abstract and elsewhere.*

We do not understand what the editor is proposing by taking out that the pioneer cells are outside the otic primordium. By primordium we refer to the otic placode, we could change this word “primordium” to “placode” if this is generating the confusion. The otic placode is a visible epithelial structure that expresses specific markers, forms during early stages of development and give rise to the inner ear. We showed in this manuscript the epithelialization of the otic placode (the formation of an otic epithelium). During this process, acquisition of epithelial character is restricted to a delimitated region, and thus the placode appears, and all the epithelial features as basal lamina and the apical lumen are established. Given that epithelialization occurs only in a restricted area of this embryonic region, cells that will be part of the later otic vesicle have to be located in the epithelializing area of the tissue (what we called resident cells) or come from other regions of the embryo to be incorporated during epithelialization. Thus, similar to what was reported for the ingression of neural crest cells into the mouse otic placode, here we report the ingression of migrating cells from an anterior region that does not belong to the placode. By definition, when a process of ingression occurs, the cells have to be outside the structure before to ingress. In our opinion, we have extensively proven that these cells are outside the epithelium of the placode. In Figure 2, Figure 2—figure supplement 1, Figure 4, Figure 4—figure supplement 1, Figure 5, Video 4, Video 5, Video 6 and Video 7 we provide all class of high resolution 4D imaging, individual cell tracking, tracing, photoconversion and ablation, proving that: 1) these cells migrate, 2) the migration occurs in individual cells, 3) during migration these cells present mesenchymal shape, 4) they are being epithelialized only in the region where the placode is epithelializing, 5) they stably incorporate into the placode, 6) cells that are about 40 µm apart form the placode move and incorporate in the placodal epithelium, 7) cells that are neurog1+ ingress in the placode, 8) neurog1+ ablated cells are about 40 µm apart from the epithelializing placode, and many other observations. Therefore, we do not understand why the editor proposes that these cells are all the time inside the otic epithelium (contrary to the placodal cells, pioneer cells do not have apicobasal polarity during migration and the epithelialization process is shown with great detail in our figures and videos). We think that saying that these cells are inside the epithelium is a misconception of what an epithelium is. This would lead to think that tissues are not separated, and to the conclusion that placodes do not exist and they are part of the neural tube. However, extensive evidence from long time ago, together with our high resolution imaging, firmly proves that each placode is an epithelializing independent tissue that will form a whole organ.

Additionally, is precisely the fact that the cells are outside the epithelium (a physical separation) what allow us to perform the ablation experiments and analyze the outcome in different type of cells (ingressing versus resident cells). Finally, if the editor thinks that these cells are inside the epithelium and do not ingress, we do not understand why is asking for the experiments in the following items 2-7, focused in the regulation of cell ingression. In summary, it would not be correct to change this part of the manuscript and say that at early stages these cells are already inside the placodal epithelium.

*2) The study begs the question of whether ngn1 is required for the pioneer cells, the ingression process, the otic epithelial cells to respond to the pioneer cells, or all of the above. It may be difficult to tease out all requirements of ngn1 but whether cell ingression proceed normally in the ngn1 knockout mutants should be investigated.*

The editor questions about the requirement of neurog1 in different processes. We understand that the “requirement in pioneer cells” refer to the requirement for the pioneer function. This function is associated to two properties of the ingressing cells: being the first neuronal progenitors to be specified, and instruct specification in the neighbouring resident placodal cells. The first property is inherently dependent on neurog1, as is precisely the expression of this gene what makes these cells the first to be specified. The instruction process also requires neurog1, as the outcome of this cell communication is neurog1 expression in the resident cells. If neurog1 is required for communication between these two group of cells is still unknown, but also is highly difficult to evaluate, as both type of cells express neurog1 and removal of neurog1 in only one type of cells is a complicated experiment. Finally, the editor asks for the evaluation of the importance of neurog1 for cell ingression. As we indicated previously, some neurog1- cells are able to ingress in the vesicle, suggesting that expression of neurog1 is not essential for this process. The editor suggests to test if cell ingression occurs in neurog1 mutant embryos. We performed an extensive analysis of the neurog1 mutant *neurog1^hi1059^* (Nature Genetics, 31, 135-140, 2002) embryos, and considered them not useful for our experiments because we observed a drastic effect of this mutation on cell proliferation. We found an increase in the number of dividing cells in the mutant compared to the heterozygotes embryos: pH3 immunostainings at 16hpf (Figure 7) or direct visualization of all mitotic events in time-lapses from 14 to 17 hpf (data not shown), indicate a 2-3 fold increase in the number of proliferating cells. Additionally, the vesicle and the neurogenic domain show increased number of cells at 18 hpf (Figure 8, quantified using the same method shown in Figure 1 of the manuscript). This leads to an increase of about 20% in the volume of the vesicle in the *neurog1^hi1059^/neurog1^hi1059^*compared to the *neurog1^hi1059^/+* embryos (1,21 +/- 0.01 fold, n= 4 and 6, respectively). These quantifications were performed before delamination becomes abundant and could contribute significantly to these measurements. These results indicate that cell proliferation, cell number and tissue volume are increased in the otic vesicle of *neurog1^hi1059^* mutant embryos, suggesting a defect in otic morphogenesis.. Therefore, using this mutant line for photoconversion experiments as the ones shown in Figure 5 of the manuscript could lead to erroneous or partial conclusions. For example, given that the cells to be photoconverted are determined by their position relative to the forming placode, the wrong organization of cells in the mutant could lead to the photoconversion of an erroneous pool of cells. Additionally, given that the cells divide at a higher rate, the quantification of the number of cells ingressed could also be overestimated.

In spite of these considerations, we have performed the photoconversion experiments, and some photconverted cells were still able to ingress in absence of neurog1 (Figure 9), once more suggesting that its expression is not essential for ingression. However, as we are aware of the limitations of these experiments using this mutant line, we consider that this conclusion is too weak to be incorporated in the manuscript. Moreover, although interesting, all these questions would add new information to the manuscript, but do not contribute to clarify our conclusions. We believe that the manuscript is long and includes a lot of information, and adding new questions will not necessarily improve the work. Therefore, we have decided to show this data to the editor but not to include it in the manuscript.

Author response image 2.Number of pH3+ cells at 16 hpf.* p<0.001, t test. n=12 for *neurog1^hi1059^/neurog1^hi1059^* and n=11 for *neurog1^hi1059^/*+.**DOI:**
http://dx.doi.org/10.7554/eLife.25543.028

Author response image 3.Number of cells in the otic vesicle (upper panel) or the neurogenic region (lower panel) at 18 hpf.* p<0.05, t test. n=4 for *neurog1^hi1059^/neurog1^hi1059^* and n=6 for *neurog1^hi1059^/*+.**DOI:**
http://dx.doi.org/10.7554/eLife.25543.029

Author response image 4.Cell ingression evaluated using NLS-Eos photoconversion in *neurog1^hi1059^* mutant embryos.The embryos were injected at 1 cell stage with memb-GFP and NLS-Eos mRNAs. For clarification, the genotype of the embryos was identified after imaging by PCR or phenotyping**DOI:**
http://dx.doi.org/10.7554/eLife.25543.030

*3) The authors concluded that FGF normally delays epithelial barrier formation in the anterior otic primordium. As a result, less pioneer cells get into the primordium to induce subsequent ngn1+ cells. The authors demonstrated a reduction of ngn1+ cells within the otic epithelium when Fgfr function was knockdown. If the authors are correct, one should expect a pileup of pioneer cells outside the placode. A video and quantification of the pioneer cells demonstrating this fact is lacking and warranted.*

The presence of a pileup is indeed a possibility if cells cannot ingress in dnfgfr1 embryos. However, other possibilities still could take place, given that because the formation of a pileup not only depends on the folding of the anterior region of the placode. One important factor is the migration of the pioneer cells. If cell migration is not stopped in the *Fgfr1* knockdown embryos, is possible that the cells that cannot ingress continue migrating to other regions, and a pileup would not form. Alternatively, the initial direction of cell migration can be also affected in the *Fgfr1* knockdown embryos, the cells migrate in other directions, and the pileup again will not form. Additionally, cell migration can be stopped in the *Fgfr1* knockdown embryos, and the pileup close to the placode will not form. It is also important to take into account that together with the defect in folding, other morphogenetic defects can also occur and affect the final location of the non-ingressed cells, perturbing the pileup formation or the location of the pileup. Remarkably, the second pool of neurog1+ cells also migrate in the direction of the placode without ingressing, and could also contribute to the formation or identification of the pileup in the *Fgfr1* knockdown embryos. These second pool of cells makes also difficult to quantify the exact number of cells present in a possible pileup. Finally, the number of ingressed cells that we observed with the photoconversion methodology is about 5, therefore mixed with the second pool of cells, a pileup can be difficult to recognize. We consider the quantification of the number of cells that ingress a much more reliable measurement of the process of cell ingression (Figure 11).

Anyhow, we have performed time lapses to track photoconverted cells in wild type and *Fgfr1* knockdown embryos to know more about what happens with some of the cells that are not able to ingress in the *Fgfr1* knockdown embryos. In the Figure 10 (coronal view), we show the tracking of some photoconverted cells that end inside the otic vesicle of wild type embryos, in agreement with the previous tracking and tracing experiments shown in Figure 2, Figure 2—figure supplement 1 and Figure 5 of the manuscript. Differently, in Figure 10 (sagittal view) and 5C (coronal view), photoconverted cells from Fgfr knockdown embryos migrate in the direction of the placode, (“chasing” the epithelium), but it folds before they reach the proper location. Therefore, the cells end outside of the vesicle (a kind of pileup) at the same stage than ingression already took place in wild type embryos. These results are in agreement with the ones shown in Figure 5 of the manuscript. Moreover, at later stages (21 hpf), these non-ingressed cells continue their movement ventrally to the placode. In summary, these videos confirm our previous results, and show the possible presence of a pileup. We think that these videos will not provide new relevant information to the manuscript.

Author response image 5.Tracking of photoconverted NLS-Eos nuclei in wild type (**A**) and *hsp70:dnfgfr1-EGFP/+* embryos (**B** and **C**).z-projections of coronal (A and C) or resliced sagittal (**B**) views are shown. These images were selected from 3D time lapses performed at 5 min time resolution, which allowed us to track individual cells over time. Arrowheads indicate examples of tracked nuclei (each color correspond to a different cell). In the two first time points of (**B**), it was not possible to track the cells indicated with white and pink arrowheads in the following time points, because they move laterally out and in of the video during their posterior migration.**DOI:**
http://dx.doi.org/10.7554/eLife.25543.031

*The premature basement membrane formation of the otic epithelium using anti-laminin staining is not convincing. While the dotted lines outline the border of the otic primordium, it also negated judgement of the data by readers.*

We don´t understand why the laminin staining is not convincing. The outlines of the placode were included overall the manuscript figures by requirement of the reviewers. However, in the laminin figure we explicitly did not draw the outline. This was done to visualize better the staining, given that an outline will in fact occlude the visualization. Thus, we are surprised by this comment.

*What happens with gain of FGF function models? Would there be an increase in cell ingression compare to wildtype? If so, would one conclude the rate-limiting step is the barrier formation of the anterior otic primordium. That is if FGF only functions in the otic epithelial cells and not in the pioneer cells.*

We showed that a reduction in FGF signaling accelerates the folding of the anterior part of the placode, indicating that endogenous FGF levels are delaying the folding. As suggested by the editor, increasing the FGF levels might delay even more the folding and favor more ingression. However, it is still possible that the endogenous levels of FGF are enough to inhibit a signal that induces folding at certain stage and adding more FGF does not further delay the folding because the FGF signaling pathway is already activated there. If this is so, an increase of cell ingression when FGF is overexpressed is not necessarily required to give support to our conclusions.

We have analyzed cell ingression in a gain of function model of FGF signaling. We overexpressed *FGF3* by applying a heat shock at 11 hpf in the *Tg(hsp70:fgf3)* line (Hammond and Whitfield, Development. 2011;138:3977). Our quantifications indicated that the number of ingressed cells does not change with respect to the wild type embryos (Figure 11). We also analyzed how morphogenesis takes place in this gain of *FGF3* function model, focusing on tissue epithelialization and folding. As we detected in our previous unpublished results, at 13 hpf the anteroposterior length of the placode is increased (Figure 11). However, the folding of the anterior part of the placode occurred normally, with the same kinetics than in wild type embryos (Figure 12). This result suggests that endogenous levels of FGFs are enough to control the proper folding of the anterior part, and that addition of more FGFs does not delay more the folding. Together with the loss of *FGFR1* function experiments, this also suggest that activity of the FGF pathway delays the folding during a restricted time window., but another signal is finally able to induce folding of the anterior region. This process seems to be independent of a later decrease in FGF levels (that would release the delaying effect) because even in presence of high levels of overexpressed *FGF3* at later times, the anterior folding still occurs normally. Surprisingly, when *FGF3* is overexpressed, the posterior part shows a large delay in folding. This delay could be explained by the absence of active FGF signaling in this posterior region in wild type embryos, and thus the over expression of *FGF3* becomes relevant in this region. Therefore, the presence of the induced *FGF3* would act in the posterior region as the endogenous FGF levels normally act in the anterior part of the placode in wild type embryos. These results are in agreement with a previous publication showing that overexpression of *FGF3* leads to acquisition of anterior features in the posterior part of the otic vesicle, while the anterior region remains unaffected (Hammond and Whitfield, Development. 2011;138:3977; note also the luminal deformation in Figure 6, arrowhead). Altogether, these results confirm our observations about a role of FGF signaling on the timing of folding of the placodal epithelium. Integrating both results (the quantification of cell ingression and the folding of the epithelium), we observed that *FGF3* overexpression did not change cell ingression neither the anterior folding of the placode. Thus, these experiments support our results (a putative increase in ingression without a change in folding would contradict instead of support our proposal about the role of folding on ingression). Given the complexity to explain all these results and that they are not adding something really new to the manuscript, we think is better not to include all these data. If the editor considers that including these data could help instead of confuse the readers, we will add them.

About the role of FGF acting exclusively in the folding and not in the pioneer cells, we cannot discard this possibility. We observed that there are still cells migrating in the loss of function of FGF (Figure 10) but small defects in migration can be also present. We already mentioned this possibility in the Discussion section of the manuscript.

Author response image 6.Cell ingression evaluated using NLS-Eos photoconversion in *FGF3* overexpressing embryos.*FGF3* expression was induced with a heat shock by incubating the 11 hpf embryos for 30 min at 39 degrees. For this purpose, the *Tg(hsp70:fgf3)* line was used. The embryos were injected at 1 cell stage with memb-GFP and NLS-Eos mRNAs. The quantification of the ingressed cells is shown in the right panel (no significant differences were found between the two experimental groups, t test, n=5 for +/+ embryos and n=6 for *Tg(hsp70:fgf3)*/+ embryos).**DOI:**
http://dx.doi.org/10.7554/eLife.25543.032

Author response image 7.Analysis of tissue folding during placode formation in *FGF3* overexpressing embryos.*FGF3* expression was induced with a heat shock by incubating the 11 hpf embryos for 30 min at 39 degrees. For this purpose, the *Tg(hsp70:fgf3)* line was used. The embryos were injected at 1 cell stage with memb-GFP. (**A**) Note that the anterior region folds before the posterior one, which remains incompletely folded at late stages (20hf). Arrowhead: a deformation of the lumen can be observed in the posterior region of the vesicle, which is found only anteriorly in wild type embryos and associated to the unfolded tissue. (**B**) Detail of the posterior unfolded tissue.**DOI:**
http://dx.doi.org/10.7554/eLife.25543.033

*4) In the Results section the argument claiming that small differences in the cell number but large differences in the mitotic event found in the neurogenic domain suggests an increase in proliferative activity is unclear. A high mitotic event should result in an increase in cell number at a later time point, at the least, or the moderate increase in cell number is due to neuroblast delamination.*

We already explained this argument in the previous answer to the reviewers, we now explain it again using a scheme (Figure 13). We propose in the manuscript that the option 2 is occurring: the neurogenic domain presents an increased number of mitosis because the proliferative activity is increased in the region, and not only because the domain has more cells. In this sentence, we only discussed about the influence of cell number on the number of mitosis because this is what allowed us to speculate about a putative stimulation of mitosis specifically in this region. However, as the editor mentioned, the opposite logic could be applied, and thus is possible to analyze the influence of mitosis in the final cell number. In this case, it is clear that before delamination begins to contribute significantly (from 19 hpf), if cell death or cell movement are not playing a role, more mitosis should increase more the number of cells. We did not discuss this because it was not relevant for our analysis.

Author response image 8.Options 1 and 2 are two different situations that can be underlying the measurements of cell number and cell division.A and B are the two regions of tissue to be compared. The circles are cells; the green circles indicate mitotic cells. In option 1, the tissue A has much less number of cells and mitosis than tissue B, but the proliferation rate (mitotic index, the probability of a cell to divide) is not different between the two tissues. In this case, the reduction in mitosis in A is just a consequence of a reduced number of cells in this tissue. In option 2, the tissue A has a few less cells than tissue B but much less mitotic events. Thus, A and B present a big difference in proliferation rate. It is possible to think that some signal stimulate proliferation in B. The observed difference in the number of mitosis is not a simple consequence of a different number of cells in the tissue, but of a different proliferative activity**DOI:**
http://dx.doi.org/10.7554/eLife.25543.034

*5) The authors cited a reference by Raible's lab in regards to ngn1 gene expression in the pioneer cells. The ngn1 expression in Figure 2 of Raible's paper is later than the onset of pioneer cells ingressing the otic epithelium described here. It may be a good idea to determine whether ngn1 transcripts are detected in the pioneer cells. It will complement the functional study of ngn1 suggested above.*

In the Raible´s paper, they showed neurog1 expression anterior to the placode in the region of the pioneer cells at 14 hpf, before these cells have ingressed in the otic placode. To confirm these results, we have analyzed neurog1 expression by in situ hybridization at 13 and 14 hpf in the otic placode and its anterior region (we now added these figures to Figure 2—figure supplement 1, panel A). We observed neurog1+ cells anterior to the otic placode, in the same region that the DsRed+ pioneer cells are located.

*6) The illustration in Figure 4 is not helpful to describe all the scenarios that the authors are considering. At least, the authors should describe/cite Figure 4 Iii in the text. The text in the legend also needs clarification.*

We now added the citation to the Figure 4Iii in the new version of the manuscript. We corrected the mistake in the text and it now it states “However, Figure 4 shows that neurog1+Hi cells” instead of “Figure 4”.

*7) The summary diagram in Figure 5 is misleading and does not serve the manuscript. The summary diagram implies that FGF signaling is required for cell ingression, but according to the authors, FGF signaling mediates cell ingression indirectly by delaying epithelial barrier formation.*

Our photoconversion experiments performed with the dnfgfr1 and SU5402, show that FGF signaling is required for cell ingression. If this pathway acts directly or indirectly is not mentioned in the manuscript, therefore cannot be a misleading of the figure. Specifically, in the figure only an arrow is depicted, and nothing is claimed about the effect being direct or indirect. Arrows in diagrams not always depicts direct effects. With the arrow we depict the effect of FGF on cell ingression. Importantly, for the cells to ingress, different events have to occur, including an epithelium that “allows” the addition of cells, and cells that are “able” to incorporate in the tissue. We showed that the FGF signaling is regulating the capacity of the epithelium to allow cells be incorporated, and thus is affecting ingression directly. Independently of all the affected events that interfere with ingression, the outcome of our experiment is the process of cell ingression, which is what the arrows wants to represent and what is explained in the text.

At the minimum, the summary diagram should be flipped so that the medial region is towards the top to be consistent with the videos and images shown in the figures.

We did not understand what the editor refers to. The medial region is towards the top on the diagram, the orientation is the same that the one used all along the manuscript. This orientation is clearly recognizable because the neurogenic domain, the region where all the cellular events occur, is depicted in the left down corner of the image, corresponding to the anterolateral region of the vesicle. This region is shown in the same location during the formation of the neurogenic domain (Figure 1), the ingression of the neurog1+ cells (Figure 2), the local specification and cell divisions (Figure 3), the analysis of neurog1 expression pattern (Figure 4) and the ingression of the photoconverted cells (Figure 5).

*The authors should modify the diagram; as it is it somehow suggests that it affects the incorporation of cells into the vesicle.*

We did not understand the meaning of this phrase. We indeed proposed and showed that the FGF signaling is affecting ingression (or “incorporation of cells into the vesicle“, see Figure 5).

[Editors' note: further revisions were requested prior to acceptance, as described below.]

*Reviewer #1:*

*While it seems we are in a silly semantic argument I think the point I made in the original review still stands.*

*I would argue that the otic primordium/placode is the field of all cells that subsequently form the otic vesicle. I interpret the authors results to show that not all otic primoprdium/placode cells join the otic vesicle at once, as some cells ingress into the otic vesicle epithelium after it forms. I originally pointed out that it looked like cells other than the ones designated as neuronal pioneer cells were also ingressing. As the zebrafish otic vesicle forms by coalescence rather than the epithelial infolding and delamination described for amniote embryos, this type of MET would perhaps be expected.*

*If a whole bunch of cells are ingressing into the otic vesicle then the behavior is not specific to the neuronal pioneer cells. Indeed I would argue that if neuronal pioneer cells and their neighbors are all joining the otic vesicle, then these cells and their neighbors are part of the otic primordium/placode even if they join the vesicle after it has formed an epithelium. However if they are interspersed with other cells that do not join the otic vesicle, that would be evidence for a distinct behavior more along the lines of what the authors are implying. The authors were unwilling to perform the timelapse analysis and in toto lineage tracing to address this point. I agreed that in toto imaging is not necessary for the main conclusions of the current work. But I do not believe that the authors should state that ingression is a special behavior unique to pioneer cells, which is what is implied as the text currently stands.*

We agree with the reviewer that the discussion is semantic. To circumvent this problem, we have replaced the word “primordium” by “epithelium” (when we talk about cell ingression) or by “placode” (in other cases). This include changes in the title, the Abstract and throughout the text. We believe that now we solved the discussion about if the migrating cells can be or not conceived as part of the otic placode, because now we refer to an ingression into the epithelium.

We never stated that ingression only occurs in neurog1+ cells, but we now also added a sentence to highlight that a few neurog1- cells could also be incorporated into the epithelium. We also removed the word “group” in some sentences to avoid the perception that ingression is a unique behavior of the pioneer cells. We have now included in the manuscript a new panel in Figure 2—figure supplement 1 showing the photoconversion experiments with the *neurog1* mutant embryos. These experiments are mentioned in a new sentence stating that neurog1 expression does not seem to be required for ingression, neither sufficient for this process (pool of neurog1+ cells that do not ingress shown in Figure 2, Figure 2—figure supplement 1, and Video 5). Additionally, we also added a sentence to highlight the results of Figure 2, in which is shown that ingression is a behavior of only some cells of this region, given that they are interspersed with cells that do not ingress.

*Reviewer #2:*

*The authors have addressed most points raised in previous review.*

*They did not change the summary diagram in Figure 5; if I remember correctly all three reviewers raised the point that the summary does not accurately reflect the findings of the authors indepdently. THis suggests that the message the authors wish to convey is not clear. In its current form the diagram suggests that FGF affects the behavior of ingressing cells directly, but the authors show that it affectes the formation of the epithelial barries – this is how arrows are interpreted even if this is not the intention of the authors. It should not be very difficult to change this to make the figure accurately reflect their findings.*

We have now changed the diagram according to the suggestion of the reviewer. We also made some changes to the text to clarify the role of FGF in epithelialization.

*Reviewer #3:*

*Specific comments:*

*Ngn1 not required for the ingression process: The lack of cell ingression in the ngn1 null mutants is interesting and is relevant information. I don't see why it should not be incorporated into the manuscript with some additional quantification.*

We have now added these results suggesting that ingression occurs normally in the neurog1 null mutants, as a panel in Figure 2—figure supplement 1 (and a sentence in subsection “The first otic neurogenic cells are specified outside the otic epithelium and ingress during placode formation”).

*Knockdown of FGF signaling reduced ingression of ngn1-positive cells: Although there is no strong evidence for the predicted ngn1 positive cells piling up outside the FGF lof placode, the authors proposed several alternate explanations for this observation, which are acceptable. The authors provided additional FGF gain-of-function results, which further supported a role of FGF in regulating barrier formation in the otic placode. These results, together with the lof experiments, argue against a direct role of FGF in the ingression process.*

*Anti-laminin staining: A higher magnification of the control in a comparable region as the treated sample would help to convince readers. The anti-laminin staining in the mutants, though seems stronger than controls, is blotchy and discontinuous and raises the question whether there is indeed a basement membrane barrier. However, the additional gain-of-FGF function experiments supported a normal role of FGF in delaying barrier formation.*

We have now added the *FGF3* overexpression experiments (Figure 5—figure supplement 1) and in subsection “FGF controls otic epithelialization”. We also added the tracking of the photoconverted NLS-Eos nuclei performed in the DNFGFR1-GFP expressing embryos (Figure 5—figure supplement 1, and a sentence in subsection “FGF controls otic epithelialization”).

We have added the high magnification figure of the control in the laminin staining.

*I appreciate the authors' attempt to explain their point of view. In Results paragraph two, it states "The large difference in the change of the two parameters (meaning low cell number increase but high percentage of mitotic events) suggests that the increased number of mitotic events is not a consequence of having more cell dividing at the same rate, but due to a specific increase in the proliferative activity of these cells". I interpreted this sentence to mean that there were not more cells going into cell division but the same cells are dividing faster, which contributed to the higher percentages of mitotic counts. If so, I am confused because I thought the authors measured mitotic events rather than measured how fast each cell go through the cell cycle. Proliferate activity and proliferate rate are not interchangeable terms in my mind. Increased proliferate activity (mitotic events) could be caused by either increased proliferate rate or increased number of cells in division. Either scenario should result in an increased in cell number. The moderate increase in cell number observed at the time of measurement could be simply due to timing. The authors measured mitotic events between 14 to 18.5 hpf. On average, each cell cycle takes about 8-12 hrs. Considering the cells have just reached the placode at 13-14 hpf, peak increase in cell number may not be apparent until after 18.5 hpf. By this time, some neuroblasts will start to delaminate from the epithelium, which will also reduce the total cell number in the region.*

*I do not find the option 1 and 2 described in the rebuttal letter particularly helpful, partly due of the interchangeable terms of proliferative rate and proliferative activity.*

As this leads to confusion and is not relevant, we have removed this sentence from the text, and also changed the term “proliferation rate”.

*Summary diagram: I might have over-interpreted those videos. Based on the examples shown in the videos, it appears that ingression of cells is occurring at the lateral edge of the placode and delamination is slightly medial to where ingression takes place. This pattern is consistent with the position of the statoacoustic ganglion being located medial to the otocyst. If this is correct, the summary diagram should reflect the spatial relationship between ingression and delamination.*

We agree that this is an overinterpretation because delamination takes place in different regions of the vesicle depending on the developmental time and plane of imaging. To avoid confusions in the readers, we have inverted the spatial distribution of these events in the diagram.

*I still think the arrow of the FGF in the summary diagram is misleading, but it is up to the authors to decide the best way to summarize their data.*

We have changed the diagram accordingly.